# Personalized Federated Learning: A Unified Framework and Universal Optimization Techniques

**Filip Hanzely**[*]                                                                                                        *fhanzely@gmail.com*
*Toyota Technological Institute at Chicago*
*Chicago, IL 60637, USA*

**Boxin Zhao**[*]                                                                                                          *boxinz@uchicago.edu*
**Mladen Kolar**                                                                                                 *mkolar@chicagobooth.edu*
*The University of Chicago Booth School of Business*
*Chicago, IL 60637, USA*

Reviewed on OpenReview: *https://openreview.net/forum?id=iLHM31lXC4*

## Abstract

We investigate the optimization aspects of personalized Federated Learning (FL). We propose general optimizers that can be applied to numerous existing personalized FL objectives, specifically a tailored variant of Local SGD and variants of accelerated coordinate descent/accelerated SVRCD. By examining a general personalized objective capable of recovering many existing personalized FL objectives as special cases, we develop a comprehensive optimization theory applicable to a wide range of strongly convex personalized FL models in the literature. We showcase the practicality and/or optimality of our methods in terms of communication and local computation. Remarkably, our general optimization solvers and theory can recover the best-known communication and computation guarantees for addressing specific personalized FL objectives. Consequently, our proposed methods can serve as universal optimizers, rendering the design of task-specific optimizers unnecessary in many instances.

## 1 Introduction

Modern personal electronic devices, such as mobile phones, wearable devices, and home assistants, can collectively generate and store vast amounts of user data. This data is essential for training and improving state-of-the-art machine learning models for tasks ranging from natural language processing to computer vision. Traditionally, the training process was performed by first collecting all the data into a data center (Dean et al., 2012), raising serious concerns about user privacy and placing a considerable burden on the storage capabilities of server suppliers. To address these issues, a novel paradigm – Federated Learning (FL) (McMahan et al., 2017; Kairouz et al., 2021) – has been proposed. Informally, the main idea of FL is to train a model locally on an individual's device, rather than revealing their data, while communicating the model updates using private and secure protocols.

While the original goal of FL was to search for a single model to be deployed on each device, this objective has been recently questioned. As the distribution of user data can vary greatly across devices, a single model might not serve all devices simultaneously (Hard et al., 2018). Consequently, data heterogeneity has become the main challenge in the search for efficient federated learning models. Recently, a range of personalized FL approaches has been proposed to address data heterogeneity (Kulkarni et al., 2020), wherein different local models are used to fit user-specific data while also capturing the common knowledge distilled from data of other devices.

---

[*]equal contribution

Since the motivation and goals of each of these personalized approaches vary greatly, examining them separately can only provide us with an understanding of a specific model. Fortunately, many personalized FL models in the literature are trained by minimizing a specially structured optimization program. In this paper, we analyze the general properties of such an optimization program, which in turn provides us with high-level principles for training personalized FL models. We aim to solve the following optimization problem

$$\min_{w,\beta} \left\{ F(w,\beta) := \frac{1}{M} \sum_{m=1}^{M} f_m(w,\beta_m) \right\}, \tag{1}$$

where $w \in \mathbb{R}^{d_0}$ corresponds to the shared parameters, $\beta = (\beta_1, \ldots, \beta_M)$ with $\beta_m \in \mathbb{R}^{d_m}$, $\forall m \in [M]$ corresponds to the local parameters, $M$ is the number of devices, and $f_m : \mathbb{R}^{d_0 + d_m} \to \mathbb{R}$ is the objective that depends on the local data at the $m$-th client.

By carefully designing the local loss $f_m(w, \beta_m)$, the objective (1) can recover many existing personalized FL approaches as special cases. The local objective $f_m$ does not need to correspond to the empirical loss of a given model on the $m$-th device's data. See Section 2 for details. Consequently, (1) serves as a unified objective that encompasses numerous existing personalized FL approaches as special cases. The primary goal of our work is to explore the problem (1) from an optimization perspective. By doing so, we develop a universal convex optimization theory that applies to many personalized FL approaches.

## 1.1 Contributions

We outline the main contributions of this work.

**Single personalized FL objective**. We propose a single objective (1) capable of recovering many existing convex personalized FL approaches by carefully constructing the local loss $f_m(w, \beta_m)$. Consequently, training different personalized FL models is equivalent to solving a particular instance of (1).

**Recovering best-known complexity and novel guarantees**. We develop algorithms for solving (1) and prove sharp convergence rates for strongly convex objectives. Specializing our rates from the general setting to the individual personalized FL objectives, we recover the best-known optimization guarantees from the literature or advance beyond the state-of-the-art with a single exception: objective (11) with $\lambda > L'$. Therefore, our results often render optimization tailored to solve a specific personalized FL unnecessary in many cases.

**Universal (convex) optimization methods and theory for personalized FL**. To develop an optimization theory for solving (1), we impose particular assumptions on the objective: $\mu-$strong convexity of $F$ and convexity and $(L^w, ML^\beta)$-smoothness of $f_m$ for all $m \in [M]$ (see Assumptions 1, 2). These assumptions are naturally satisfied for the vast majority of personalized FL objectives in the literature, with the exception of personalized FL approaches that are inherently nonconvex, such as MAML (Finn et al., 2017). Under these assumptions, we propose three algorithms for solving the general personalized FL objective (1): i) Local Stochastic Gradient Descent for Personalized FL (LSGD-PFL), ii) Accelerated Block Coordinate Descent for Personalized FL (ACD-PFL), and iii) Accelerated Stochastic Variance Reduced Coordinate Descent for Personalized FL (ASVRCD-PFL). The convergence rates of these methods are summarized in Table 1. We emphasize that these optimizers *can be used to solve many (convex) personalized FL objectives from the literature by casting a given objective as a special case of* (1), *oftentimes matching or outperforming algorithms originally designed for the particular scenario.*

**Minimax optimal rates**. We provide lower complexity bounds for solving (1). Using the construction of Hendrikx et al. (2021), we show that to solve (1), one requires at least $\mathcal{O}\left(\sqrt{L^w/\mu} \log \epsilon^{-1}\right)$ communication rounds. Note that communication is often the bottleneck when training distributed and personalized FL models. Furthermore, one needs at least $\mathcal{O}\left(\sqrt{L^w/\mu} \log \epsilon^{-1}\right)$ evaluations of $\nabla_w F$ and at least $\mathcal{O}\left(\sqrt{L^\beta/\mu} \log \epsilon^{-1}\right)$ evaluations of $\nabla_\beta F$. Given the $n$-finite sum structure of $f_m$ with $(\mathcal{L}^w, M\mathcal{L}^\beta)$-smooth components, we show that one requires at least $\mathcal{O}\left(n + \sqrt{n\mathcal{L}^w/\mu} \log \epsilon^{-1}\right)$ stochastic gradient evaluations with respect to $w$-parameters and at least $\mathcal{O}\left(n + \sqrt{n\mathcal{L}^\beta/\mu} \log \epsilon^{-1}\right)$ stochastic gradient evaluations with

| Alg. | Communication | $\# \nabla_w$ | $\# \nabla_\beta$ |
|---|---|---|---|
| LSGD-PFL | $\dfrac{\max(L^\beta \tau^{-1}, L^w)}{\mu} + \dfrac{\sigma^2}{MB\tau\mu\epsilon} + \dfrac{1}{\mu}\sqrt{\dfrac{L^w(\zeta_*^2 + \sigma^2 B^{-1})}{\epsilon}}$ | $\dfrac{\max(L^\beta, \tau L^w)}{\mu} + \dfrac{\sigma^2}{MB\mu\epsilon} + \dfrac{\tau}{\mu}\sqrt{\dfrac{L^w(\zeta_*^2 + \sigma^2 B^{-1})}{\epsilon}}$ | $\dfrac{\max(L^\beta, \tau L^w)}{\mu} + \dfrac{\sigma^2}{MB\mu\epsilon} + \dfrac{\tau}{\mu}\sqrt{\dfrac{L^w(\zeta_*^2 + \sigma^2 B^{-1})}{\epsilon}}$ |
| ACD-PFL | $\sqrt{L^w/\mu}$ ✿ | $\sqrt{L^w/\mu}$ ✿ | $\sqrt{L^\beta/\mu}$ ✿ |
| ASVRCD-PFL | $n + \sqrt{n\mathcal{L}^w/\mu}$ | $n + \sqrt{n\mathcal{L}^w/\mu}$ ✿ | $n + \sqrt{n\mathcal{L}^\beta/\mu}$ ✿ |

Table 1: Complexity guarantees of the proposed methods when ignoring constant and log factors. $\#\nabla_w / \#\nabla_\beta$: number of (stochastic) gradient calls with respect to the $w/\beta$-parameters. Symbol ✿, indicates the minimax optimal complexity. Local Stochastic Gradient Descent (LSGD): Local access to $B$-minibatches of stochastic gradients, each with $\sigma^2$-bounded variance. Each device takes $(\tau - 1)$ local steps in between the communication rounds. Accelerated Coordinate Descent (ACD): Access to the full local gradient, yielding both the optimal communication complexity and the optimal computational complexity (both in terms of $\nabla_w$ and $\nabla_\beta$). ASVRCD: Assuming that $f_i$ is an $n$-finite sum, the oracle provides access to a single stochastic gradient with respect to that sum. The corresponding local computation is either optimal with respect to $\nabla_w$ or with respect to $\nabla_\beta$. Achieving both optimal rates simultaneously remains an open problem.

respect to $\beta$-parameters. We show that ACD-PFL is always optimal in terms of communication and local computation when the full gradients are available, while ASVRCD-PFL can be optimal either in terms of the number of evaluations of the $w$-stochastic gradient or the $\beta$-stochastic gradient. However, note that ASVRCD-PFL cannot achieve optimal rate for both evaluations of the $w$-stochastic gradient and the $\beta$-stochastic gradient simultaneously, which we leave for future research.

**Personalization and communication complexity**. Given that a specific FL objective contains a parameter that determines the amount of personalization, we observe that the value of $\sqrt{L^w/\mu}$ is always a non-increasing function of this parameter. Since the communication complexity of (1) is equal to $\sqrt{L^w/\mu}$ up to constant and log factors, we conclude that personalization has a positive effect on the communication complexity of training an FL model.

**New personalized FL objectives**. The universal personalized FL objective (1) enables us to obtain a range of novel personalized FL formulations as special cases. While we study various (parametric) extensions of known models, we believe that the objective (1) can lead to easier development of brand new objectives as well. However, we stress that proposing novel personalized FL models is not the main focus of our work; the paper's primary focus is on providing universal optimization guarantees for (convex) personalized FL.

Despite the aforementioned benefits of our proposed unified framework, we acknowledge that this is neither the only nor the universally best approach for personalized federated learning. However, providing a general framework that can include many existing methods as special cases can help us gain a clear understanding and motivate us to propose new personalized methods.

## 1.2 Assumptions and Notations

**Complexity Notations.** For two sequences $\{a_n\}$ and $\{b_n\}$, $a_n = \mathcal{O}(b_n)$ if there exists $C > 0$ such that $|a_n/b_n| \leq C$ for all $n$ large enough; $a_n = \Theta(b_n)$ if $a_n = \mathcal{O}(b_n)$ and $b_n = \mathcal{O}(a_n)$ simultaneously. Similarly, $a_n = \tilde{\mathcal{O}}(b_n)$ if $a_n = \mathcal{O}(b_n \log^k b_n)$ for some $k \geq 0$; $a_n = \tilde{\Theta}(b_n)$ if $a_n = \tilde{\mathcal{O}}(b_n)$ and $b_n = \tilde{\mathcal{O}}(a_n)$ simultaneously.

**Local Objective.** We assume three different ways to access the gradient of the local objective $f_m$. The first, and the simplest case, corresponds to having access to the full gradient of $f_m$ with respect to either $w$ or $\beta_m$ for all $m \in [M]$ simultaneously. The second case corresponds to a situation where $\nabla f_m(w, \beta_m)$ is the expectation itself, i.e.,

$$\nabla f_m(w, \beta_m) = \mathbb{E}_{\xi \in \mathcal{D}_m}\left[\nabla \hat{f}_m(w, \beta_m; \xi)\right], \tag{2}$$

while having access to stochastic gradients with respect to either $w$ or $\beta_m$ simultaneously for all $m \in M$, where $\hat{f}_m$ represents the loss function on a single data point. The third case corresponds to a finite sum $f_m$:

$$f_m(w, \beta_m) = \frac{1}{n} \sum_{i=1}^{n} f_{m,i}(w, \beta_m), \tag{3}$$

having access to $\nabla_w f_{m,i}(w, \beta_m)$ or to $\nabla_\beta f_{m,i}(w, \beta_m)$ for all $m \in [M]$ and $i \in [n]$ selected uniformly at random.

**Assumptions.** We argue that the objective (1) is capable of recovering virtually any (convex) personalized FL objective. Since the structure of the individual personalized FL objectives varies greatly, it is important to impose reasonable assumptions on the problem (1) in order to obtain meaningful rates in the special cases.

**Assumption 1.** *The function $F(w, \beta)$ is jointly $\mu$-strongly convex for $\mu \geq 0$, while for all $m \in [M]$, function $f_m(w, \beta_m)$ is jointly convex, $L^w$-smooth w.r.t. parameter $w$ and $(ML^\beta)$-smooth w.r.t. parameter $\beta_m$. In the case when $\mu = 0$, assume additionally that (1) has a unique solution, denoted as $w^\star$ and $\beta^\star = (\beta_1^\star, \ldots, \beta_M^\star)$.*

When $f_m$ is a finite sum (3), we require the smoothness of the finite sum components.

**Assumption 2.** *Suppose that for all $m \in [M], i \in [n]$, function $f_{m,i}(w, \beta_m)$ is jointly convex, $\mathcal{L}^w$-smooth w.r.t. parameter $w$ and $(M\mathcal{L}^\beta)$-smooth w.r.t. parameter $\beta_m$.*[1]

In Section 2 we justify Assumptions 1 and 2 and characterize the constants $\mu$, $L^w$, $L^\beta$, $\mathcal{L}^w$, $\mathcal{L}^\beta$ for special cases of (1). Table 2 summarizes these parameters.

**Price of generality.** Since Assumption 1 is the only structural assumption we impose on (1), one cannot hope to recover the minimax optimal rates, that is, the rates that match the lower complexity bounds, for all individual personalized FL objectives as a special case of our general guarantees. Note that any given instance of (1) has a structure that is not covered by Assumption 1, but can be exploited by an optimization algorithm to improve either communication or local computation. Therefore, our convergence guarantees are optimal in light of Assumption 1 only. Despite this, our general rates specialize surprisingly well as we show in Section 2: our complexities are state-of-the-art in all scenarios with a single exception: the communication/computation complexity of (11).

**Individual treatment of $w$ and $\beta$.** Throughout this work, we allow different smoothness of the objective with respect to global parameters $w$ and local parameters $\beta$. At the same time, our algorithm is allowed to exploit the separate access to gradients with respect to $w$ and $\beta$, given that these gradients can be efficiently computed separately. Without such a distinction, one might not hope for the communication complexity better than $\Theta\left(\max\{L^w, L^\beta\}/\mu \log \epsilon^{-1}\right)$, which is suboptimal in the special cases. Similarly, the computational guarantees would be suboptimal as well. See Section 2 for more details.

**Data heterogeneity.** While the convergence rate of LSGD-PFL relies on data heterogeneity (See Theorem 1), we allow for an arbitrary dissimilarity among the individual clients for analyzing ACD-PFL and ASVRCD-PFL (see Theorem 7 and Theorem 8). Our experimental results also support that ASCD-PFL (ACD-PFL with stochastic gradient to reduce computation) and ASVRCD-PFL are more robust to data heterogeneity compared to the widely used Local SGD.

The rest of the paper is organized as follows. In Section 2, we show how (1) can be used to recover various personalized federated learning objectives in the literature. In Section 3, we propose a local-SGD based algorithm, LSGD-PFL, for solving (1). We further establish computational upper bounds for LSGD-PFL in strongly convex, weakly convex, and nonconvex cases. In Section 4, we discuss the minimax optimal algorithms for solving (1). We first show the minimax lower bounds in terms of the number of communication rounds, number of evaluations of the gradient of global parameters, and number of evaluations of the gradient of local parameters, respectively. We subsequently propose two coordinate-descent based algorithms, ACD-PFL and ASVRCD-PFL, which can match the lower bounds. In Section 5 and Section 6, we use experiments on synthetic and real data to illustrate the performance of the proposed algorithms and empirically validate the theorems. Finally, we conclude the paper with Section 7. Technical proofs are deferred to the Appendix.

---

[1]It is easy to see that $\mathcal{L}^w \geq L^w \geq \frac{\mathcal{L}^w}{n}$ and $\mathcal{L}^\beta \geq L^\beta \geq \frac{\mathcal{L}^\beta}{n}$.

## 2   Personalized FL objectives

We recover a range of known personalized FL approaches as special cases of (1). In this section, we detail the optimization challenges that arise in each one of the special cases. We discuss the relation to our results, particularly focusing on how Assumptions 1, 2 and our general rates (presented in Sections 3 and 4) behave in the special cases. Table 2 presents the smoothness and strong convexity constants with respect to (1) for the special cases, while Table 3 provides the corresponding convergence rates for our methods when applied to these specific objectives. For the sake of convenience, define

$$F_i(w, \beta) := \frac{1}{M} \sum_{m=1}^{M} f_{m,i}(w, \beta_m).$$

in the case when functions $f_m$ have a finite sum structure (3).

### 2.1   Traditional FL

The traditional, non-personalized FL objective (McMahan et al., 2017) is given as

$$\min_{w \in \mathbb{R}^d} F'(w) := \frac{1}{M} \sum_{m=1}^{M} f'_m(w), \tag{4}$$

where $f'_m$ corresponds to the loss on the $m$-th client's data. Assume that $f'_m$ is $L'$-smooth and $\mu'-$ strongly convex for all $m \in [M]$. The minimax optimal communication to solve (4) up to $\epsilon$-neighborhood of the optimum is $\tilde{\Theta}\left(\sqrt{L'/\mu'} \log \epsilon^{-1}\right)$ (Scaman et al., 2018). When $f'_m = \frac{1}{n} \sum_{j=1}^{n} f'_{m,j}(w)$ is an $n$-finite sum with convex and $\mathcal{L}'-$smooth components, the minimax optimal local stochastic gradient complexity is $\tilde{\Theta}\left(\left(n + \sqrt{n\mathcal{L}'/\mu}\right) \log \epsilon^{-1}\right)$ (Hendrikx et al., 2021). The FL objective (4) is a special case of (1) with $d_1 = \cdots = d_M = 0$ and our theory recovers the aforementioned rates.

### 2.2   Fully Personalized FL

At the other end of the spectrum lies the fully personalized FL where the $m$-th client trains their own model without any influence from other clients:

$$\min_{\beta_1, \ldots, \beta_M \in \mathbb{R}^d} F_{full}(\beta) := \frac{1}{M} \sum_{m=1}^{M} f'_m(\beta_m). \tag{5}$$

The above objective is a special case of (1) with $d_0 = 0$. As the objective is separable in $\beta_1, \ldots, \beta_M$, we do not require any communication to train it. At the same time, we need $\tilde{\Theta}\left(\left(n + \sqrt{n\mathcal{L}'/\mu}\right) \log \epsilon^{-1}\right)$ local stochastic oracle calls to solve it (Lan & Zhou, 2018) – which is what our algorithms achieve.

### 2.3   Multi-Task FL of Li et al. (2020)

The objective is given as

$$\min_{\beta_1, \ldots, \beta_M \in \mathbb{R}^d} F_{MT}(\beta) = \frac{1}{M} \sum_{i=1}^{M} \left(f'_m(\beta_m) + \frac{\lambda}{2}\|\beta_m - (w')^*\|^2\right), \tag{6}$$

where $(w')^*$ is a solution of the traditional FL in (4) and $\lambda \geq 0$. Assuming that $(w')^*$ is known (which Li et al. (2020) does), the problem (6) is a particular instance of (5); thus our approach achieves the optimal complexity.

A more challenging objective (in terms of optimization) is the following relaxed version of (6):

$$\min_{w, \beta} \frac{1}{M} \sum_{m=1}^{M} \left(\Lambda f'_m(w) + f'_m(\beta_m) + \lambda\|w - \beta_m\|^2\right), \tag{7}$$

| $F(w,\beta)$ | $\mu$ | $L^w$ | $L^\beta$ | $\mathcal{L}^w$ | $\mathcal{L}^\beta$ | Rate? |
|---|---|---|---|---|---|---|
| Traditional FL ((4)) (McMahan et al., 2017) | $\mu'$ | $L'$ | $0$ | $\mathcal{L}'$ | $0$ | recovered |
| Fully Personalized FL ((5)) | $\frac{\mu'}{M}$ | $0$ | $\frac{L'}{M}$ | $0$ | $\frac{\mathcal{L}'}{M}$ | recovered |
| MT2 ((8)) (Li et al., 2020) | $\frac{\lambda}{2M}$ | $\frac{\Lambda L'+\lambda}{2M}$ | $\frac{L'+\lambda}{2M}$ | $\frac{\Lambda\mathcal{L}'+\lambda}{2M}$ | $\frac{\mathcal{L}'+\lambda}{2M}$ | new♣ |
| MX2 ((11)) (Smith et al., 2017) | $\frac{\mu'}{3M}$ | $\frac{\lambda}{M}$ | $\frac{L'+\lambda}{M}$ | $\frac{\lambda}{M}$ | $\frac{\mathcal{L}'+\lambda}{M}$ | recovered♠ |
| APFL2 ((14)) (Deng et al., 2020) | $\frac{\mu'(1-\alpha_{max})^2}{M}$ | $\frac{(\Lambda+\alpha_{max}^2)L'}{M}$ | $\frac{(1-\alpha_{min})^2L'}{M}$ | $\frac{(\Lambda+\alpha_{max}^2)\mathcal{L}'}{M}$ | $\frac{(1-\alpha_{min})^2\mathcal{L}'}{M}$ | new♣ |
| WS2 ((16)) (Liang et al., 2020) | $\mu'$ | $L'$ | $L'$ | $\mathcal{L}'$ | $\mathcal{L}'$ | new |
| Fed Residual ((18)) (Agarwal et al., 2020) | $\mu$ | $L_R^w$ | $L_R^\beta$ | $\mathcal{L}_R^w$ | $\mathcal{L}_R^\beta$ | new |

Table 2: Parameters in Assumptions 1 and 2 for personalized FL objectives, with a note about the rate: we either recover the best-known rate for a given objective, or provide a novel rate that is the best under the given assumptions. ♣: Rate for the novel personalized FL objective (extension of a known one). ♠: Best-known communication complexity recovered only for $\lambda = \mathcal{O}(L')$.

| $F(w,\beta)$ | # Comm | $\#\nabla_w F$ | $\#\nabla_\beta F$ | $\#\nabla_w F_i$ | $\#\nabla_\beta F_i$ |
|---|---|---|---|---|---|
| Traditional FL ((4)) (McMahan et al., 2017) | $\sqrt{\frac{L'}{\mu'}}$ | $\sqrt{\frac{L'}{\mu'}}$ | $0$ | $n+\sqrt{\frac{n\mathcal{L}'}{\mu'}}$ | $0$ |
| Fully Personalized FL ((5)) | $0$ | $0$ | $\sqrt{\frac{L'}{\mu'}}$ | $0$ | $n+\sqrt{\frac{n\mathcal{L}'}{\mu'}}$ |
| MT2 ((8)) (Li et al., 2020) | $\sqrt{\frac{\Lambda L'}{\lambda}}$ | $\sqrt{\frac{\Lambda L'}{\lambda}}$ | $\sqrt{\frac{L'}{\lambda}}$ | $n+\sqrt{\frac{n\Lambda\mathcal{L}'}{\lambda}}$ | $n+\sqrt{\frac{n\mathcal{L}'}{\lambda}}$ |
| MX2 ((11)) (Smith et al., 2017) | $\sqrt{\frac{\lambda}{\mu'}}$ | $\sqrt{\frac{\lambda}{\mu'}}$ | $\sqrt{\frac{L'+\lambda}{\mu'}}$ | $-$ | $n+\sqrt{\frac{n(\mathcal{L}'+\lambda)}{\mu'}}$ |
| APFL2 ((14)) (Deng et al., 2020) | $\sqrt{\frac{(\Lambda+\alpha_{max}^2)L'}{(1-\alpha_{max})^2\mu'}}$ | $\sqrt{\frac{(\Lambda+\alpha_{max}^2)L'}{(1-\alpha_{max})^2\mu'}}$ | $\sqrt{\frac{(1-\alpha_{min})^2L'}{(1-\alpha_{max})^2\mu'}}$ | $n+\sqrt{\frac{n(\Lambda+\alpha_{max}^2)\mathcal{L}'}{(1-\alpha_{max})^2\mu'}}$ | $n+\sqrt{\frac{n(1-\alpha_{min})^2\mathcal{L}'}{(1-\alpha_{max})^2\mu'}}$ |
| WS2 ((16)) (Liang et al., 2020) | $\sqrt{\frac{L'}{\mu'}}$ | $\sqrt{\frac{L'}{\mu'}}$ | $\sqrt{\frac{L'}{\mu'}}$ | $n+\sqrt{\frac{n\mathcal{L}'}{\mu'}}$ | $n+\sqrt{\frac{n\mathcal{L}'}{\mu'}}$ |
| Fed Residual ((18)) (Agarwal et al., 2020) | $\sqrt{\frac{L_R^w}{\mu}}$ | $\sqrt{\frac{L_R^w}{\mu}}$ | $\sqrt{\frac{L_R^\beta}{\mu}}$ | $n+\sqrt{\frac{n\mathcal{L}_R^w}{\mu'}}$ | $n+\sqrt{\frac{n\mathcal{L}_R^\beta}{\mu'}}$ |

Table 3: Complexity of solving personalized FL objectives by Algorithms 2 (second, third, and fourth column) and 3 (fifth and sixth column). Constant and log factors are ignored.

where $\Lambda \geq 0$ is the relaxation parameter, recovering the original objective for $\Lambda \to \infty$. Note that, since $\Lambda \to \infty$, finding a minimax optimal method for the optimization of (6) is straightforward. First, one has to compute a minimizer $(w')^*$ of the classical FL objective (4), which can be done with a minimax optimal complexity. Next, one needs to compute the local solution $\beta_m^* = \arg\min_{\beta_m \in \mathbb{R}^d} f_m'(\beta_m) + \lambda \|w^* - \beta_m\|^2$, which only depends on the local data and thus can also be optimized with a minimax optimal algorithm.

A more interesting scenario is obtained when we do not set $\Lambda \to \infty$ in (7), but rather consider a finite $\Lambda > 0$ that is sufficiently large. To obtain the right smoothness/strong convexity parameter (according to Assumption 1), we scale the global parameter $w$ by a factor of $M^{-\frac{1}{2}}$ and arrive at the following objective:

$$\min_{w, \beta_1, \ldots, \beta_M \in \mathbb{R}^d} F_{MT2}(w, \beta) = \frac{1}{M} \sum_{m=1}^{M} f_m(w, \beta_m), \tag{8}$$

where

$$f_m(w, \beta_m) = \Lambda f_m'(M^{-\frac{1}{2}}w) + f_m'(\beta_m) + \frac{\lambda}{2}\|\beta_m - M^{-\frac{1}{2}}w\|^2.$$

The next lemma determines parameters $\mu, L^w, L^\beta, \mathcal{L}^w, \mathcal{L}^\beta$ in Assumption 1. See the proof in Appendix B.1.

**Lemma 1.** *Let $\Lambda \geq 3\lambda/(2\mu')$. Then, the objective (8) is jointly $(\lambda/(2M))-$strongly convex, while the function $f_m$ is jointly convex, $((\Lambda L' + \lambda)/M)$-smooth w.r.t. $w$ and $(L' + \lambda)$-smooth w.r.t. $\beta_m$. Similarly, the function $f_{m,j}$ is jointly convex, $((\Lambda \mathcal{L}' + \lambda)/M)$-smooth w.r.t. $w$ and $(\mathcal{L}' + \lambda)$-smooth w.r.t. $\beta_m$.*

**Evaluating gradients.** Note that evaluating $\nabla_w f_m(x, \beta_m)$ under the objective (8) can be perfectly decoupled from evaluating $\nabla_\beta f_m(x, \beta_m)$. Therefore, we can make full use of our theory and take advantage of different complexities w.r.t. $\nabla_w$ and $\nabla_\beta$. Resulting communication and computation complexities for solving (8) are presented in Table 3.

## 2.4 Multi-Task Personalized FL and Implicit MAML

In its simplest form, the multi-task personalized objective (Smith et al., 2017; Wang et al., 2018) is given as

$$\min_{\beta_1, \ldots, \beta_M \in \mathbb{R}^d} F_{MX}(\beta) = \frac{1}{M} \sum_{m=1}^{M} f_m'(\beta_m) + \frac{\lambda}{2M} \sum_{m=1}^{M} \|\bar{\beta} - \beta_m\|^2, \tag{9}$$

where $\bar{\beta} := \frac{1}{M} \sum_{m=1}^{M} \beta_m$ and $\lambda \geq 0$ (Hanzely & Richtárik, 2020). On the other hand, the goal of implicit MAML (Rajeswaran et al., 2019; Dinh et al., 2020) is to minimize

$$\min_{w \in \mathbb{R}^d} F_{ME}(w) = \frac{1}{M} \sum_{i=1}^{M} \left( \min_{\beta_m \in \mathbb{R}^d} \left( f_m'(\beta_m) + \frac{\lambda}{2}\|w - \beta_m\|^2 \right) \right). \tag{10}$$

By reparametrizing (1), we can recover an objective that is simultaneously equivalent to both (9) and (10). In particular, by setting

$$f_m(w, \beta_m) = f_m'(\beta_m) + \lambda \|M^{-\frac{1}{2}}w - \beta_m\|^2,$$

the objective (1) becomes

$$\min_{w, \beta_1, \ldots, \beta_M \in \mathbb{R}^d} F_{MX2}(w, \beta) := \frac{1}{M} \sum_{m=1}^{M} f_m'(\beta_m) + \frac{\lambda}{2M} \sum_{m=1}^{M} \|M^{-\frac{1}{2}}w - \beta_m\|^2. \tag{11}$$

It is a simple exercise to notice the equivalence of (11) to both (9) and (10).[2] Indeed, we can always minimize (11) in $w$, arriving at $w^* = M^{\frac{1}{2}}\bar{\beta}$, and thus recovering the solution of (9). Similarly, by minimizing (11) in $\beta$ we arrive at (10).

Next, we establish the parameters in Assumptions 1 and 2.

---

[2]To the best of our knowledge, we are the first to notice the equivalence of (9) and (10).

**Lemma 2.** *Let $\mu' \leq \lambda/2$. Then the objective* (11) *is jointly $(\mu'/(3M))$-strongly convex, while $f_m$ is $(\lambda/M)$-smooth w.r.t. $w$ and $(L' + \lambda)$-smooth w.r.t. $\beta$. The function*

$$f_{m,i}(w, \beta_m) = f'_{m,i}(\beta_m) + (\lambda/2)\|M^{-\frac{1}{2}}w - \beta_m\|^2$$

*is jointly convex, $(\lambda/M)$-smooth w.r.t. $w$ and $(\mathcal{L}' + \lambda)$-smooth w.r.t. $\beta$.*

The proof is given in Appendix B.2. Hanzely et al. (2020a) showed that the minimax optimal communication complexity to solve (9) (and therefore to solve (10) and (11)) is $\Theta\left(\sqrt{\min(L', \lambda)/\mu'} \log \epsilon^{-1}\right)$. Furthermore, they showed that the minimax optimal number of gradients w.r.t. $f'$ is $\tilde{\Theta}\left(\left(\sqrt{L'/\mu'}\right)\log \epsilon^{-1}\right)$ and proposed a method that has the complexity $\Theta\left(\left(n + \sqrt{n(\mathcal{L}' + \lambda)/\mu'}\right)\log \epsilon^{-1}\right)$ w.r.t. the number of $f'_{m,j}$-gradients. We match the aforementioned communication guarantees when $\lambda = \mathcal{O}(L')$ and computation guarantees when $L' = \mathcal{O}(\lambda)$. Furthermore, when $\lambda = \mathcal{O}(L')$, our complexity guarantees are strictly better compared to the guarantees for solving the implicit MAML objective (10) directly (Rajeswaran et al., 2019; Dinh et al., 2020).

## 2.5 Adaptive Personalized FL (Deng et al., 2020)

The objective is given as

$$\min_{\beta_1, \ldots, \beta_M} F_{APFL}(\beta) = \frac{1}{M} \sum_{m=1}^{M} f'_m((1 - \alpha_m)\beta_m + \alpha_m(w'^*)), \tag{12}$$

where $(w')^* = \arg\min_{w \in \mathbb{R}^d} F'(w)$ is a solution to (4) and $0 < \alpha_1, \ldots \alpha_M < 1$. Assuming that $(w')^*$ is known, as was done in Deng et al. (2020), the problem (12) is an instance of (5); thus our approach achieves the optimal complexity.

A more interesting case (in terms of optimization) is when considering a relaxed variant of (12), given as

$$\min_{w, \beta} \frac{1}{M} \sum_{m=1}^{M} \left(\Lambda f'_m(w) + f'_m((1 - \alpha_m)\beta_m + \alpha_m w)\right) \tag{13}$$

where $\Lambda \geq 0$ is the relaxation parameter that allows recovering the original objective when $\Lambda \to \infty$. Such a choice, alongside with the usual rescaling of the parameter $w$ results in the following objective:

$$\min_{w, \beta_1, \ldots, \beta_M \in \mathbb{R}^d} F_{APFL2}(w, \beta) := \frac{1}{M} \sum_{i=1}^{M} f(w, \beta_m), \tag{14}$$

where

$$f(w, \beta_m) = \Lambda f'_m(M^{-\frac{1}{2}}w) + f'_m((1 - \alpha_m)\beta_m + \alpha_m M^{-\frac{1}{2}}w).$$

**Lemma 3.** *Let $\alpha_{\min} := \min_{1 \leq m \leq M} \alpha_m$ and $\alpha_{\max} := \max_{1 \leq m \leq M} \alpha_m$. If*

$$\Lambda \geq \max_{1 \leq m \leq M} (3\alpha_m^2 + (1 - \alpha_m)^2/2),$$

*then the function $F_{APFL2}$ is jointly $\left(\mu'(1 - \alpha_{\max})^2/M\right)$-strongly convex, $\left((\Lambda + \alpha_{\max}^2)L'/M\right)$-smooth w.r.t. $w$ and $\left((1 - \alpha_{\min})^2 L'/M\right)$-smooth w.r.t. $\beta$.*

The proof is given in Appendix B.3.

## 2.6 Personalized FL with Explicit Weight Sharing

The most typical example of the weight sharing setting is when parameters $w, \beta$ correspond to different layers of the same neural network. For example, $\beta_1, \ldots, \beta_M$ could be the weights of first few layers of a neural network, while $w$ are the weights of the remaining layers (Liang et al., 2020). Or, alternatively, each

of $\beta_1, \ldots, \beta_M$ can correspond to the weights of last few layers, while the remaining weights are included in the global parameter $w$ (Arivazhagan et al., 2019). Overall, we can write the objective as follows:

$$\min_{\substack{w \in \mathbb{R}^{d_w}, \\ \beta_1, \ldots, \beta_M \in \mathbb{R}^{d_\beta}}} F_{WS}(w, \beta) = \frac{1}{M} \sum_{m=1}^{M} f'_m([w, \beta_m]), \tag{15}$$

where $d_w + d_\beta = d$. Using an equivalent reparameterization of the $w-$space, we aim to minimize

$$\min_{\substack{w \in \mathbb{R}^{d_w}, \\ \beta_1, \ldots, \beta_M \in \mathbb{R}^{d_\beta}}} F_{WS2}(w, \beta) = \frac{1}{M} \sum_{m=1}^{M} f'_m([M^{-\frac{1}{2}} w, \beta_m]), \tag{16}$$

which is an instance of (1) with $f_m(w, \beta_m) = f'_m([M^{-\frac{1}{2}} w, \beta_m])$.

**Lemma 4.** *The function $F_{WS2}$ is jointly $\mu'$-strongly convex, $\left(\frac{L'}{M}\right)$-smooth w.r.t. $w$ and $L'$-smooth w.r.t. $\beta$. Similarly, the function $f_m$ is jointly convex, $\left(\frac{\mathcal{L}'}{M}\right)$-smooth w.r.t. $w$ and $\mathcal{L}'$-smooth w.r.t. $\beta$.*

The proof is straightforward and, therefore, omitted. A distinctive characteristic of the explicit weight sharing paradigm is that evaluating a gradient w.r.t. $w$-parameters automatically grants either free or highly cost-effective access to the gradient w.r.t. $\beta$-parameters (and vice versa).

## 2.7 Federated Residual Learning (Agarwal et al., 2020)

Agarwal et al. (2020) proposed federated residual learning:

$$\min_{\substack{w \in \mathbb{R}^{d_w}, \\ \beta_1, \ldots, \beta_M \in \mathbb{R}^{d_\beta}}} F_R(w, \beta) = \frac{1}{M} \sum_{m=1}^{M} l_m(A^w(w, x_m^w), A^\beta(\beta_m, x_m^\beta)), \tag{17}$$

where $(x_m^w, x_m^\beta)$ is a local feature vector (there may be an overlap between $x_m^w$ and $x_m^\beta$), $A^w(w, x_m^w)$ represents the model prediction using global parameters/features, $A^\beta(\beta, x_m^\beta)$ denotes the model prediction using local parameters/features, and $l(\cdot, \cdot)$ is a loss function. Clearly, we can recover (17) with

$$f_m(w, \beta_m) = l(A^w(M^{-\frac{1}{2}} w, x_m^w), A^\beta(\beta_m, x_m^\beta)). \tag{18}$$

Unlike the other objectives, here we cannot relate constants $\mu', L', \mathcal{L}'$ to $F_R$, since we do not write $f_m$ as a function of $f'm$. However, it seems natural to assume $L_R^w$ (or $L_R^\beta$)-smoothness of $l(A^w(w, x^w m), a_m^\beta)$ (or $l(a_m^w, A^\beta(\beta_m, x_m^\beta)))$ as a function of $w$ (or $\beta$) for any $a_m^\beta, x_m^\beta, a_m^w, x_m^w$. Let us define $\mathcal{L}_R^w, \mathcal{L}_R^\beta$ analogously, given that $l$ has an $n$-finite sum structure. Assuming, furthermore, that $F$ is $\mu$-strongly convex and $f_m$ is convex (for each $m \in M$), we can apply our theory.

## 2.8 MAML Based Approaches

Meta-learning has recently been employed for personalization (Chen et al., 2018; Khodak et al., 2019; Jiang et al., 2019; Fallah et al., 2020; Lin et al., 2020). Notably, the model-agnostic meta-learning (MAML) (Finn et al., 2017) based personalized FL objective is given as

$$\min_{w \in \mathbb{R}^d} F_{MAML}(w) = \frac{1}{M} \sum_{m=1}^{M} f'_m(w - \alpha \nabla f'_m(w)). \tag{19}$$

Although we can recover (19) as a special case of (1) by setting $f_m(w, \beta_m) = f'_m(w - \alpha \nabla f'_m(w))$, our (convex) convergence theory does not apply due to the inherent non-convex structure of (19). Specifically, objective $F_{MAML}$ is non-convex even if function $f'_m$ is convex. In this scenario, only our non-convex rates of Local SGD apply.

---

**Algorithm 1** LSGD-PFL

---

**input** Stepsizes $\{\eta_k\}_{k \geq 0} \in \mathbb{R}$, starting point $w^0 \in \mathbb{R}^{d_0}$, $\beta_m^0 \in \mathbb{R}^{d_m}$ for all $m \in [M]$, communication period $\tau$.

   **for** $k = 0, 1, 2, \ldots$ **do**

      **if** $k \bmod \tau = 0$ **then**

         Send all $w_m^k$'s to server, let $w^k = \frac{1}{M} \sum_{m=1}^{M} w_m^k$

         Send $w^k$ to each device, set $w_m^k = w^k, \forall m \in [M]$

      **end if**

      **for** $m = 1, 2, \ldots, M$ in parallel **do**

         Sample $\xi_{1,m}^k, \ldots \xi_{B,m}^k \sim \mathcal{D}_m$ independently

         Compute $g_m^k = \frac{1}{B} \sum_{j=1}^{B} \nabla \hat{f}_m(w_m^k, \beta_m^k; \xi_{j,m}^k)$

         Update the iterates $(w_m^{k+1}, \beta_m^{k+1}) = (w_m^k, \beta_m^k) - \eta_k \cdot g_m^k$

      **end for**

   **end for**

---

## 3 Local SGD

The most popular optimizer to train non-personalized FL models is the local SGD/FedAvg (McMahan et al., 2016; Stich, 2019). We devise a local SGD variant tailored to solve the personalized FL objective (1) – LSGD-PFL. See the detailed description in Algorithm 1. Specifically, LSGD-PFL can be seen as a combination of local SGD applied on global parameters $w$ and SGD applied on local parameters $\beta$. To mimic the non-personalized setup of local SGD, we assume access to the local objective $f_m(w, \beta_m)$ in the form of an unbiased stochastic gradient with bounded variance.

Admittedly, LSGD-PFL was already proposed by Arivazhagan et al. (2019) and Liang et al. (2020) to solve a particular instance of (1). However, no optimization guarantees were provided. In contrast, we provide convergence guarantees for LSGD-PFL that recover the convergence rate of LSGD when $d_1 = d_2 = \cdots = d_M = 0$ and the rate of SGD when $d_0 = 0$. Next, we demonstrate that LSGD-PFL works best when applied to an objective with rescaled $w$-space, unlike what was proposed in the aforementioned papers.

We will need the following assumption on the stochastic gradients.

**Assumption 3.** *Assume that the stochastic gradients $\nabla_w \hat{f}_m(w, \beta_m, \zeta)$ and $\nabla_\beta \hat{f}_m(w, \beta_m, \zeta)$ satisfy the following conditions for all $m \in [M]$, $w \in \mathbb{R}^{d_0}$, and $\beta_m \in \mathbb{R}^{d_m}$:*

$$\mathbb{E}\nabla_w \hat{f}_m(w, \beta_m, \zeta) = \nabla_w f_m(w, \beta_m),$$

$$\mathbb{E}\nabla_\beta \hat{f}_m(w, \beta_m, \zeta) = \nabla_\beta f_m(w, \beta_m),$$

$$\mathbb{E}\|\nabla_w \hat{f}_m(w, \beta_m, \zeta) - \nabla_w f_m(w, \beta_m)\|^2 \leq \sigma^2, \text{ and}$$

$$\mathbb{E}\sum_{m=1}^{M} \|\nabla_\beta \hat{f}_m(w, \beta_m, \zeta) - \nabla_\beta f_m(w, \beta_m)\|^2 \leq M\sigma^2.$$

Let

$$(\overline{w}^K, \overline{\beta}^K) := \left( \sum_{k=0}^{K} (1 - \eta\mu)^{-k-1} \right)^{-1} \sum_{k=0}^{K} (1 - \eta\mu)^{-k-1} (w^k, \beta^k).$$

We are now ready to state the convergence rate of LSGD-PFL.

**Theorem 1.** *Suppose that Assumptions 1 and 3 hold. Let $\eta_k = \eta$ for all $k \geq 0$, where $\eta$ satisfies*

$$0 < \eta \leq \min\left\{ \frac{1}{4L^\beta}, \frac{1}{8\sqrt{3e}(\tau - 1)L^w} \right\}.$$

Let $\zeta_*^2 := \frac{1}{M} \sum_{m=1}^{M} \|\nabla f_m(w^*, \beta^*)\|^2$ be the data heterogeneity parameter at the optimum. The iteration complexity of Algorithm 1 to achieve $\mathbb{E} f(\overline{w}^K, \overline{\beta}^K) - f(w^*, \beta^*) \leq \epsilon$ is

$$\tilde{\mathcal{O}} \left( \frac{\max\left(L^\beta, \tau L^w\right)}{\mu} + \frac{\sigma^2}{MB\mu\epsilon} + \frac{\tau}{\mu} \sqrt{\frac{L^w(\zeta_*^2 + \sigma^2 B^{-1})}{\epsilon}} \right).$$

The iteration complexity of LSGD-PFL can be seen as a sum of two complexities: the complexity of minibatch SGD to minimize a problem with a condition number $L^\beta/\mu$ and the complexity of local SGD to minimize a problem with a condition number $L^w/\mu$. Note that the key reason why we were able to obtain such a rate for LSGD-PFL is the rescaling of the $w$-space by a constant $M^{-\frac{1}{2}}$. Arivazhagan et al. (2019) and Liang et al. (2020), where LSGD-PFL was introduced without optimization guarantees, did not consider such a reparametrization.

We also have the following result for weakly convex objectives.

**Theorem 2.** *Suppose that the conditions of Theorem 1 hold. Let $\mu = 0$. The iteration complexity of Algorithm 1 to achieve $\mathbb{E} f(\overline{w}^K, \overline{\beta}^K) - f(w^*, \beta^*) \leq \epsilon$ is*

$$\tilde{O} \left( \frac{\max\left(L^\beta, \tau L^w\right)}{\epsilon} + \frac{\sigma^2}{MB\epsilon^2} + \frac{\tau \sqrt{L^w(\zeta_*^2 + \sigma^2 B^{-1})}}{\epsilon^{\frac{3}{2}}} \right).$$

The proofs of Theorem 1 and Theorem 2 can be found in Section 3.2. The reparametrization of $w$ plays a key role in proving the iteration complexity bound. Unlike the convergence guarantees for ACD-PFL and ASVRCD-PFL, which are introduced in Section 4, we do not claim any optimality properties for the rates obtained in Theorem 1 and Theorem 2. However, given the popularity of Local SGD as an optimizer for non-personalized FL models, Algorithm 1 is a natural extension of Local SGD to personalized FL models, and the corresponding convergence rate is an important contribution.

### 3.1 Nonconvex Theory for LSGD-PFL

To demonstrate that LSGD-PFL works in the nonconvex setting as well, we develop a non-convex theory for it. Therefore, the algorithm is also applicable, for example, for solving the explicit MAML objective. Note that we do not claim the optimality of our results.

Before stating the results, we need to make the following assumptions, which are slightly different from the rest of the paper. First, we need a smoothness assumption on local objective functions.

**Assumption 4.** *The local objective function $f_m(\cdot)$ is differentiable and $L$-smooth, that is, $\|\nabla f_m(u) - \nabla f_m(v)\| \leq L\|u - v\|$ for all $u, v \in \mathbb{R}^{d_0 + d_m}$ and $m \in [M]$.*

This condition is slightly different compared to the smoothness assumption on the objective stated in Assumption 1. Next, we need local stochastic gradients to have bounded variance.

**Assumption 5.** *The stochastic gradients $\nabla_w \hat{f}_m(w, \beta_m, \zeta)$, $\nabla_\beta \hat{f}_m(w, \beta_m, \zeta)$ satisfy for all $m \in [M]$, $w \in \mathbb{R}^{d_0}$, $\beta_m \in \mathbb{R}^{d_m}$:*

$$\mathbb{E}_\zeta \left[ \|\nabla_w \hat{f}_m(w, \beta_m, \zeta) - \nabla_w f_m(w, \beta_m)\|^2 \right] \leq C_1 \|\nabla_w f_m(w, \beta_m)\|^2 + \frac{\sigma_1^2}{B},$$

$$\mathbb{E}_\zeta \left[ \|\nabla_\beta \hat{f}_m(w, \beta_m, \zeta) - \nabla_\beta f_m(w, \beta_m)\|^2 \right] \leq C_2 \|\nabla_\beta f_m(w, \beta_m)\|^2 + \frac{\sigma_2^2}{B},$$

*for all $m \in [M]$, where $C_1, C_2, \sigma_1^2, \sigma_2^2$ are all positive constants.*

This assumption is common in the literature. See, for example, Assumption 3 in Haddadpour & Mahdavi (2019). Note that this assumption is weaker than Assumption 3. We also need an assumption on data heterogeneity.

**Assumption 6** (Bounded Dissimilarity)**.** *There is a positive constant $\lambda > 0$ such that for all $w \in \mathbb{R}^{d_0}$ and $\beta_m \in \mathbb{R}^{d_m}$, $m \in [M]$, we have*

$$\frac{1}{M} \sum_{m=1}^{M} \|\nabla f_m(w, \beta_m)\|^2 \leq \lambda \left\| \frac{1}{M} \sum_{m=1}^{M} \nabla f_m(w, \beta_m) \right\|^2 + \sigma_{dif}^2.$$

This way of characterizing data heterogeneity was used in Haddadpour & Mahdavi (2019) – see Definition 1 therein. Given the above assumptions, we can establish the following convergence rate of LSGD-PFL for general non-convex objectives.

**Theorem 3.** *Suppose that Assumptions 4-6 hold. Let $\eta_k = \eta$, for all $k \geq 0$, where $\eta$ is small enough to satisfy*

$$-1 + \eta L \lambda \left( \frac{C_1}{M} + C_2 + 1 \right) + \lambda \eta^2 L^2 (\tau - 1) \tau (C_1 + 1) \leq 0. \tag{20}$$

*We have*

$$\frac{1}{K} \sum_{k=0}^{K-1} \mathbb{E} \left[ \left\| \frac{1}{M} \sum_{m=1}^{M} \nabla f_m(w^k, \beta_m^k) \right\|^2 \right]$$

$$\leq \frac{2\mathbb{E}\left[ \frac{1}{M} \sum_{m=1}^{M} f_m(w^0, \beta_m^0) - f^* \right]}{\eta K} + \eta L \lambda \left\{ \left( \frac{C_1}{M} + C_2 + 1 \right) \sigma_{dif}^2 + \frac{\sigma_1^2}{MB} + \frac{\sigma_2^2}{B} \right\}$$

$$+ \eta^2 L^2 \sigma_{dif}^2 (\tau - 1)^2 (C_1 + 1) + \frac{\eta^2 L^2 \sigma_1^2 (\tau - 1)^2}{B},$$

*where $w^k := \frac{1}{M} \sum_{m=1}^{M} w_m^k$ is a sequence of so-called virtual iterates.*

The following assumption is commonly used to characterize non-convex objectives in the literature.

**Assumption 7** ($\mu$-Polyak-Łojasiewicz (PL))**.** *There exists a positive constant $\mu > 0$, such that for all $w \in \mathbb{R}^{d_0}$ and $\beta_m \in \mathbb{R}^{d_m}$, $m \in [M]$, we have*

$$\frac{1}{2} \left\| \frac{1}{M} \sum_{m=1}^{M} \nabla f_m(w, \beta_m) \right\|^2 \geq \mu \left( \frac{1}{M} \sum_{m=1}^{M} f_m(w, \beta_m) - f^* \right).$$

When the local objective functions satisfy the PL-condition, we obtain a faster convergence rate stated in the theorem below.

**Theorem 4.** *Suppose that Assumptions 4-7 hold. Suppose $\eta_k = 1/(\mu(k + \beta\tau + 1))$, where $\beta$ is a positive constant satisfying*

$$\beta > \max \left\{ \frac{2\lambda L}{\mu} \left( \frac{C_1}{M} + C_2 + 1 \right) - 2, \frac{2L^2 \lambda (C_1 + 1)}{\mu^2}, 1 \right\}$$

*and $\tau$ is large enough such that*

$$\tau \geq \sqrt{\frac{\max \left\{ (2L^2 \lambda (C_1 + 1)/\mu^2) e^{1/\beta} - 4, 0 \right\}}{\beta^2 - (2L^2 \lambda (C_1 + 1)/\mu^2) e^{\frac{1}{\beta}}}}.$$

*Then*

$$\mathbb{E} \left[ \frac{1}{M} \sum_{m=1}^{M} f_m \left( w^K, \beta_m^k \right) - f^* \right]$$

$$\leq \frac{b^3}{(K + \beta\tau)^3} \mathbb{E} \left[ \frac{1}{M} \sum_{m=1}^{M} f_m \left( w^0, \beta_m^0 \right) - f^* \right] + \frac{2L^2 (\tau - 1)^2 K}{\mu^3 (K + \beta\tau)^3} \left\{ \sigma_{dif}^2 (C_1 + 1) + \frac{\sigma_1^2}{B} \right\}$$

$$+ \frac{LK(K + 2\beta\tau + 2)}{4\mu^2 (K + \beta\tau)^3} \left\{ \sigma_{dif}^2 \left( \frac{C_1}{M} + C_2 + 1 \right) + \frac{\sigma_1^2}{MB} + \frac{\sigma_2^2}{B} \right\}.$$

The proofs of Theorem 3 and Theorem 4 can be found in Appendix B.7.

### 3.2 Proof of Theorem 1 and Theorem 2

The main idea consists of invoking the framework for analyzing local SGD methods introduced in Gorbunov et al. (2021) with several minor modifications. In particular, Algorithm 1 is an intriguing method that runs a local SGD on $w$-parameters and SGD on $\beta$-parameters. Therefore, we shall treat these parameter sets differently. Define $V_k := \frac{1}{M} \sum_{m=1}^{M} \|w^k - w_m^k\|^2$ where $w^k := \frac{1}{M} \sum_{m=1}^{M} w_m^k$ is defined as in Theorem 3.

The first step towards the convergence rate is to figure out the parameters of Assumption 2.3 from Gorbunov et al. (2021). The following lemma is an analog of Lemma G.1 in Gorbunov et al. (2021).

**Lemma 5.** *Let Assumptions 1 and 3 hold. Let $L = \max\{L^w, L^\beta\}$. Then, we have:*

$$\frac{1}{M} \sum_{m=1}^{M} \|\nabla_w f_m(w_m^k, \beta_m^k)\|^2 \leq 6L^w \left(f(w^k, \beta_m^k) - f(w^*, \beta^*)\right) + 3(L^w)^2 V_k + 3\zeta_*^2, \tag{21}$$

*and*

$$\left\|\frac{1}{M} \sum_{m=1}^{M} \nabla_w f_m(w_m^k, \beta_m^k)\right\|^2 + \frac{1}{M^2} \sum_{m=1}^{M} \left\|\nabla_\beta f_m(w_m^k, \beta_m^k)\right\|^2 \leq 4L \left(f(w^k, \beta_m^k) - f(w^*, \beta^*)\right) + 2(L^w)^2 V_k. \tag{22}$$

The proof is given in Appendix B.4. Using Lemma 5 we recover a set of crucial parameters of Assumption 2.3 from Gorbunov et al. (2021).

**Lemma 6.** *Let $g_{w,m}^k := (g_m^k)_{1:d_0}$ and $g_{\beta,m}^k := (g_m^k)_{(d_0+1):(d_0+d_m)}$. Then*

$$\frac{1}{M} \sum_{m=1}^{M} \mathbb{E}\|g_{w,m}^k\|^2 \leq 6L^w \left(f(w^k, \beta_m^k) - f(w^*, \beta^*)\right) + 3(L^w)^2 V_k + \frac{\sigma^2}{B} + 3\zeta_*^2, \tag{23}$$

*and*

$$\mathbb{E}\left\|\frac{1}{M} \sum_{m=1}^{M} g_{w,m}^k\right\|^2 + \frac{1}{M^2} \sum_{m=1}^{M} \left\|g_{\beta,m}^k\right\|^2 \leq 4L \left(f(w^k, \beta_m^k) - f(w^*, \beta^*)\right) + 2(L^w)^2 V_k + \frac{2\sigma^2}{BM}. \tag{24}$$

The proof is given in Appendix B.5. Finally, the following lemma is an analog of Lemma E.1 in Gorbunov et al. (2021) and gives us the remaining parameters of Assumption 2.3 therein.

**Lemma 7.** *Suppose that Assumptions 1 and 3 hold and*

$$\eta \leq \frac{1}{8\sqrt{3e}(\tau - 1)L^w}.$$

*Then*

$$2L^w \sum_{k=0}^{K} (1-\eta\mu)^{-k-1} \mathbb{E}V_k \leq \frac{1}{2} \sum_{k=0}^{K} (1-\eta\mu)^{-k-1} \mathbb{E}F(w^k, \beta^k) - F(w^*, \beta^*) + 2L^w D\eta^2 \sum_{k=0}^{K} (1-\eta\mu)^{-k-1}, \tag{25}$$

*where*

$$D = 2e(\tau - 1)\tau \left(3\zeta_*^2 + \frac{\sigma^2}{B}\right).$$

The proof is given in Appendix B.6. With these preliminary results, we are ready to state the main convergence result for Algorithm 1.

**Theorem 5.** *Suppose that Assumptions 1 and 3 hold and the stepsize $\eta$ satisfies*

$$0 < \eta \leq \min\left\{\frac{1}{4L^\beta}, \frac{1}{8\sqrt{3e}(\tau - 1)L^w}\right\}.$$

*Define*

$$(\overline{w}^K, \overline{\beta}^K) := \left(\sum_{k=0}^{K}(1-\eta\mu)^{-k-1}\right)^{-1} \sum_{k=0}^{K}(1-\eta\mu)^{-k-1}(w^k, \beta^k),$$

$$\Phi^0 := \frac{2\|w^0 - w^*\|^2 + \sum_{m=1}^{M}\|\beta_m^0 - \beta_m^*\|^2}{\eta}, \ and$$

$$\Psi^0 := \frac{2\sigma^2}{BM} + 8L^w\eta e(\tau-1)\tau\left(3\zeta_*^2 + \frac{\sigma^2}{B}\right).$$

*Then, if $\mu > 0$, we have*

$$\mathbb{E}f(\overline{w}^K, \overline{\beta}^K) - f(w^*, \beta^*) \leq (1-\eta\mu)^K \Phi^0 + \eta\Psi^0, \tag{26}$$

*while, in the case when $\mu = 0$, we have*

$$\mathbb{E}f(\overline{w}^K, \overline{\beta}^K) - f(w^*, \beta^*) \leq \frac{\Phi^0}{K} + \eta\Psi^0. \tag{27}$$

The proof follows directly from Theorem 2.1 of Gorbunov et al. (2021), once the conditions are verified, as has been done in Lemma 5, Lemma 6, and Lemma 7. Theorem 1 and Theorem 2 then follow from Corollary D.1 and Corollary D.2 of Gorbunov et al. (2021).

## 4 Minimax Optimal Methods

We discuss the complexity of solving (1) in terms of the number of communication rounds required to reach an $\epsilon$-solution and the amount of local computation, both in terms of the number of (stochastic) gradients with respect to global $w$-parameters and local $\beta$-parameters.

### 4.1 Lower Complexity Bounds

We provide lower complexity bounds for solving (1) when $f_m$ is a finite sum (3). We show that any algorithm with access to the communication oracle and local (stochastic) gradient oracle with respect to the $w$ or $\beta$ parameters requires at least a certain number of oracle calls to approximately solve (1).

**Oracle.** The considered oracle allows us at any iteration to compute either:

- $\nabla_w f_{m,i}(w_m, \beta_m)$ on each device for a randomly selected $i \in [n]$ and any $w_m, \beta_m$; or
- $\nabla_\beta f_{m,i}(w_m, \beta_m)$ on each device for a randomly selected $i \in [n]$ and any $w_m, \beta_m$; or
- the average of $w_m$'s alongside broadcasting the average back to clients (communication step).

Our lower bound is provided for iterative algorithms whose iterates lie in the span of historical oracle queries only. Let us denote such a class of algorithms as $\mathcal{A}$. In particular, for each $m, k$ we must have

$$\beta_m^k \in \text{Lin}\left(\beta_m^0, \nabla_\beta f_m(w_m^0, \beta_m^0), \dots, \nabla_\beta f_m(w_m^{k-1}, \beta_m^{k-1})\right)$$

and

$$w_m^k \in \text{Lin}\left(w^0, \nabla_w f_m(w_m^0, \beta_m^0), \dots, \nabla_w f_m(w_m^{k-1}, \beta_m^{k-1}), Q_l(k)\right),$$

where

$$Q^k = \bigcup_{m=1}^{M}\left\{w^0, \nabla_w f_m(w_m^0, \beta_m^0), \dots, \nabla_w f_m(w_m^{l(k)}, \beta_m^{l(k)})\right\},$$

with $l(k)$ being the index of the last communication round until iteration $k$. While such a restriction is widespread in the classical optimization literature (Nesterov, 2018; Scaman et al., 2018; Hendrikx et al., 2021; Hanzely et al., 2020a), it can be avoided by more complex arguments (Nemirovskij & Yudin, 1983; Woodworth & Srebro, 2016; Woodworth et al., 2018).

We then have the following theorem regarding the minimal calls of oracles for solving (1).

---

**Algorithm 2** ACD-PFL

---

**input** $0 < \theta < 1$, $\eta, \nu > 0$, $w_y^0 = w_z^0 \in \mathbb{R}^{d_0}$, $\beta_{y,m}^0 = \beta_{z,m}^0 \in \mathbb{R}^{d_m}$ for $1 \le m \le M$.

    **for** $k = 0, 1, 2, \ldots$ **do**

        $w_x^{k+1} = (1 - \theta)w_y^k + \theta w_z^k$

        **for** $m = 1, \ldots, M$ in parallel **do**

            $\beta_{x,m}^{k+1} = (1 - \theta)\beta_{y,m}^k + \theta\beta_{z,m}^k$

        **end for**

$$\xi = \begin{cases} 1, & \text{with probability } p_w = \frac{\sqrt{L^w}}{\sqrt{L^w} + \sqrt{L^\beta}} \\ 0, & \text{with probability } p_\beta = \frac{\sqrt{L^\beta}}{\sqrt{L^w} + \sqrt{L^\beta}} \end{cases}$$

        **if** $\xi = 0$ **then**

            $w_y^{k+1} = w_x^{k+1} - \frac{1}{L^w}\frac{1}{M}\sum_{m=1}^M \nabla_w f_m(w_x^{k+1}, \beta_{x,m}^{k+1})$

            $w_z^{k+1} = \frac{1}{1+\eta\nu}\left(w_z^k + \eta\nu w_x^{k+1} - \frac{\eta}{\sqrt{L^w}(\sqrt{L^w} + \sqrt{L^\beta})}\frac{1}{M}\sum_{m=1}^M \nabla_w f_m(w_x^{k+1}, \beta_{x,m}^{k+1})\right)$

            **for** $m = 1, \ldots, M$ in parallel **do**

                $\beta_{z,m}^{k+1} = \frac{1}{1+\eta\nu}\left(\beta_{z,m}^k + \eta\nu\beta_{x,m}^{k+1}\right)$

            **end for**

        **else**

            **for** $m = 1, \ldots, M$ in parallel **do**

                $\beta_{y,m}^{k+1} = \beta_{x,m}^{k+1} - \frac{1}{L^\beta}\nabla_\beta f_m(w_x^{k+1}, \beta_{x,m}^{k+1})$

            **end for**

            $\beta_{z,m}^{k+1} = \frac{1}{1+\eta\nu}\left(\beta_{z,m}^k + \eta\nu\beta_{x,m}^{k+1} - \frac{\eta}{\sqrt{L^\beta}(\sqrt{L^w} + \sqrt{L^\beta})}\nabla_\beta f_m(w_x^{k+1}, \beta_{x,m}^{k+1})\right)$

            $w_z^{k+1} = \frac{1}{1+\eta\nu}\left(w_z^k + \eta\nu w_x^{k+1}\right)$

        **end if**

    **end for**

---

**Theorem 6.** *Let $F$ satisfy Assumptions 1 and 2. Then, any algorithm from the class $\mathcal{A}$ requires at least $\Omega(\sqrt{L^w/\mu}\log\epsilon^{-1})$ communication rounds, $\Omega\left(n + \sqrt{n\mathcal{L}^w/\mu}\log\epsilon^{-1}\right)$ calls to $\nabla_w$-oracle and $\Omega\left(n + \sqrt{n\mathcal{L}^\beta/\mu}\log\epsilon^{-1}\right)$ calls to $\nabla_\beta$-oracle to reach the $\epsilon$-solution.*

The proof is given in Appendix B.8. In the special case where $n = 1$, Theorem 6 provides a lower complexity bound for solving (1) with access to the full gradient locally. Specifically, it shows both the communication complexity and local gradient complexity with respect to $w$-variables of the order $\Omega\left(\sqrt{\frac{L^w}{\mu}}\log\frac{1}{\epsilon}\right)$, and the local gradient complexity with respect to $\beta$-variables of the order $\Omega\left(\sqrt{\frac{L^\beta}{\mu}}\log\frac{1}{\epsilon}\right)$.

### 4.2 Accelerated Coordinate Descent for PFL

We apply Accelerated Block Coordinate Descent (ACD) (Allen-Zhu et al., 2016; Nesterov & Stich, 2017; Hanzely & Richtárik, 2019) to solve (1). We separate the domain into two blocks of coordinates to sample from: the first one corresponding to $w$ parameters and the second one corresponding to $\beta = [\beta_1, \beta_2, \ldots, \beta_M]$. Specifically, at every iteration, we toss an unfair coin. With probability $p_w = \sqrt{L^w}/(\sqrt{L^w} + \sqrt{L^\beta})$, we compute $\nabla_w F(w, \beta)$ and update block $w$. Alternatively, with probability $p_\beta = 1 - p_w$, we compute $\nabla_\beta F(w, \beta)$ and update block $\beta$. Plugging the described sampling of coordinate blocks into ACD, we arrive at Algorithm 2. Note that ACD from Allen-Zhu et al. (2016) only allows for subsampling individual coordinates and does not allow for "blocks." A variant of ACD that provides the right convergence guarantees for block sampling was proposed in Nesterov & Stich (2017) and Hanzely & Richtárik (2019).

We provide an optimization guarantee for Algorithm 2 in the following theorem.

**Theorem 7.** *Suppose that Assumption 1 holds. Let*

$$\nu = \frac{\mu}{(\sqrt{L^w} + \sqrt{L^\beta})^2}, \; \theta = \frac{\sqrt{\nu^2 + 4\nu} - \nu}{2}, \; and \; \eta = \theta^{-1}.$$

*The iteration complexity of ACD-PFL is*

$$\mathcal{O}\left(\sqrt{(L^w + L^\beta)/\mu}\log \epsilon^{-1}\right).$$

The proof follows directly from Theorem 4.2 of Hanzely & Richtárik (2019). Since $\nabla_w F(w, \beta)$ is evaluated on average once every $1/p_w$ iterations, ACD-PFL requires $\mathcal{O}\left(\sqrt{L^w/\mu}\log \epsilon^{-1}\right)$ communication rounds and $\mathcal{O}\left(\sqrt{L^w/\mu}\log \epsilon^{-1}\right)$ gradient evaluations with respect to $w$, thus matching the lower bound. Similarly, as $\nabla_\beta F(w, \beta)$ is evaluated on average once every $1/p_\beta$ iterations, we require $\mathcal{O}\left(\sqrt{L^\beta/\mu}\log \epsilon^{-1}\right)$ evaluations of $\nabla_\beta F(w, \beta)$ to reach an $\epsilon$-solution; again matching the lower bound. Consequently, ACD-PFL is minimax optimal in terms of all three quantities of interest simultaneously.

We are not the first to propose a variant of coordinate descent (Nesterov, 2012) for personalized FL. Wu et al. (2021) introduced block coordinate descent to solve a variant of (11) formulated over a network. However, they do not argue about any form of optimality for their approach, which is also less general as it only covers a single personalized FL objective.

## 4.3 Accelerated SVRCD for PFL

Despite being minimax optimal, the main drawback of ACD-PFL is the necessity of having access to the full gradient of local loss $f_m$ with respect to either $w$ or $\beta$ at each iteration. Specifically, computing the full gradient with respect to $f_m$ might be very expensive when $f_m$ is a finite sum (3). Ideally, one would desire an algorithm that is i) subsampling the global/local variables $w$ and $\beta$ just as ACD-PFL, ii) subsampling the local finite sum, iii) employing control variates to reduce the variance of the local stochastic gradient (Johnson & Zhang, 2013; Defazio et al., 2014), and iv) accelerated in the sense of Nesterov (1983).

We propose a method – ASVRCD-PFL – that satisfies all four conditions above, by carefully designing an instance of ASVRCD (Accelerated proximal Stochastic Variance Reduced Coordinate Descent) (Hanzely et al., 2020b) applied to minimize an objective in a lifted space that is equivalent to (1). We are not aware of any other algorithm capable of satisfying i)-iv) simultaneously.

The construction of ASVRCD-PFL involves four main ingredients. First, we rewrite the original problem in a lifted space, which corresponds to the problem form discussed in Hanzely et al. (2020b). Second, we construct an unbiased stochastic gradient estimator by sampling coordinate blocks. Next, we enrich the stochastic gradient by control variates as in SVRG. Finally, we incorporate Nesterov's momentum. We explain the construction of ASVRCD-PFL in detail below.

**Lifting the problem space.** ASVRCD-PFL is an instance of ASVRCD applied to an objective (1) in a lifted space. We have that

$$\min_{w\in\mathbb{R}_0^d,\beta_m\in\mathbb{R}^{d_m},\forall m\in[M]} F(w,\beta) = \min_{\substack{X[1,:,:]\in\mathbb{R}^{M\times n\times d_0}\\X[2,m,:]\in\mathbb{R}^{n\times d_m},\forall m\in M}}\left\{\mathbf{P}(X) \coloneqq \mathbf{F}(X) + \psi(X)\right\},$$

where

$$\mathbf{F}(X) \coloneqq \frac{1}{M}\sum_{m=1}^{M}\left(\frac{1}{n}\sum_{j=1}^{n}f_{m,j}(X[1,m,j],X[2,m,j])\right)$$

and

$$\psi(X) \coloneqq \begin{cases} 0 & \text{if } m, m' \in [M], j, j' \in [n] : X[1,m,j] = X[1,m',j'],\ X[2,m,j] = X[2,m,j']\\ \infty & \text{otherwise.}\end{cases}$$

Variables $X[1, m, j]$ correspond to $w$ for all $m \in [M]$ and $j \in [n]$, while variables $X[2, m, j]$ correspond to $\beta_m$ for all $j \in [n]$. The equivalence between the objective (1) and the objective in the lifted space is ensured with

the indicator function $\psi(X)$, which forces different $X$ variables to take the same values. We apply ASVRCD with a carefully chosen non-uniform sampling of coordinate blocks to minimize $\mathbf{P}(X)$.

**Sampling of coordinate blocks.** The key component of ASVRCD-PFL is the construction of the unbiased stochastic gradient estimator of $\nabla \mathbf{F}(X)$, which we describe here. We consider two independent sources of randomness when sampling the coordinate blocks.

First, we toss an unfair coin $\zeta$. With probability $p_w$, we have $\zeta = 1$. In such a case, we ignore the local variables and update the global variables only, corresponding to $w$ or $X[1]$ in our current notation. Alternatively, $\zeta = 2$ with probability $p_\beta := 1 - p_w$. In such a case, we ignore the global variables and update local variables only, corresponding to $\beta$ or $X[2]$ in our current notation.

Second, we consider local subsampling. At each iteration, the stochastic gradient is constructed using $\nabla F_j$ only, where $F_j(w, \beta) := \frac{1}{M} \sum_{m=1}^{M} f_{m,j}(w, \beta_m)$ and $j$ is selected uniformly at random from $[n]$. For the sake of simplicity, we assume that all clients sample the same index, i.e., the randomness is synchronized. A similar rate can be obtained without shared randomness.

With these sources of randomness, we arrive at the following construction of $\mathbf{G}(X)$, which is an unbiased stochastic estimator of $\nabla \mathbf{F}(X)$:

$$\mathbf{G}(X)[1, m, j'] = \begin{cases} \frac{1}{p^w} \nabla_w f_{j',m}(X[1,m,j'], X[2,m,j']) & \text{if } \zeta = 1 \text{ and } j' = j; \\ 0 \in \mathbb{R}^{d_0} & \text{otherwise;} \end{cases}$$

$$\mathbf{G}(X)[2, m, j'] = \begin{cases} \frac{1}{p^\beta} \nabla_\beta f_{j',m}(X[1,m,j'], X[2,m,j']) & \text{if } \zeta = 2 \text{ and } j' = j; \\ 0 \in \mathbb{R}^{d_m} & \text{otherwise.} \end{cases}$$

**Control variates and acceleration.** We enrich the stochastic gradient by incorporating control variates, resulting in an SVRG-style stochastic gradient estimator. In particular, the resulting stochastic gradient will take the form of $\mathbf{G}(X) - \mathbf{G}(Y) + \nabla \mathbf{F}(Y)$, where $Y$ is another point that is updated upon a successful toss of a $\rho$-coin. The last ingredient of the method is to incorporate Nesterov's momentum.

Combining the above ingredients, we arrive at the ASVRCD-PFL procedure, which is detailed in Algorithm 3 in the lifted notation. Algorithm 4 details ASVRCD-PFL in the notation consistent with the rest of the paper.

The following theorem provides convergence guarantees for ASVRCD-PFL.

**Theorem 8.** *Suppose Assumptions 1 and 2 hold. The iteration complexity of ASVRCD-PFL with*

$$\eta = \frac{1}{4\mathcal{L}}, \quad \theta_2 = \frac{1}{2}, \quad \gamma = \frac{1}{\max\{2\mu, 4\theta_1/\eta\}},$$

$$\nu = 1 - \gamma\mu, \quad \theta_1 = \min\left\{\frac{1}{2}, \sqrt{\eta\mu \max\left\{\frac{1}{2}, \frac{\theta_2}{\rho}\right\}}\right\}, \quad \text{and} \quad p_w = \frac{\mathcal{L}^w}{\mathcal{L}^\beta + \mathcal{L}^w}$$

*is*

$$\mathcal{O}\left(\left(\rho^{-1} + \sqrt{(\mathcal{L}^w + \mathcal{L}^\beta)/(\rho\mu)}\right) \log \epsilon^{-1}\right),$$

*where $\rho$ is the frequency of updating the control variates.*

The communication complexity and the local stochastic gradient complexity with respect to $w$-parameters of order $\mathcal{O}\left(\left(n + \sqrt{n\mathcal{L}^w/\mu}\right) \log \epsilon^{-1}\right)$, is obtained by setting $\rho = \mathcal{L}^w/\left((\mathcal{L}^w + \mathcal{L}^\beta)n\right)$. Analogously, setting $\rho = \mathcal{L}^\beta/((\mathcal{L}^w + \mathcal{L}^\beta)n)$ yields the local stochastic gradient complexity with respect to $\beta$-parameters of order $\mathcal{O}\left(\left(n + \sqrt{n\mathcal{L}^\beta/\mu}\right) \log \epsilon^{-1}\right)$. In contrast with Theorem 6, this result shows that ASVRCD-PFL can be optimal in terms of the local computation either with respect to $\beta$-variables or in terms of the $w$-variables. Unfortunately, these bounds are not achieved simultaneously unless $\mathcal{L}^w, \mathcal{L}^\beta$ are of a similar order, which we leave for future research. The proof is given in Appendix B.9. Additional discussion on how to choose the tuning parameters is given in Theorem 9.

---

**Algorithm 3** ASVRCD-PFL (lifted notation)

---

**input** $0 < \theta_1, \theta_2 < 1$, $\eta, \nu, \gamma > 0$, $\rho \in (0,1)$, $Y^0 = Z^0 = X^0$.

$\quad$ **for** $k = 0, 1, 2, \ldots$ **do**

$\qquad X^k = \theta_1 Z^k + \theta_2 V^k + (1 - \theta_1 - \theta_2)Y^k$

$\qquad g^k = \mathbf{G}(X^k) - \mathbf{G}(V^k) + \nabla\mathbf{F}(V^k)$

$\qquad Y^{k+1} = \text{prox}_{\eta\psi}(X^k - \eta g^k)$

$\qquad Z^{k+1} = \nu Z^k + (1 - \nu)X^k + \frac{\gamma}{\eta}(Y^{k+1} - Y^k)$

$$V^{k+1} = \begin{cases} Y^k, & \text{with probability } \rho \\ V^k, & \text{with probability } 1 - \rho \end{cases}$$

$\quad$ **end for**

---

# 5 Simulations

We present an extensive numerical evaluation to verify and support the theoretical claims. We perform experiments on both synthetic and real data, with a range of different objectives and methods (both ours and the baselines from the literature). The experiments are designed to shed light on various aspects of the theory. In this section, we present the results on synthetic data, while in the next section, we illustrate the performance of different methods on real data. The code to reproduce the experiments is publicly available at

https://github.com/boxinz17/PFL-Unified-Framework.

The experiments on synthetic data were conducted on a personal laptop with a CPU (Intel(R) Core(TM) i7-9750H CPU@2.60GHz). The results are summarized over 30 independent runs.

## 5.1 Multi-Task Personalized FL and Implicit MAML Objective

In this section, we focus on the performance of different methods when solving the objective (11). We implement three proposed algorithms – LSGD-PFL, ASCD-PFL[3], and ASVRCD-PFL – and compare them with two baselines – L2SGD+ (Hanzely & Richtárik, 2020) and pFedMe (Dinh et al., 2020). As both L2SGD+ and pFedMe were designed specifically to solve (11), the aim of this experiment is to demonstrate that our universal approach is competitive with these specifically designed methods.

**Data and model.** We perform this experiment on synthetically generated data which allows us to properly control the data heterogeneity level. As a model, we choose logistic regression. We generate $w^* \in \mathbb{R}^d$ with i.i.d. entries from Uniform$[0.49, 0.51]$, and set $\beta_m^* = w^* + \Delta\beta_m^* \in \mathbb{R}^d$, where entries of $\Delta\beta_m^*$ are generated i.i.d. from Uniform$[\mu_m - 0.01, \mu_m + 0.01]$ and $\mu_m \sim N(0, \sigma_h^2)$ for all $m = 1, 2, \ldots, M$. Thus, $\sigma_h$ can be regarded as a measure of heterogeneity level, with a large $\sigma_h$ corresponding to large heterogeneity. Finally, for each device $m = 1, 2, \ldots, M$, we generate $\mathbf{x}_{m,i} \in \mathbb{R}^d$ with entries i.i.d. from Uniform$[0.2, 0.5]$ for all $i = 1, 2, \ldots, n$, and $y_{m,i} \sim$ Bernoulli$(p_{m,i})$, where $p_{m,i} = 1/(1 + \exp(\beta_m^{*\top}\mathbf{x}_{m,i}))$. We set $d = 15$, $n = 1000$, $M = 20$, and let $\sigma_h \in \{0.1, 0.3, 1.0\}$ to explore different levels of heterogeneity.

**Objective function.** We use objective (11) with $f_m'(\beta_m)$ being the cross-entropy loss function. We set $\lambda = \sigma_h \cdot 10^{-2}$, so that larger heterogeneity level will induce a larger penalty, which will further encourage parameters on each device to be closer to their geometric center. In addition to the training loss, we also record the estimation error in the training process, defined as $\|\hat{w} - w^*\|^2 + \sum_{m=1}^{M} \|\hat{\beta}_m - \beta_m^*\|^2$.

**Tuning parameters of proposed algorithms.** For LSGD-PFL (Algorithm 1), we set the batch size to compute the stochastic gradient $B = 1$, the average period $\tau = 5$, and the learning rate $\eta = 0.01$. For $p_w$ and $p_\beta$ in ASCD-PFL (Algorithm 5) and ASVRCD-PFL (Algorithm 4), we set them as $p_w = \mathcal{L}^w/(\mathcal{L}^\beta + \mathcal{L}^w)$ and $p_\beta = 1 - p_w$, where $\mathcal{L}^w = \lambda/M$ and $\mathcal{L}^\beta = (\mathcal{L}' + \lambda)/M$. We set $\mathcal{L}' = \max_{1 \le m \le M, 1 \le i \le n} \|\mathbf{x}_{m,i}\|^2/4$. For $\eta, \theta_2, \gamma, \nu$ and $\theta_1$ in ASVRCD-PFL, we set them according to Theorem 9, where $\mathcal{L} = 2\max\{\mathcal{L}^w/p_w, \mathcal{L}^\beta/p_\beta\}$, $\rho = p_w/n$, and $\mu = \mu'/(3M)$. We let $\mu'$ be the smallest eigenvalue of $\frac{1}{nM} \sum_{m=1}^{M} \sum_{i=1}^{n} \exp(\beta_m^{*\top}\mathbf{x}_{m,i})/(1 +$

---

[3]ASCD-PFL is ASVRCD-PFL without control variates. See the detailed description in Appendix A.

---

**Algorithm 4** ASVRCD-PFL

---

**input** $0 < \theta_1, \theta_2 < 1$, $\eta, \nu, \gamma > 0$, $\rho \in (0,1)$, $p_w \in (0,1)$, $p_\beta = 1 - p_w$, $w_y^0 = w_z^0 = w_v^0 \in \mathbb{R}^{d_0}$, $\beta_{y,m}^0 = \beta_{z,m}^0 = \beta_{v,m}^0 \in \mathbb{R}^{d_m}$ for $1 \leq m \leq M$.

  **for** $k = 0, 1, 2, \ldots$ **do**

    $w_x^k = \theta_1 w_z^k + \theta_2 w_v^k + (1 - \theta_1 - \theta_2) w_y^k$

    **for** $m = 1, \ldots, M$ in parallel **do**

      $\beta_{x,m}^k = \theta_1 \beta_{z,m}^k + \theta_2 \beta_{v,m}^k + (1 - \theta_1 - \theta_2)\beta_{y,m}^k$

    **end for**

    Sample random $j \in \{1, 2, \ldots, n\}$ and $\zeta = \begin{cases} 1, & \text{with probability } p_w \\ 2, & \text{with probability } p_\beta \end{cases}$

$$g_w^k = \begin{cases} \dfrac{1}{p_w}\left(\dfrac{1}{M}\sum_{m=1}^{M}\nabla_w f_{m,j}(w_x^k, \beta_{x,m}^k) - \dfrac{1}{M}\sum_{m=1}^{M}\nabla_w f_{m,j}(w_v^k, \beta_{v,m}^k)\right) + \nabla_w F(w_v^k, \beta_v^k) & \text{if } \zeta = 1 \\[6mm] \nabla_w F(w_v^k, \beta_v^k) & \text{if } \zeta = 2 \end{cases}$$

    $w_y^{k+1} = w_x^k - \eta g_w^k$

    $w_z^{k+1} = \nu w_z^k + (1 - \nu)w_x^k + \frac{\gamma}{\eta}(w_y^{k+1} - w_x^k)$

    $w_v^{k+1} = \begin{cases} w_y^k, & \text{with probability } \rho \\ w_v^k, & \text{with probability } 1 - \rho \end{cases}$

    **for** $m = 1, \ldots, M$ in parallel **do**

$$g_{\beta,m}^k = \begin{cases} \dfrac{1}{M}\nabla_\beta f_m(w_v^k, \beta_{v,m}^k) & \text{if } \zeta = 1 \\[6mm] \dfrac{1}{p_\beta M}\left(\nabla_\beta f_{m,j}(w_x^k, \beta_{x,m}^k) - \nabla_\beta f_{m,j}(w_v^k, \beta_{v,m}^k)\right) + \dfrac{1}{M}\nabla_\beta f_m(w_v^k, \beta_{v,m}^k) & \text{if } \zeta = 2 \end{cases}$$

      $\beta_{y,m}^{k+1} = \beta_{x,m}^k - \eta g_{\beta,m}^k$

      $\beta_{z,m}^{k+1} = \nu \beta_{z,m}^k + (1 - \nu)\beta_{x,m}^k + \frac{\gamma}{\eta}(\beta_{y,m}^{k+1} - \beta_{x,m}^k)$

      $\beta_{v,m}^{k+1} = \begin{cases} \beta_{y,m}^k, & \text{with probability } \rho \\ \beta_{v,m}^k, & \text{with probability } 1 - \rho \end{cases}$

    **end for**

  **end for**

---

$\exp(\beta_m^{*\top}\mathbf{x}_{m,i}))^2 \mathbf{x}_{m,i}\mathbf{x}_{m,i}^\top$. The $\eta, \nu, \gamma, \rho$ in ASCD-PFL are the same as in ASVRCD-PFL, and we let $\theta = \min\{0.8, 1/\eta\}$. In addition, we initialize all iterates at zero for all algorithms.

**Tuning Parameters of pFedMe.** For pFedMe (Algorithm 1 in Dinh et al. (2020)), we set all parameters according to the suggestions in Section 5.2 of Dinh et al. (2020). Specifically, we set the local computation rounds to $R = 20$, computation complexity to $K = 5$, Mini-Batch size to $|\mathcal{D}| = 5$, and $\eta = 0.005$. We also set $S = M = 20$. To solve the objective (7) in Dinh et al. (2020), we use gradient descent, as suggested in the paper. Additionally, we initialize all iterates at zero.

**Tuning Parameters of L2SGD+.** For L2SGD+ [4], we set the stepsize $\eta$ (the parameter $\alpha$ in Hanzely & Richtárik (2020)) and the probability of averaging $p_w$ to be the same as in ASRVCD-PFL and ASCD-PFL. We also initialize all iterates at zero.

---

[4]Algorithm 2 in Hanzely & Richtárik (2020)

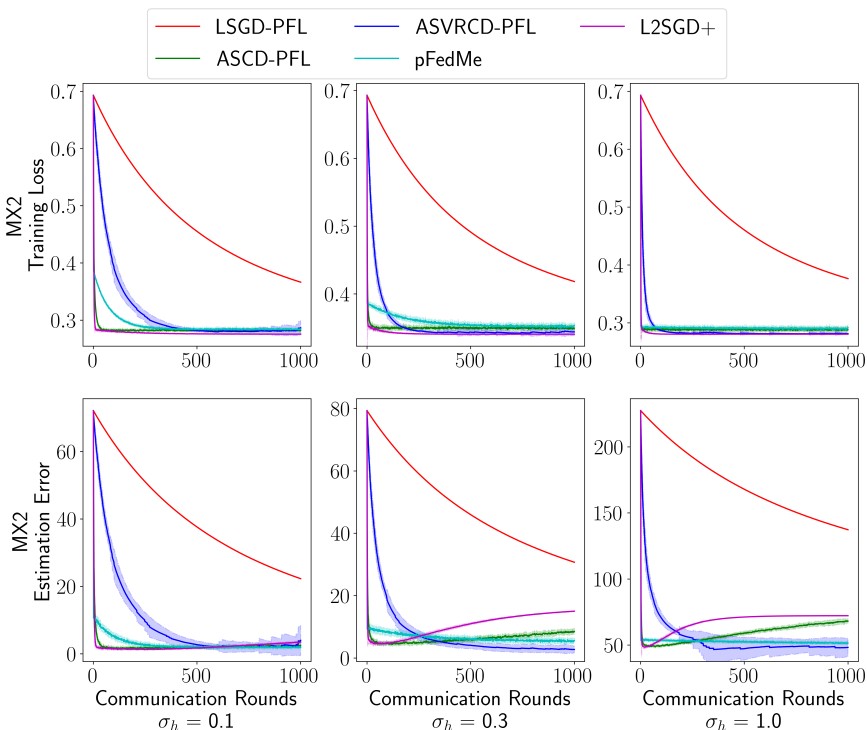

Figure 1: Comparison of various methods for minimizing (11). Each experiment was repeated 30 times, and the solid line represents the average performance, while the shaded region indicates the mean $\pm$ standard error. We set the communication period of LSGD-PFL at 5, while other methods synchronize based on the corresponding theory. The first row shows the training loss, and the second row shows the estimation error. The different columns correspond to varying heterogeneity levels, which are parameterized by $\sigma_h$. As $\sigma_h$ increases, the heterogeneity level also increases.

| Algorithm $\diagdown$ $\sigma_h$ | 0.1 | 0.3 | 1.0 |
|---|---|---|---|
| LSGD | 260.98 | 256.16 | 237.18 |
| ASCD | 28.79 | 51.71 | 142.59 |
| ASVRCD | 47.94 | 97.61 | 274.60 |
| pFedMe | 399.33 | 370.42 | 370.35 |
| L2SGDplus | 25.83 | 47.59 | 182.29 |

Table 4: The average wall-clock running time in seconds over 30 independent runs when solving the objective (11). Each entry in the table reports the average time for 1,000 communication rounds. We ignore any additional communication costs that might occur in practice.

**Results.** The results are summarized in Figure 1. We observe that our general-purpose optimizers are competitive with L2SGD+ and pFedMe. In particular, both ASVRCD-PFL and L2SGD+ consistently achieve the same training error as the other methods, which is well predicted by our theory. Although L2SGD+ is slightly faster in terms of convergence due to the specific parameter setting, it is not as general as the methods we propose. Furthermore, we note that the widely used LSGD-PFL suffers from data heterogeneity on different devices, whereas ASVRCD-PFL is not affected by this heterogeneity, as predicted by our theory. The average running time over 30 independent runs is reported in Table 4.

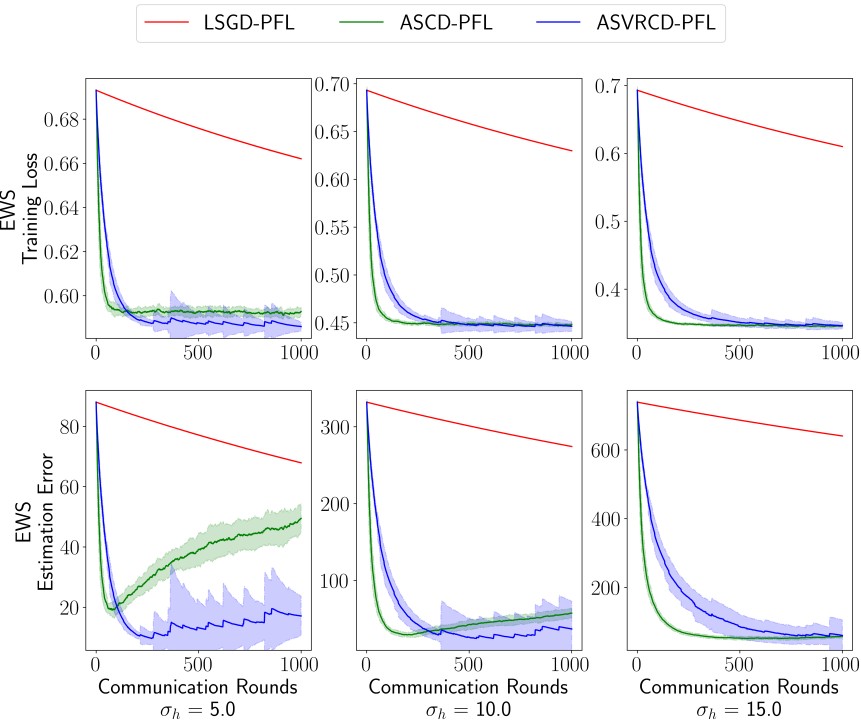

Figure 2: Comparison of the three proposed algorithms – LSGD-PFL, ASCD-PFL, and ASVRCD-PFL – when minimizing (16). Each experiment is repeated 30 times; the solid line represents the mean performance, and the shaded region covers the mean ± standard error. We set the communication period of LSGD-PFL to 5, while other methods synchronize according to their respective theories. The first row represents the training loss, and the second row corresponds to the estimation error. Different columns indicate various heterogeneity levels, parameterized by $\sigma_h$. The heterogeneity level increases with $\sigma_h$.

## 5.2 Explicit Weight Sharing Objective

In this section, we present another experiment on synthetic data that aims to optimize the explicit weight sharing objective (16). Since there is no good baseline algorithm for this objective, the purpose of this experiment is to compare the three proposed algorithms: LSGD-PFL, ASCD-PFL, and ASVRCD-PFL.

**Data and Model.** As a model, we choose logistic regression. We generate $w^* \in \mathbb{R}^{d_g}$ with i.i.d. entries from $N(0,1)$, and $\beta_m^* \in \mathbb{R}^{d_l}$ with i.i.d. entries from Uniform$[\mu_m - 0.01, \mu_m + 0.01]$, where $\mu_m \sim N(0, \sigma_h^2)$ for all $m = 1, 2, \ldots, M$. Thus, $\sigma_h$ can be regarded as a measure of the heterogeneity level, with a large $\sigma_h$ corresponding to a high degree of heterogeneity. Finally, for each device $m = 1, 2, \ldots, M$, we generate $\mathbf{x}_{m,i} \in \mathbb{R}^{d_g + d_l}$ with entries i.i.d. from Uniform$[0.0, 0.1]$ for all $i = 1, 2, \ldots, n$, and $y_{m,i} \sim$ Bernoulli$(p_{m,i})$, where $p_{m,i} = 1/(1 + \exp((w^{*\top}, \beta_m^{*\top})\mathbf{x}_{m,i}))$. We set $d_g = 10$, $d_l = 5$, $n = 1000$, $M = 20$, and let $\sigma_h \in \{5.0, 10.0, 15.0\}$ to explore different levels of heterogeneity.

**Objective function.** We use objective (11) with $f'_m(\beta_m)$ representing the cross-entropy loss function. We set $\lambda = \sigma_h \cdot 10^{-2}$, so smaller heterogeneity levels will induce a larger penalty, encouraging parameters on each device to be closer to their geometric center. In addition to the training loss, we also record the estimation error during the training process, defined as $\|\hat{w} - w^*\|^2 + \sum_{m=1}^{M} \|\hat{\beta}_m - \beta_m^*\|^2$.

**Results.** The results are summarized in Figure 2. When examining the training loss, we observe that ASCD-PFL drives the loss down quickly initially, while ASVRCD-PFL ultimately achieves a better optimum. This suggests that we can apply ASCD-PFL at the beginning of training and add control variates to reduce the variance at a later stage of training, thus combining the benefits of both algorithms. Both ASCD-PFL and ASVRCD-PFL perform much better than the widely used LSGD-PFL. When analyzing the estimation

| Algorithms $\diagdown$ $\sigma_h$ | 5.0 | 10.0 | 15.0 |
|---|---|---|---|
| LSGD | 238.01 | 237.67 | 234.33 |
| ASCD | 17.00 | 16.55 | 16.46 |
| ASVRCD | 23.35 | 22.12 | 23.51 |

Table 5: The average wall-clock running time in seconds over 30 independent runs when solving the objective (16) is presented. Each entry of the table reports the average time for 1,000 communication rounds. We ignore any additional communication costs that might occur in practice.

error, we observe that when the heterogeneity level is small, there is a tendency for overfitting, especially for ASCD-PFL; and when the heterogeneity level increases, there is less concern for overfitting. In general, however, ASCD-PFL and ASVRCD-PFL still achieve better estimation error than LSGD-PFL. The average running time over 30 independent runs is reported in Table 5.

## 6 Real Data Experiment Results

In this section, we use real data to illustrate the performance and various properties of the proposed methods. In Section 6.1, we compare the performance of the three proposed algorithms. In Section 6.2, we illustrate the effect of communication frequency of global parameters for ASCD-PFL and demonstrate that the theoretical choice based on Theorem 9 can generate the best communication complexity. Finally, in Section 6.3, we show the effect of reparametrizing $w$ for ASCD-PFL.

### 6.1 Performance of the Proposed Methods on Real Data

We compare the three proposed algorithms – LSGD-PFL, ASCD-PFL (ASVRCD-PFL without control variates), and ASVRCD-PFL – across four image classification datasets – MNIST (Deng, 2012), KMIN-IST (Clanuwat et al., 2018), FMINST (Xiao et al., 2017), and CIFAR-10 (Krizhevsky, 2009) with three objective functions (8), (11), and (14). As a model, we use a multiclass logistic regression, which is a single-layer fully connected neural network combined with a softmax function and cross-entropy loss. Experiments were conducted on a personal laptop (Intel(R) Core(TM) i7-9750H CPU@2.60GHz) with a GPU (NVIDIA GeForce RTX 2070 with Max-Q Design).

**Data preparation.** We set the number of devices $M = 20$. We focus on a non-i.i.d. setting of McMahan et al. (2017) and Liang et al. (2020) by assigning $K$ classes out of ten to each device. We let $K = 2, 4, 8$ to generate different levels of heterogeneity. A larger $K$ means a more balanced data distribution and thus smaller data heterogeneity. We then randomly select $n = 100$ samples for each device based on its class assignment for training and $n' = 300$ samples for testing. We normalize each dataset in two steps: first, we normalize the columns (features) to have mean zero and unit variance; next, we normalize the rows (samples) so that every input vector has a unit norm.

**Model.** Given a grayscale picture with a label $y \in \{1, 2, \ldots, C\}$, we unroll its pixel matrix into a vector $x \in \mathbb{R}^p$. Then, given a parameter matrix $\Theta \in \mathbb{R}^{p \times C}$, the function $f'_m(\cdot)$ in (8), (11), and (14) is defined as

$$f'_m(\Theta) := l_{\mathrm{CE}}\left(\varsigma\left(\Theta x\right); y\right),$$

where $\varsigma(\cdot) : \mathbb{R}^K \to \mathbb{R}^K$ is the softmax function and $l_{\mathrm{CE}}(\cdot)$ is the cross-entropy loss function. In this setting, the function $f'_m(\cdot)$ is convex.

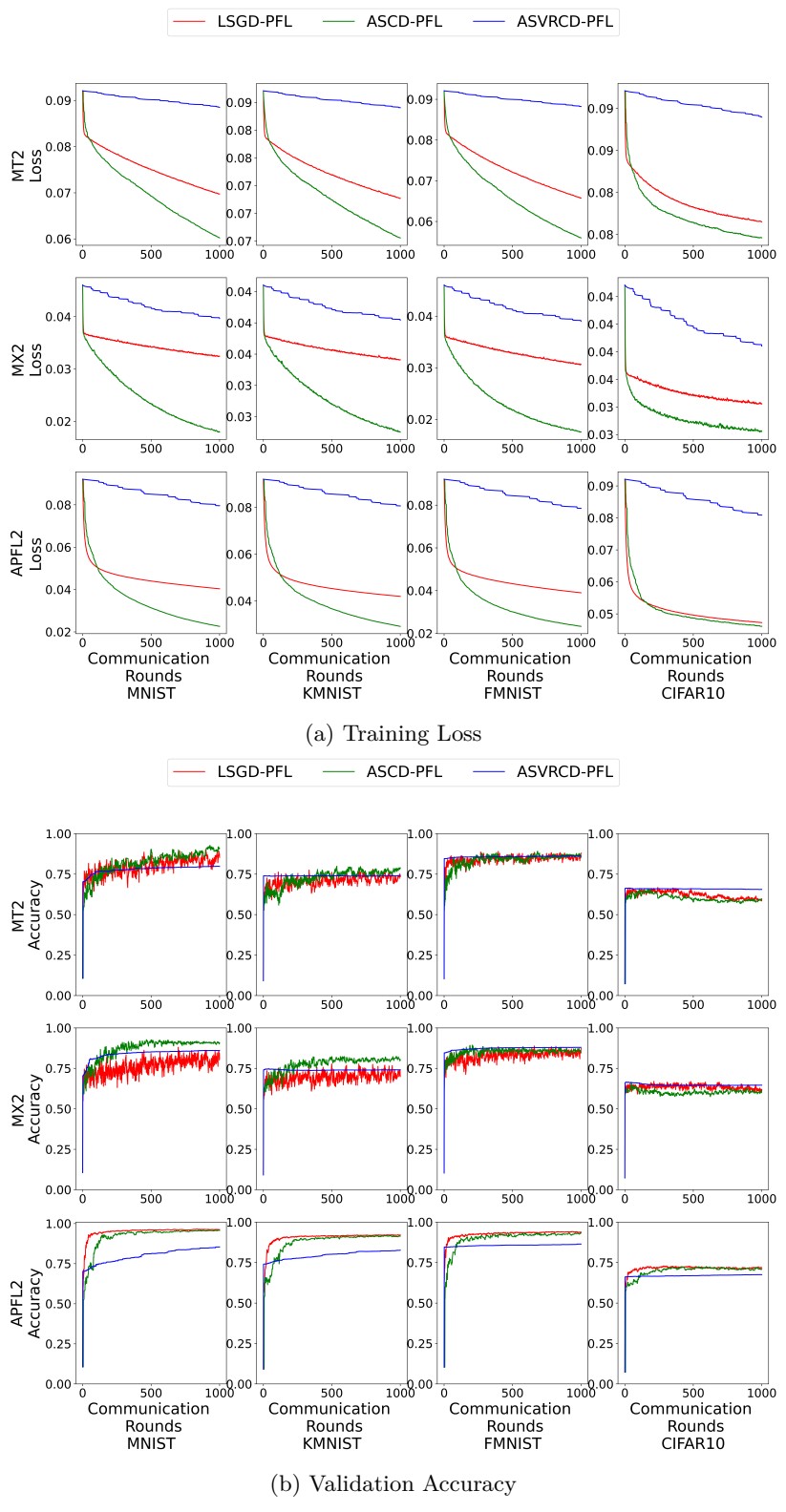

Figure 3: The real data experiment results for $K = 2$. Different rows correspond to different objective functions, and columns correspond to different datasets.

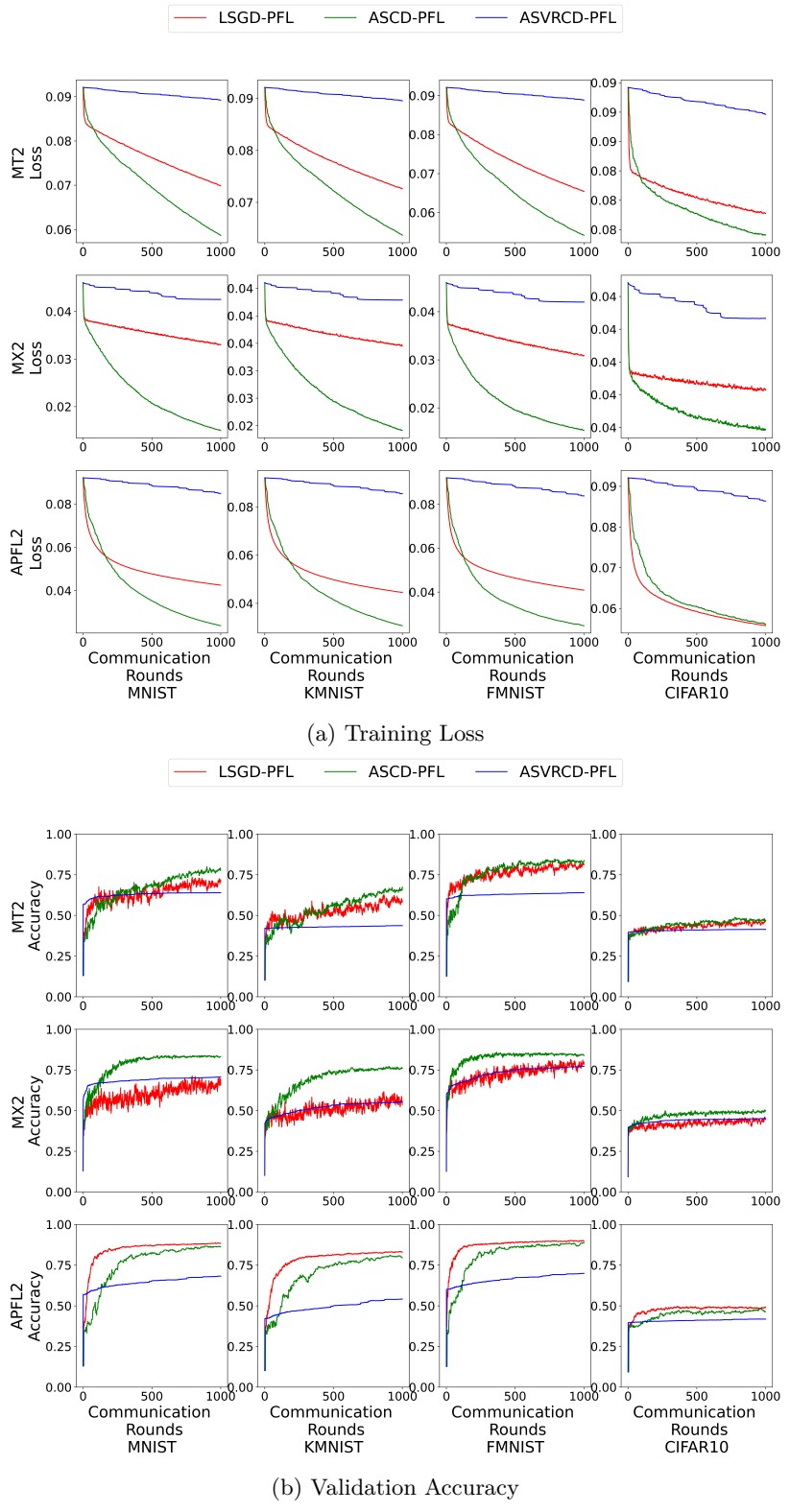

Figure 4: The real data experiment results for $K = 4$. Different rows orrespond to different objective functions, and columns correspond to different datasets.

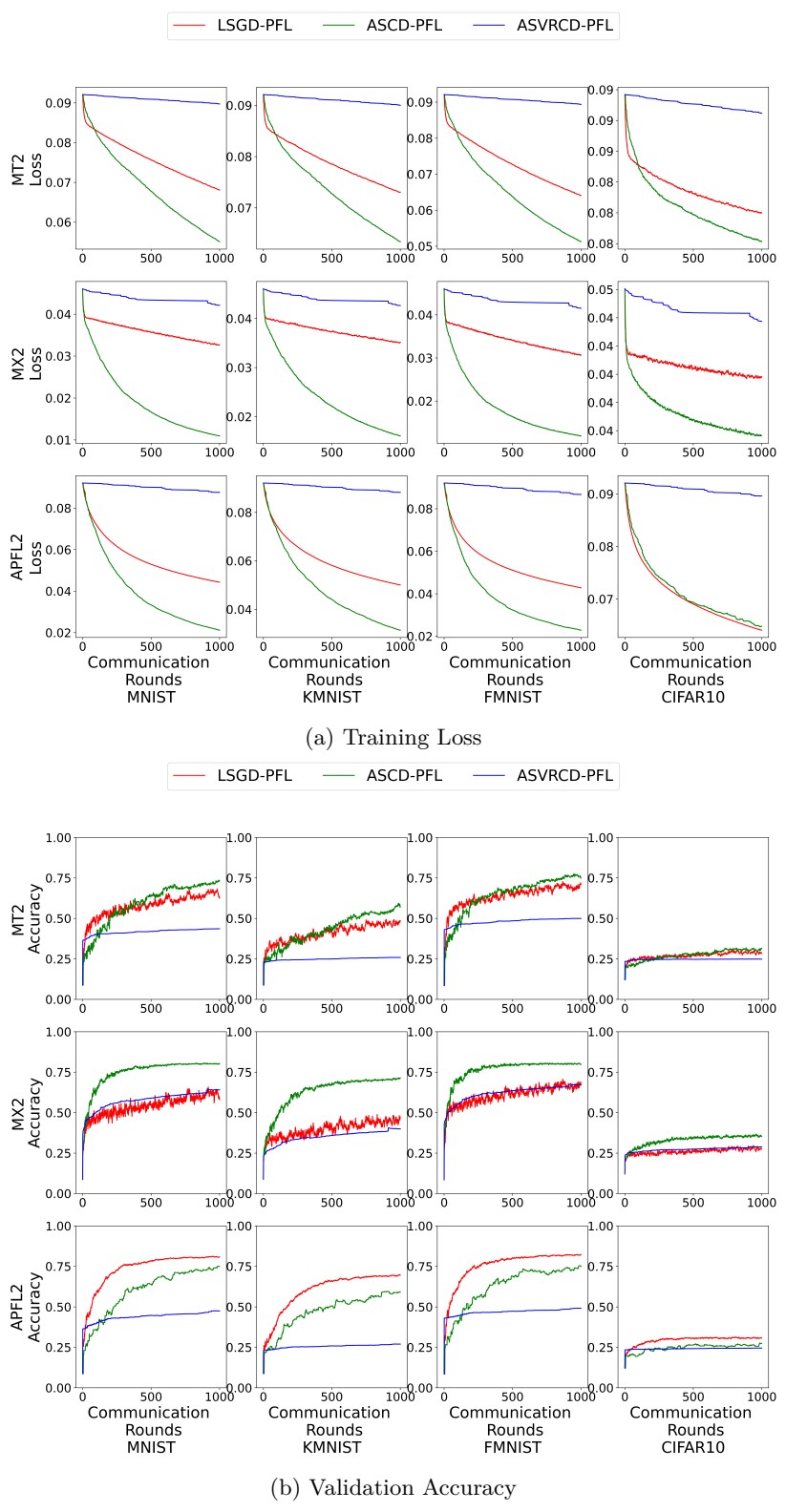

Figure 5: The real data experiment results for $K = 8$. Different rows correspond to different objective functions, and columns correspond to different datasets.

**Personalized FL objectives.** We consider three different objectives:

1. the multitask FL objective (8) with $\Lambda = 1.0$ and $\lambda = 1.0/K$;

2. the mixture FL objective (11), with $\lambda = 1.0/K$; and

3. the adaptive personalized FL objective (14), with $\Lambda = 1.0$ and $\alpha_m = 0.05 \times K$ for all $m \in [M]$,

where $K$ is the number of labels for each device. When computing the testing accuracy, we only use the local parameters. Note that the choices of hyperparameters in the chosen objectives are purely heuristic. Our purpose is to demonstrate the convergence properties of the proposed algorithms on the training loss. Thus, it is possible that a smaller training loss does not necessarily imply better testing accuracy (or generalization ability). How to choose hyperparameters optimally is not the focus of this paper and requires further research.

**Tuning parameters of proposed algorithms.** For LSGD-PFL (Algorithm 1), we set the batch size to compute the stochastic gradient $B = 1$ and set the average period $\tau = 5$. For $p_w$ in ASCD-PFL (Algorithm 6) and ASVRCD-PFL (Algorithm 7), we set it as $p_w = \mathcal{L}^w/(\mathcal{L}^\beta + \mathcal{L}^w)$. For the objective $F_{MT2}$ in (8), we set $\mathcal{L}^w = (\Lambda \mathcal{L}' + \lambda)/M$ and $\mathcal{L}^\beta = \mathcal{L}' + \lambda$; for the objective $F_{MX2}$ in (11), we set $\mathcal{L}^w = \lambda/M$ and $\mathcal{L}^\beta = \mathcal{L}' + \lambda$; for the objective $F_{APFL2}$ in (14), we set $\mathcal{L}^w = (\Lambda + \max_{1 \leq m \leq M} \alpha_m^2)\mathcal{L}'/M$ and $\mathcal{L}^\beta = (1 - \max_{1 \leq m \leq M} \alpha_m)^2 \mathcal{L}'/M$. We set $\mathcal{L}' = 1.0$ for all objectives. We set $\rho = p_w/n$ for ASCD-PFL and ASVRCD-PFL. For $\eta, \theta_2, \gamma, \nu$ and $\theta_1$ in ASVRCD-PFL, we set them according to Theorem 9, where $\mathcal{L} = 2\max\{\mathcal{L}^w/p_w, \mathcal{L}^\beta/p_\beta\}$, $\rho = p_w/n$, and $\mu = \mu'/(3M)$. We let $\mu' = 0.01$. Since the dimension of the iterates is larger than the sample size, the objective is weakly convex and, thus, $\mu' = 0$. Therefore, our choice of $\mu'$ is aimed at improving the numerical behavior of algorithms. The $\eta, \nu, \gamma, \rho$ in ASCD-PFL are the same as in ASVRCD-PFL, and we let $\theta = \min\{0.8, 1/\eta\}$. In addition, we initialize all iterates at zero for all algorithms.

**Results.** The results are summarized in Figure 3, Figure 4, and Figure 5 for $K = 2, 4, 8$ respectively. We observe that ASCD-PFL outperforms the widely-used LSGD-PFL. We also observe that ASVRCD-PFL converges slowly when minimizing the training loss. As we are working in the over parametrization regimes, which makes $\mu' = 0$, the assumptions of our theory are violated. As a result, it is more advisable to use ASCD-PFL during the initial phase of training and use ASVRCD-PFL when the iterates get closer to the optimum.

## 6.2 Subsampling of the Global and Local Parameters

We demonstrate that the choice of $p_w$ based on Theorem 9, specifically setting $p_w = \mathcal{L}^w/(\mathcal{L}^w + \mathcal{L}^\beta)$, results in the best communication complexity of ASCD-PFL. More precisely, based on Theorem 9, we set the learning rate $\eta = 1/(4\mathcal{L})$, where $\mathcal{L} := 2\max\{\mathcal{L}^w/p_w, \mathcal{L}^\beta/p_\beta\}$. The expressions of $\mathcal{L}^w$ and $\mathcal{L}^\beta$ for $F_{MT2}$, $F_{MX2}$, and $F_{APFL2}$ are stated in Lemma 1, Lemma 2, Lemma 3, and also restated in the previous section, where $\mathcal{L}'$ is 1 after normalization. We set $\rho = p_w/n$. We compare the performance of ASCD-PFL using the theoretically suggested $p_w$ with other choices of $p_w \in \{0.1, 0.3, 0.5, 0.7, 0.9\}$. We fix those parameters that are independent of $p_w$. See more details about the choice of tuning parameters in the previous section.

We plot the loss against the number of communication rounds, which illustrates the communication complexity. The number of classes for each device is $K = 2$. The results are summarized in Figure 6, which also includes the test accuracy. We observe that choosing $p_w$ based on theoretical considerations leads to the best communication complexity.

## 6.3 Effect of Reparametrization in ASCD-PFL.

We demonstrate the importance of reparametrization of the global parameter $w$ by rescaling $w$ by a factor of $M^{-\frac{1}{2}}$. We run reparameterized and non-reparameterized ASCD-PFL across different objectives and datasets. We set the number of classes for each device $K = 2$. The results are summarized in Figure 7. For the training loss, we observe that reparametrization improves the convergence of ASCD-PFL, except for the APFL2 objective (14), where the non-reparametrized variant performs slightly better in three of the four datasets. On the other hand, when considering the testing accuracy, reparametrization always helps improve the results, indicating that reparametrization can help prevent overfitting. Based on the experiment, we

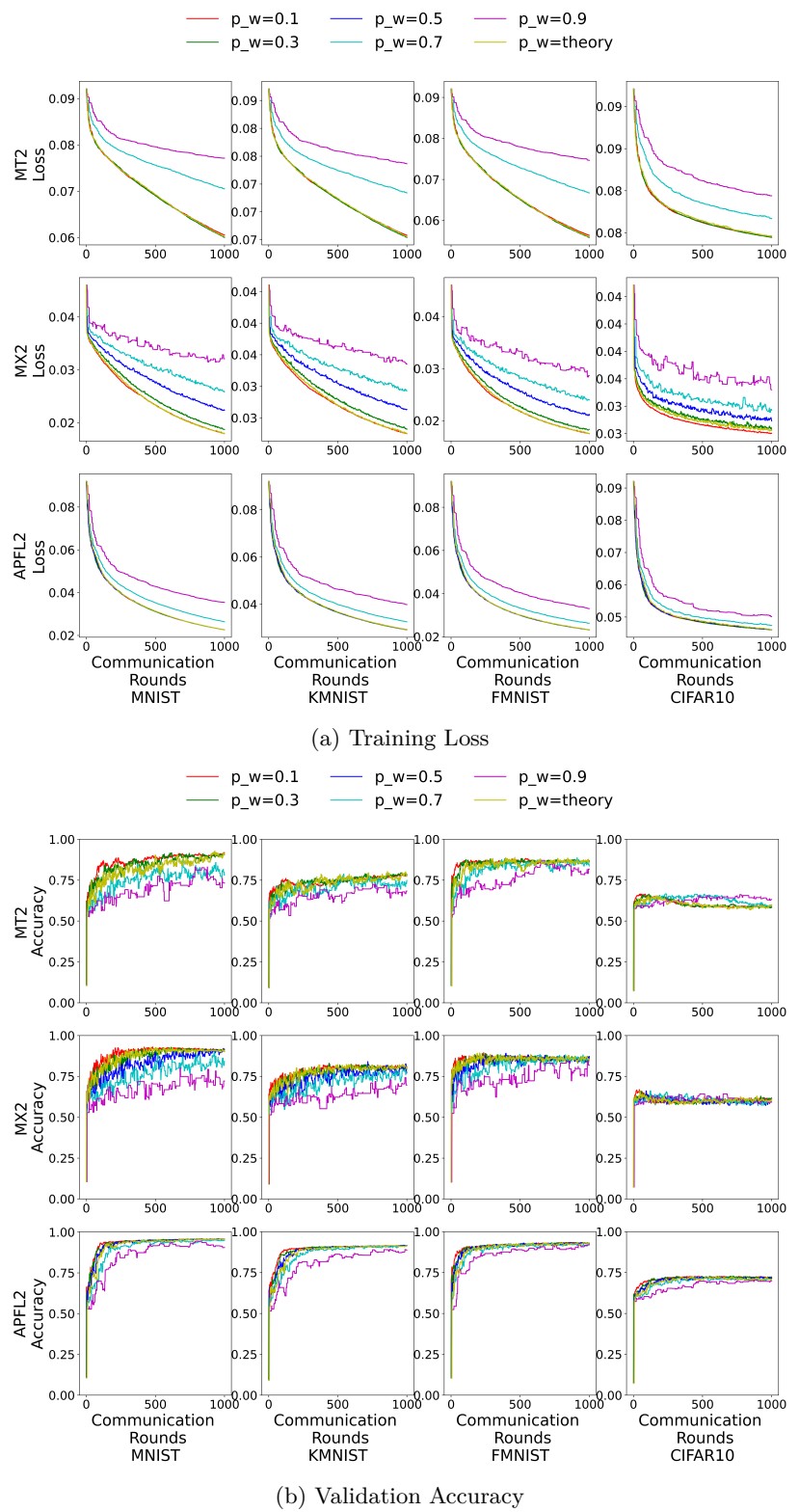

Figure 6: Communication complexity for different choices of $p_w$. The theoretical choice based on Theorem 9 can provide the best communication complexity. Specifically, the theoretical choice of $p_w$ results in $p_w = 0.5$ for $F_{MT2}$, $p_w = 0.25$ for $F_{MX2}$, and $p_w = 0.55$ for $F_{APFL2}$. Different rows correspond to different objective functions, and columns correspond to different datasets.

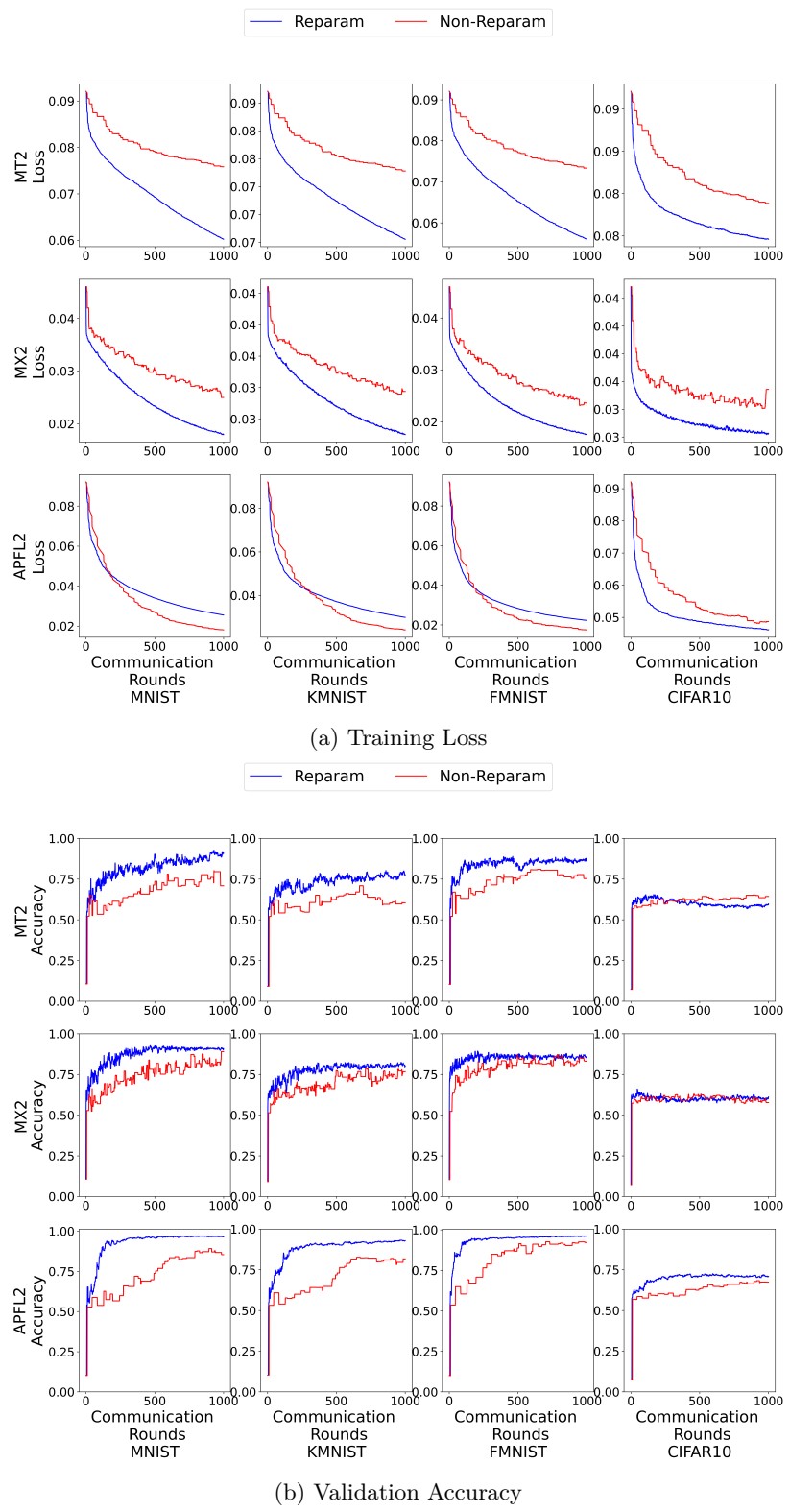

Figure 7: Effect of reparametrization of global space in ASCD-PFL. Reparametrization generally helps achieve faster convergence in minimizing training loss and consistently improves testing accuracy. Different rows correspond to different objective functions, and columns correspond to different datasets.

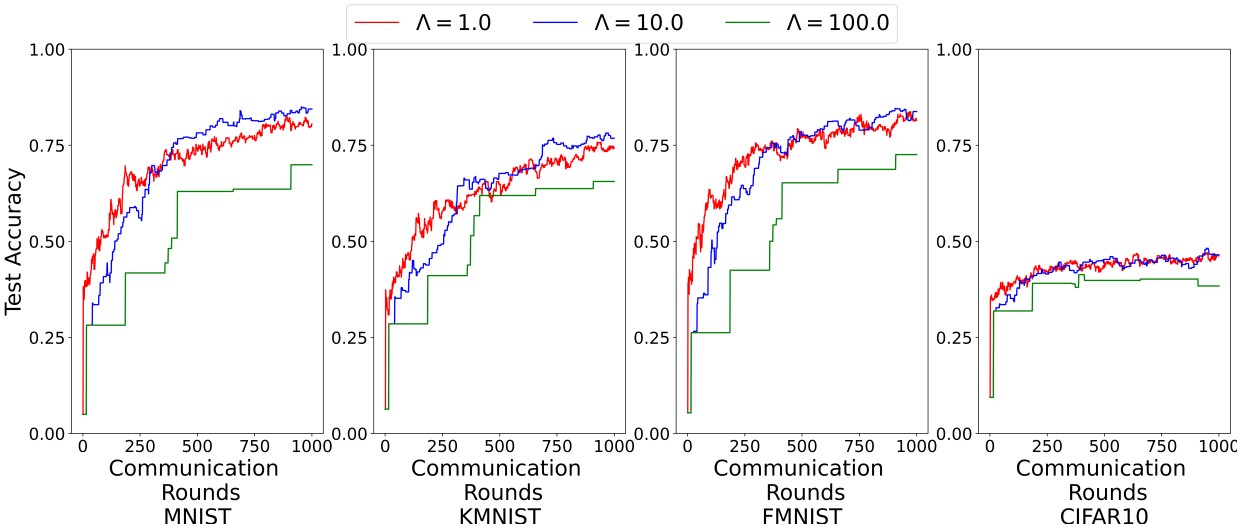

Figure 8: Empirical exploration of the effect of $\Lambda$ on the performance of the objective in (7).

suggest always using reparametrization to ensure the scale of the learning rate is appropriate for both global and local parameters.

### 6.4 Potential Benefits of Extended Objectives

We empirically justify the potential benefits of our extended objectives. More specifically, we vary the relaxation parameter $\Lambda$ in (7) to show the change in performance. Since the multitask objective proposed by Li et al. (2020) is equivalent to setting $\Lambda \to \infty$, our main interest is to explore whether a larger $\Lambda$ always implies better performance. We set $K = 4$, $n = 30$, $n' = 100$, $M = 20$, and $\lambda = 0.1$. The remaining settings are the same as in Section 6.1. We vary $\Lambda$ over the values $\{1.0, 10.0, 100.0\}$, and the resulting performance is shown in Figure 8. The plot shows that the performance slightly improves as $\Lambda$ increases from 1.0 to 10.0, but then drops when $\Lambda = 100.0$. This result suggests that by selecting an appropriate $\Lambda$, it is possible to achieve better empirical performance. Furthermore, although proposing new personalized FL objectives is not the main focus of this paper, the above empirical result suggests the potential benefits of a general framework.

## 7 Conclusions and Directions for Future Research

We proposed a general convex optimization theory for personalized FL. While our work answers a range of important questions, there are many directions in which our work can be extended in the future, such as partial participation, minimax optimal rates for specific personalized FL objectives, brand new personalized FL objectives, and non-convex theory.

**Partial participation and client sampling.** An essential aspect of FL that is not covered in this work is the partial participation or client sampling when one has access to only a subset of devices at each iteration. While we did not cover partial participation and focused on answering orthogonal questions, we believe that partial participation should be considered when extending our results in the future. Typically, when one chooses clients uniformly, the theorems in this paper should be extended easily; however, a more interesting question is how to sample clients with a non-uniform distribution to speed up the convergence. We leave this problem for future study.

**Minimax optimal rates for specific personalized FL objectives.** As outlined in Section 1.2, one cannot hope for the general optimization framework to be minimax optimal in every single special case. Consequently, there is still a need to keep exploring the optimization aspects of individual personalized FL

objectives as one might come up with a more efficient optimizer that exploits the specific structure not covered by Assumption 1 or Assumption 2.

**Brand new personalized FL objectives.** While in this work we propose a couple of novel personalized FL objectives obtained as an extension of known objectives, we believe that seeing personalized FL as an instance of (1) might lead to the development of brand new approaches for personalized FL.

**Non-convex theory.** In this work, we have focused on a general convex optimization theory for personalized FL. Our convex rates are meaningful – they are minimax optimal and correspond to the empirical convergence. However, an inherent drawback of such an approach is the inability to cover non-convex FL approaches, such as MAML (see Section 2.8), or non-convex FL models. We believe that obtaining minimax optimal rates in the non-convex world would be very valuable.

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

# A    Additional Algorithms Used in Simulations

In this section, we detail algorithms that are used in Section 5. We detail ASCD-PFL in Algorithm 5. ASCD-PFL is a simplified version of ASVRCD-PFL that does not incorporate control variates. SCD-PFL is detailed in Algorithm 6. SCD-PFL is a simplified version of ASCD-PFL that does not incorporate the control variates or Nesterov's acceleration. SVRCD-PFL is detailed in Algorithm 7. SVRCD-PFL is a simplified version of ASVRCD-PFL that does not incorporate Nesterov's acceleration.

---

**Algorithm 5** ASCD-PFL

---

**input** $0 < \theta < 1$, $\eta, \nu, \gamma > 0$, $\rho \in (0, 1)$, $p_w \in (0, 1)$, $p_\beta = 1 - p_w$, $w_y^0 = w_z^0 \in \mathbb{R}^{d_0}$, $\beta_{y,m}^0 = \beta_{z,m}^0 \in \mathbb{R}^{d_m}$ for $1 \le m \le M$.

 **for** $k = 0, 1, 2, \ldots$ **do**

  $w_x^k = \theta w_z^k + (1 - \theta) w_y^k$

  **for** $m = 1, \ldots, M$ in parallel **do**

   $\beta_{x,m}^k = \theta \beta_{z,m}^k + (1 - \theta) \beta_{y,m}^k$

  **end for**

  Sample random $j \in \{1, 2, \ldots, n\}$ and $\zeta = \begin{cases} 1 & \text{with probability } p_w \\ 2 & \text{with probability } p_\beta \end{cases}$

  **if** $\zeta = 1$ **then**

   $g_w^k = \frac{1}{p_w M} \sum_{m=1}^M \nabla_w f_{m,j}(w_x^k, \beta_{x,m}^k)$

   $w_y^{k+1} = w_x^k - \eta g_w^k$

   $w_z^{k+1} = \nu w_z^k + (1 - \nu) w_x^k + \frac{\gamma}{\eta}(w_y^{k+1} - w_x^k)$

   $w_v^{k+1} = \begin{cases} w_y^k, & \text{with probability } \rho \\ w_v^k, & \text{with probability } 1 - \rho \end{cases}$

  **else**

   **for** $m = 1, \ldots, M$ in parallel **do**

    $g_{\beta,m}^k = \frac{1}{p_\beta M} \nabla_\beta f_{m,j}(w_x^k, \beta_{x,m}^k)$

    $\beta_{y,m}^{k+1} = \beta_{x,m}^k - \eta g_{\beta,m}^k$

    $\beta_{z,m}^{k+1} = \nu \beta_{z,m}^k + (1 - \nu) \beta_{x,m}^k + \frac{\gamma}{\eta}(\beta_{y,m}^{k+1} - \beta_{x,m}^k)$

    $\beta_{v,m}^{k+1} = \begin{cases} \beta_{y,m}^k, & \text{with probability } \rho \\ \beta_{v,m}^k, & \text{with probability } 1 - \rho \end{cases}$

   **end for**

  **end if**

 **end for**

---

---

**Algorithm 6** SCD-PFL

---

**input** $\eta > 0$, $p_w \in (0,1)$, $p_\beta = 1 - p_w$, $w^0 \in \mathbb{R}^d$, $\beta_m^0 \in \mathbb{R}^d$ for $1 \le m \le M$.

 **for** $k = 0, 1, 2, \ldots K - 1$ **do**

  Sample random $j_m \in \{1, 2, \ldots, n_m\}$ for $1 \le m \le M$ and $\zeta = \begin{cases} 1 & \text{with probability } p_w \\ 2 & \text{with probability } p_\beta \end{cases}$

  $g_w^k = \begin{cases} \frac{1}{p_w M} \sum_{m=1}^M \nabla_w f_{m,j_m}(w^k, \beta_m^k) & \text{if } \zeta = 1 \\ 0 & \text{if } \zeta = 2 \end{cases}$

  $w^{k+1} = w^k - \eta g_w^k$

  **for** $m = 1, \ldots, M$ in parallel **do**

   $g_{\beta,m}^k = \begin{cases} 0 & \text{if } \zeta = 1 \\ \frac{1}{p_\beta M} \nabla_\beta f_{m,j_m}(w^k, \beta_m^k) & \text{if } \zeta = 2 \end{cases}$

   $\beta_m^{k+1} = \beta_m^k - \eta g_{\beta,m}^k$

  **end for**

 **end for**

**output** $w^K$, $\beta_m^K$ for $1 \le m \le M$.

---

---

**Algorithm 7** SVRCD-PFL

---

**input** $\eta > 0$, $p_w \in (0,1)$, $p_\beta = 1 - p_w$, $\rho \in (0,1)$, $w_y^0 = w_v^0 \in \mathbb{R}^d$, $\beta_{y,m}^0 = \beta_{v,m}^0 \in \mathbb{R}^d$ for $1 \le m \le M$.

 **for** $k = 0, 1, 2, \ldots K - 1$ **do**

  Sample random $j_m \in \{1, 2, \ldots, n_m\}$ for $1 \le m \le M$ and $\zeta = \begin{cases} 1 & \text{with probability } p_w \\ 2 & \text{with probability } p_\beta \end{cases}$

  $g_w^k = \begin{cases} \frac{1}{p_w M} \sum_{m=1}^M \nabla_w f_{m,j_m}(w_y^k, \beta_{y,m}^k) + \nabla_w F(w_v^k, \beta_v^k) & \text{if } \zeta = 1 \\ \nabla_w F(w_v^k, \beta_v^k) & \text{if } \zeta = 2 \end{cases}$

  $w_y^{k+1} = w_y^k - \eta g_w^k$

  $w_v^{k+1} = \begin{cases} w_y^k, & \text{with probability } \rho \\ w_v^k, & \text{with probability } 1 - \rho \end{cases}$

  **for** $m = 1, \ldots, M$ in parallel **do**

   $g_{\beta,m}^k = \begin{cases} \frac{1}{M} \nabla_\beta f_m(w_v^k, \beta_{v,m}^k) & \text{if } \zeta = 1 \\ \frac{1}{p_\beta M} \nabla_\beta f_{m,j_m}(w_y^k, \beta_{y,m}^k) + \frac{1}{M} \nabla_\beta f_m(w_v^k, \beta_{v,m}^k) & \text{if } \zeta = 2 \end{cases}$

   $\beta_{y,m}^{k+1} = \beta_{y,m}^k - \eta g_{\beta,m}^k$

   $\beta_{v,m}^{k+1} = \begin{cases} \beta_{y,m}^k, & \text{with probability } \rho \\ \beta_{v,m}^k, & \text{with probability } 1 - \rho \end{cases}$

  **end for**

 **end for**

**output** $w_y^K$, $\beta_{y,m}^K$ for $1 \le m \le M$.

---

## B   Technical Proofs

Throughout this section, we use $\boldsymbol{I}_{d'}$ to denote the $d' \times d'$ identity matrix, $0_{d'_1 \times d'_2}$ to denote the $d'_1 \times d'_2$ zero matrix, and $\boldsymbol{1}'_d \in \mathbb{R}^{d'}$ to denote the vector of ones.

### B.1   Proof of Lemma 1

To show the strong convexity, we shall verify the positive definiteness of

$$
\nabla^2 F_{MT2}(w, \beta) - \frac{\lambda}{2M} \boldsymbol{I}_{d(M+1)}
$$

$$
= \begin{pmatrix} \frac{\Lambda}{M} \nabla F'(w) + \frac{\lambda}{M} I_d & -\frac{\lambda}{M^{\frac{3}{2}}}(\boldsymbol{1}_M^\top \otimes I_d) \\ -\frac{\lambda}{M^{\frac{3}{2}}}(\boldsymbol{1}_M \otimes I_d) & \frac{\lambda}{M}(I_m \otimes I_d) + \mathrm{Diag}(\nabla^2 f'_1(\beta_1), \ldots, \nabla^2 f'_M(\beta_M)) \end{pmatrix} - \frac{\lambda}{2M} \boldsymbol{I}_{d(M+1)}
$$

$$
\succeq \begin{pmatrix} \left(\frac{\Lambda \mu'}{M} + \frac{\lambda}{2M}\right) I_d & -\frac{\lambda}{M^{\frac{3}{2}}}(\boldsymbol{1}_M^\top \otimes I_d) \\ -\frac{\lambda}{M^{\frac{3}{2}}}(\boldsymbol{1}_M \otimes I_d) & \left(\frac{\lambda}{2M} + \frac{\mu'}{M}\right)(I_m \otimes I_d) \end{pmatrix}
$$

$$
= \frac{1}{M} \underbrace{\begin{pmatrix} \Lambda \mu' + \frac{\lambda}{2} & -\frac{\lambda}{M^{\frac{1}{2}}} \boldsymbol{1}_M^\top \\ -\frac{\lambda}{M^{\frac{1}{2}}} \boldsymbol{1}_M & \left(\frac{\lambda}{2} + 2\mu'\right) I_m \end{pmatrix}}_{:=\boldsymbol{M}} \otimes I_d.
$$

Note that $\boldsymbol{M}$ can be written as a sum of $M$ matrices, each of them having

$$
\boldsymbol{M}_m = \begin{pmatrix} \frac{\Lambda \mu' + \frac{\lambda}{2}}{M} & -\frac{\lambda}{M^{\frac{1}{2}}} \\ -\frac{\lambda}{M^{\frac{1}{2}}} & \left(\frac{\lambda}{2} + 2\mu'\right) \end{pmatrix}
$$

as a $(m+1) \times (m+1)$ submatrix and zeros everywhere else. To verify positive semidefiniteness of $\boldsymbol{M}_m$, we shall prove that the determinant is positive:

$$
\det(\boldsymbol{M}_m) = \frac{1}{M}\left(\left(\Lambda \mu' + \frac{\lambda}{2}\right)\left(\frac{\lambda}{2} + 2\mu'\right) - \lambda^2\right) \geq \frac{1}{M}\left((2\lambda)\left(\frac{\lambda}{2} + 2\mu'\right) - \lambda^2\right) \geq 0
$$

as desired. Verifying the smoothness constants is straightforward.

### B.2   Proof of Lemma 2

We have

$$
\nabla^2 F_{MFL2}(w, \beta) - \frac{\mu'}{3M} \boldsymbol{I}_{d(M+1)}
$$

$$
= \begin{pmatrix} \frac{\lambda}{M} I_d & -\frac{\lambda}{M^{\frac{3}{2}}}(\boldsymbol{1}_M^\top \otimes I_d) \\ -\frac{\lambda}{M^{\frac{3}{2}}}(\boldsymbol{1}_M \otimes I_d) & \frac{\lambda}{M}(I_m \otimes I_d) + \mathrm{Diag}(\nabla^2 f'_1(\beta_1), \ldots, \nabla^2 f'_M(\beta_M)) \end{pmatrix} - \frac{\mu'}{3M} \boldsymbol{I}_{d(M+1)}
$$

$$
\succeq \begin{pmatrix} \left(\frac{\lambda}{M} - \frac{\mu'}{3M}\right) I_d & -\frac{\lambda}{M^{\frac{3}{2}}}(\boldsymbol{1}_M^\top \otimes I_d) \\ -\frac{\lambda}{M^{\frac{3}{2}}}(\boldsymbol{1}_M \otimes I_d) & \left(\frac{\lambda}{M} + \frac{2\mu'}{3M}\right)(I_m \otimes I_d) \end{pmatrix}
$$

$$
= \frac{1}{M} \underbrace{\begin{pmatrix} \lambda - \frac{\mu'}{3} & -\frac{\lambda}{M^{\frac{1}{2}}} \boldsymbol{1}_M^\top \\ -\frac{\lambda}{M^{\frac{1}{2}}} \boldsymbol{1}_M & \left(\lambda + \frac{2\mu'}{3}\right) I_m \end{pmatrix}}_{:=\boldsymbol{M}} \otimes I_d.
$$

Note that $\boldsymbol{M}$ can be written as a sum of $M$ matrices, each of them having $\frac{\lambda}{M} - \frac{\mu'}{3M}$ at the position $(1, 1)$, $-\frac{\lambda}{M^{\frac{1}{2}}}$ at positions $(1, m), (m, 1)$ and $\left(\frac{\lambda}{M} + \frac{2\mu'}{3M}\right)$ at the position $(m, m)$. Using the assumption $\mu' \leq \frac{\lambda}{2}$,

it is easy to see that each of these matrices is positive semidefinite, and thus so is $\boldsymbol{M}$. Consequently, $\nabla F_{MFL2}(w, \beta) - \frac{\mu'}{3M} \boldsymbol{I}_{d(M+1)}$ is positive semidefinite and thus $F_{MFL2}$ is jointly $\frac{\mu'}{3M}$- strongly convex. Verifying the smoothness constants is straightforward.

### B.3  Proof of Lemma 3

Let $x_m = (1 - \alpha_m)\beta_m + \alpha_m M^{-\frac{1}{2}}w$ for notational simplicity. We have

$$\nabla^2 f_m(w, \beta_m) = \begin{pmatrix} \frac{\Lambda}{M}\nabla^2 f'(M^{-\frac{1}{2}}w) + \frac{\alpha_m^2}{M}\nabla^2 f_m'(x_m) & \frac{\alpha_m(1-\alpha_m)}{M^{\frac{1}{2}}}\nabla^2 f_m'(x_m) \\ \frac{\alpha_m(1-\alpha_m)}{M^{\frac{1}{2}}}\nabla^2 f_m'(x_m) & (1-\alpha_m)^2\nabla^2 f_m'(x_m) \end{pmatrix}$$

$$= \begin{pmatrix} \frac{\Lambda}{M}\nabla^2 f'(M^{-\frac{1}{2}}w) & 0_{d\times d} \\ 0_{d\times d} & 0_{d\times d} \end{pmatrix} + \frac{1}{M}\begin{pmatrix} \frac{\alpha_m^2}{M} & \frac{\alpha_m(1-\alpha_m)}{M^{\frac{1}{2}}} \\ \frac{\alpha_m(1-\alpha_m)}{M^{\frac{1}{2}}} & (1-\alpha_m)^2 \end{pmatrix} \otimes \nabla^2 f_m'(x_m)$$

$$\succeq \begin{pmatrix} \frac{\Lambda\mu'}{M}\boldsymbol{I}_d & 0_{d\times d} \\ 0_{d\times d} & 0_{d\times d} \end{pmatrix} + \begin{pmatrix} \frac{\alpha_m^2}{M} & \frac{\alpha_m(1-\alpha_m)}{M^{\frac{1}{2}}} \\ \frac{\alpha_m(1-\alpha_m)}{M^{\frac{1}{2}}} & (1-\alpha_m)^2 \end{pmatrix} \otimes (\mu'\boldsymbol{I}_d)$$

$$= \mu' \underbrace{\begin{pmatrix} \frac{\Lambda+\alpha_m^2}{M} & \frac{\alpha_m(1-\alpha_m)}{M^{\frac{1}{2}}} \\ \frac{\alpha_m(1-\alpha_m)}{M^{\frac{1}{2}}} & (1-\alpha_m)^2 \end{pmatrix}}_{:=\boldsymbol{M}_m} \otimes \boldsymbol{I}_d.$$

Next, we show that

$$\boldsymbol{M}_m \succeq \begin{pmatrix} \frac{(1-\alpha_m)^2}{2M} & 0 \\ 0 & \frac{(1-\alpha_m)^2}{2} \end{pmatrix}. \tag{28}$$

For that, it suffices to show that

$$\det\left(\boldsymbol{M}_m - \begin{pmatrix} \frac{(1-\alpha_m)^2}{2M} & 0 \\ 0 & \frac{(1-\alpha_m)^2}{2} \end{pmatrix}\right) \geq 0,$$

which holds since

$$\det\left(\boldsymbol{M}_m - \begin{pmatrix} \frac{(1-\alpha_m)^2}{2M} & 0 \\ 0 & \frac{(1-\alpha_m)^2}{2} \end{pmatrix}\right) = \left(\frac{\Lambda + \alpha_m^2 - \frac{(1-\alpha_m)^2}{2}}{M}\right)\frac{(1-\alpha_m)^2}{2} - \frac{\alpha_m^2(1-\alpha_m)^2}{M}$$

$$\geq \left(2\frac{\alpha_m^2}{M}\right)\frac{(1-\alpha_m)^2}{2} - \frac{\alpha_m^2(1-\alpha_m)^2}{M} = 0.$$

Finally, using (28) $M$ times, it is easy to see that

$$\nabla^2 F_{APFL2}(w, \beta) \succeq \mu'\frac{(1-\alpha_{\max})^2}{M}\boldsymbol{I}_{d(M+1)}$$

as desired. Verifying the smoothness constants is straightforward.

### B.4  Proof of Lemma 5

We have

$$\frac{1}{M}\sum_{m=1}^{M}\|\nabla_w f_m(w_m^k, \beta_m^k)\|^2 \leq \frac{3}{M}\sum_{m=1}^{M}\|\nabla_w f_m(w_m^k, \beta_m^k) - \nabla_w f_m(w^k, \beta_m^k)\|^2$$

$$+ \frac{3}{M}\sum_{m=1}^{M}\|\nabla_w f_m(w^k, \beta_m^k) - \nabla_w f_m(w^*, \beta^*)\|^2$$

$$+ \frac{3}{M} \sum_{m=1}^{M} \|\nabla_w f_m(w^*, \beta^*)\|^2.$$

Then, using Assumption 1, the above display is bounded as

$$\frac{3L^2}{M} \sum_{m=1}^{M} \|w_m^k - w^k\|^2 + \frac{6L}{M} \sum_{m=1}^{M} D_{f_m}((w^k, \beta_m^k), (w^*, \beta^*)) + 3\zeta_*^2$$
$$= 6L^w \left( f(w^k, \beta_m^k) - f(w^*, \beta^*) \right) + 3(L^w)^2 V_k + 3\zeta_*^2,$$

which shows (21).

To establish (22), we have

$$\left\| \frac{1}{M} \sum_{m=1}^{M} \nabla_w f_m(w_m^k, \beta_m^k) \right\|^2 + \frac{1}{M^2} \sum_{m=1}^{M} \|\nabla_\beta f_m(w_m^k, \beta_m^k)\|^2$$
$$\leq \frac{2}{M} \sum_{i=1}^{M} \|\nabla_w f_m(w_m^k, \beta_m^k) - \nabla_w f_m(w^k, \beta_m^k)\|^2$$
$$+ \frac{2}{M} \sum_{m=1}^{M} \|\nabla_\beta f_m(w_m^k, \beta_m^k) - \nabla_\beta f_m(w^*, \beta^*)\|^2$$
$$+ \frac{2}{M^2} \sum_{m=1}^{M} \|\nabla_\beta f_m(w_m^k, \beta_m^k) - \nabla_\beta f_m(w^*, \beta^*)\|^2.$$

Then, using Assumption 1, the above display is bounded as

$$\frac{2(L^w)^2}{M} \sum_{m=1}^{M} \|w_m^k - w^k\|^2 + \frac{4L}{M} \sum_{m=1}^{M} D_{f_m}((w^k, \beta_m^k), (w^*, \beta^*)) = 4L \left( f(w^k, \beta_m^k) - f(w^*, \beta^*) \right) + 2(L^w)^2 V_k.$$

This completes the proof.

### B.5   Proof of Lemma 6

Let us start with establishing (23). We have

$$\frac{1}{M} \sum_{m=1}^{M} \mathbb{E}\|g_{w,m}^k\|^2 = \frac{1}{M} \sum_{m=1}^{M} \left( \mathbb{E}\|g_{w,m}^k - \nabla_w f_m(w_m^k, \beta_m^k)\|^2 + \|\nabla_w f_m(w_m^k, \beta_m^k)\|^2 \right)$$
$$\leq \frac{\sigma^2}{B} + \|\nabla_w f_m(w_m^k, \beta_m^k)\|^2.$$

Now (23) follows from an application of (21). Similarly, to show (24), we have

$$\mathbb{E}\left\| \frac{1}{M} \sum_{m=1}^{M} g_{w,m}^k \right\|^2 + \frac{1}{M^2} \sum_{m=1}^{M} \|g_{\beta,m}^k\|^2$$
$$= \mathbb{E}\left\| \frac{1}{M} \sum_{m=1}^{M} (g_{w,m}^k - \nabla_w f_m(w_m^k, \beta_m^k)) \right\|^2 + \left\| \frac{1}{M} \sum_{m=1}^{M} \nabla_w f_m(w_m^k, \beta_m^k) \right\|^2$$
$$+ \frac{1}{M^2} \sum_{m=1}^{M} \left( \mathbb{E}\|g_{\beta,m}^k - \nabla_\beta f_m(w_m^k, \beta_m^k)\|^2 + \|\nabla_\beta f_m(w_m^k, \beta_m^k)\|^2 \right)$$
$$\leq \frac{\sigma^2}{MB} + \left\| \frac{1}{M} \sum_{m=1}^{M} \nabla_w f_m(w_m^k, \beta_m^k) \right\|^2$$

$$+ \frac{\sigma^2}{MB} + \frac{1}{M^2} \sum_{m=1}^{M} \left\| \nabla_\beta f_m(w_m^k, \beta_m^k) \right\|^2.$$

Now (24) follows from (22), which completes the proof.

### B.6 Proof of Lemma 7

The proof is identical to the proof of Lemma E.1 from (Gorbunov et al., 2021) with a single difference – using inequality (23) instead of Assumption E.1 from (Gorbunov et al., 2021). We omit the details.

### B.7 Proof of Theorem 3 and Theorem 4

We start by introducing additional notation. We set $k_p = p \cdot \tau$, where $\tau \in \mathbb{N}^+$ is the length of the averaging period. Let $k_p = p\tau + \tau - 1 = k_{p+1} - 1 = v_p$. Denote the total number of iterations as $K$ and assume that $K = k_{\bar{p}}$ for some $\bar{p} \in \mathbb{N}^+$. The final result is set to be that $\hat{w} = w^K$ and $\hat{\beta}_m = \beta_m^K$ for all $m \in [M]$. We assume that the solution to (1) is $w^*, \beta_1^*, \ldots, \beta_M^*$ and that the optimal value is $f^*$. Let $w^k = \frac{1}{M} \sum_{m=1}^{M} w_m^k$ for all $k$. Note that this quantity will not be actually computed in practice unless $k = k_p$ for some $p \in \mathbb{N}$, where we have $w^{k_p} = w_m^{k_p}$ for all $m \in [M]$. In addition, let $\xi_m^k = \{\xi_{1,m}^k, \xi_{2,m}^k, \ldots, \xi_{B,m}^k\}$ and $\xi^k = \{\xi_1^k, \xi_2^k, \ldots, \xi_M^k\}$.

Let $\theta_m = ((w_m)^\top, (\beta_m)^\top)^\top$, $\theta_m^k = ((w_m^k)^\top, (\beta_m^k)^\top)^\top$, $\theta_m^* = ((w^*)^\top, (\beta_m^*)^\top)^\top$ and $\hat{\theta}_m^k = ((w^k)^\top, (\beta_m^k)^\top)^\top$. Let

$$g_m^k = \frac{1}{B} \nabla \hat{f}_m(w_m^k, \beta_m^k; \xi_m^k), \tag{29}$$

where

$$\nabla \hat{f}_m(w_m^k, \beta_m^k; \xi_m^k) = \sum_{j=1}^{B} \nabla \hat{f}_m(w_m^k, \beta_m^k; \xi_{j,m}^k).$$

We assume that the gradient is unbiased, that is,

$$\mathbb{E}\left[g_m^k\right] = \nabla f_m(w_m^k, \beta_m^k).$$

Let

$$g_{m,1}^k = \frac{1}{B} \nabla_w \hat{f}_m(w_m^k, \beta_m^k; \xi_m^k), \qquad g_{m,2}^k = \frac{1}{B} \nabla_{\beta_m} \hat{f}_m(w_m^k, \beta_m^k; \xi_m^k), \tag{30}$$

so that $g_m^k = ((g_{m,1}^k)^\top, (g_{m,2}^k)^\top)^\top$. We update the parameters by

$$(w_m^{k+1}, \beta_m^{k+1}) = (w_m^k, \beta_m^k) - \eta_k g_m^k.$$

In addition, we define

$$h^k = \frac{1}{M} \sum_{m=1}^{M} g_{m,1}^k, \qquad V^k = \frac{1}{M} \sum_{m=1}^{M} \|w_m^k - w^k\|^2.$$

Then $w^{k+1} = w^k - \eta_k h^k$ for all $k$.

We denote the Bregman divergence associated with $f_m$ for $\theta_m$ and $\bar{\theta}_m$ as

$$D_{f_m}(\theta_m, \bar{\theta}_m) := f_m(\theta_m) - f(\bar{\theta}_m) - \langle \nabla f_m(\bar{\theta}_m), \theta_m - \bar{\theta}_m \rangle.$$

Finally, we define the sum of residuals as

$$r^k = \|w^k - w^*\|^2 + \frac{1}{M} \sum_{m=1}^{M} \|\beta_m^k - \beta_m^*\|^2 = \frac{1}{M} \sum_{m=1}^{M} \|\hat{\theta}_m^k - \theta_m^*\|^2 \tag{31}$$

and let $\sigma_{\text{dif}}^2 = \frac{1}{M} \sum_{m=1}^{M} \|\nabla f_m(\theta_m^*)\|^2$.

The following proposition states some useful results that will be used in the proof below. The results are are standard and can be found in, for example, Nesterov (2018).

**Proposition 1.** *If the function $f$ is differentiable and $L$-smooth, then*

$$f(x) - f(y) - \langle \nabla f(y), x - y \rangle \leq \frac{L}{2} \|x - y\|^2. \tag{32}$$

*If $f$ is also convex, then*

$$\|\nabla f(x) - \nabla f(y)\|^2 \leq 2L D_f(x, y) \tag{33}$$

*for all $x, y$.*

*For all vectors $x, y$, we have*

$$2\langle x, y \rangle \leq \xi \|x\|^2 + \xi^{-1} \|y\|^2, \quad \forall \xi > 0, \tag{34}$$

$$-\langle x, y \rangle = -\frac{1}{2}\|x\|^2 - \frac{1}{2}\|y\|^2 + \frac{1}{2}\|x - y\|^2. \tag{35}$$

*For vectors $v_1, v_2, \ldots, v_n$, by the Jensen's inequality and the convexity of the map: $x \mapsto \|x\|^2$, we have*

$$\left\| \frac{1}{n} \sum_{i=1}^{n} v_i \right\|^2 \leq \frac{1}{n} \sum_{i=1}^{n} \|v_i\|^2. \tag{36}$$

Next, we establish a few technical results.

**Lemma 8.** *Suppose Assumption 4 holds. Given $\{\theta_m^k\}_{m \in [M]}$, we have*

$$\mathbb{E}_{\xi^k} \left[ \frac{1}{M} \sum_{m=1}^{M} f_m(\hat{\theta}_m^{k+1}) \right] - \frac{1}{M} \sum_{m=1}^{M} f_m(\hat{\theta}_m^k)$$

$$\leq -\eta_k \left\langle \frac{1}{M} \sum_{m=1}^{M} \nabla_w f_m(\hat{\theta}_m^k), \frac{1}{M} \sum_{m=1}^{M} \nabla_w f_m(\theta_m^k) \right\rangle$$

$$- \frac{\eta_k}{M} \sum_{m=1}^{M} \langle \nabla_{\beta_m} f_m(\hat{\theta}_m^k), \nabla_{\beta_m} f_m(\theta_m^k) \rangle$$

$$+ \frac{\eta_k^2 L}{2} \mathbb{E}_{\xi^k} \left[ \|h^k\|^2 \right] + \frac{\eta_k^2 L}{2M} \sum_{m=1}^{M} \mathbb{E}_{\xi_m^k} \left[ \|g_{m,2}^k\|^2 \right],$$

*where the expectation is taken only with respect to the randomness in $\xi^k$.*

*Proof.* By the $L$-smoothness assumption on $f_m(\cdot)$ and (32), we have

$$f_m(\hat{\theta}_m^{k+1}) - f_m(\hat{\theta}_m^k) - \langle \nabla f_m(\hat{\theta}_m^k), \hat{\theta}_m^{k+1} - \hat{\theta}_m^k \rangle \leq \frac{L}{2} \|\hat{\theta}_m^{k+1} - \hat{\theta}_m^k\|^2.$$

Thus, we have

$$f_m(\hat{\theta}_m^{k+1}) - f_m(\hat{\theta}_m^k) \leq -\eta_k \langle \nabla_w f_m(\hat{\theta}_m^k), h^k \rangle - \eta_k \langle \nabla_{\beta_m} f_m(\hat{\theta}_m^k), g_{m,2}^k \rangle + \frac{\eta_k^2 L}{2} \|h^k\|^2 + \frac{\eta_k^2 L}{2} \|g_{m,2}^k\|^2,$$

which further implies that

$$\frac{1}{M} \sum_{m=1}^{M} f_m(\hat{\theta}_m^{k+1}) - \frac{1}{M} \sum_{m=1}^{M} f_m(\hat{\theta}_m^k)$$

$$\leq -\eta_k \left\langle \frac{1}{M} \sum_{m=1}^{M} \nabla_w f_m(\hat{\theta}_m^k), h^k \right\rangle - \frac{\eta_k}{M} \sum_{m=1}^{M} \langle \nabla_{\beta_m} f_m(\hat{\theta}_m^k), g_{m,2}^k \rangle + \frac{\eta_k^2 L}{2} \|h^k\|^2$$

$$+ \frac{\eta_k^2 L}{2M} \sum_{m=1}^{M} \|g_{m,2}^k\|^2.$$

The result follows by taking the expectation with respect to the randomness in $\xi^k$, while keeping the other quantities fixed. $\qquad\square$

**Lemma 9.** *Suppose Assumptions 5 and 6 hold. Given $\{\theta_m^k\}_{m \in [M]}$, we have*

$$\mathbb{E}_{\xi^k}\left[\|h^k\|^2\right] + \frac{1}{M}\sum_{m=1}^{M}\mathbb{E}_{\xi_m^k}\left[\|g_{m,2}^k\|^2\right]$$

$$\leq \left(\frac{C_1}{M} + C_2 + 1\right)\frac{1}{M}\sum_{m=1}^{M}\|\nabla f_m(\theta_m^k)\|^2 + \frac{\sigma_1^2}{MB} + \frac{\sigma_2^2}{B}$$

$$\leq \lambda\left(\frac{C_1}{M} + C_2 + 1\right)\left\|\frac{1}{M}\sum_{m=1}^{M}\nabla f_m(\theta_m^k)\right\|^2 + \left(\frac{C_1}{M} + C_2 + 1\right)\sigma_{dif}^2 + \frac{\sigma_1^2}{MB} + \frac{\sigma_2^2}{B},$$

*where the expectation is taken only with respect to the randomness in $\xi^k$.*

*Proof.* Note that

$$\mathbb{E}_{\xi^k}\left[\|h^k\|^2\right] = \mathbb{E}_{\xi^k}\left[\left\|\frac{1}{M}\sum_{m=1}^{M}g_{m,1}^k\right\|^2\right]$$

$$\stackrel{(i)}{=} \mathbb{E}_{\xi^k}\left[\left\|\frac{1}{M}\sum_{m=1}^{M}\left(g_{m,1}^k - \nabla_w f_m(\theta_m^k)\right)\right\|^2\right] + \left\|\frac{1}{M}\sum_{m=1}^{M}\nabla_w f_m(\theta_m^k)\right\|^2$$

$$\stackrel{(ii)}{=} \frac{1}{M^2}\sum_{m=1}^{M}\mathbb{E}_{\xi_m^k}\left[\|g_{m,1}^k - \nabla_w f_m(\theta_m^k)\|^2\right] + \left\|\frac{1}{M}\sum_{m=1}^{M}\nabla_w f_m(\theta_m^k)\right\|^2$$

$$\stackrel{(iii)}{\leq} \frac{1}{M^2}\sum_{m=1}^{M}\left(C_1\|\nabla f_m(\theta_m^k)\|^2 + \frac{\sigma_1^2}{B}\right) + \left\|\frac{1}{M}\sum_{m=1}^{M}\nabla_w f_m(\theta_m^k)\right\|^2$$

$$\stackrel{(iv)}{\leq} \frac{C_1}{M^2}\sum_{m=1}^{M}\|\nabla f_m(\theta_m^k)\|^2 + \frac{\sigma_1^2}{MB} + \frac{1}{M}\sum_{m=1}^{M}\|\nabla_w f_m(\theta_m^k)\|^2,$$

where (i) is due to $g_{m,1}^k$ being unbiased, (ii) is by the fact that $\xi_1^k, \xi_2^k, \ldots, \xi_M^k$ are independent, (iii) is by Assumption 5, and (iv) is by (36). Similarly, we have

$$\frac{1}{M}\sum_{m=1}^{M}\mathbb{E}_{\xi_m^k}\left[\|g_{m,2}^k\|^2\right] = \frac{1}{M}\sum_{m=1}^{M}\mathbb{E}_{\xi_m^k}\left[\|g_{m,2}^k - \nabla_{\beta_m}f_m(\theta_m^k)\|^2\right] + \frac{1}{M}\sum_{m=1}^{M}\|\nabla_{\beta_m}f_m(\theta_m^k)\|^2$$

$$\leq \frac{C_2}{M}\sum_{m=1}^{M}\|\nabla f_m(\theta_m^k)\|^2 + \frac{\sigma_2^2}{B} + \frac{1}{M}\sum_{m=1}^{M}\|\nabla_{\beta_m}f_m(\theta_m^k)\|^2.$$

The lemma follows by combining the two inequalities. $\qquad\square$

**Lemma 10.** *Under Assumption 4, we have*

$$-\eta_k\left\langle\frac{1}{M}\sum_{m=1}^{M}\nabla_w f_m(\hat{\theta}_m^k), \frac{1}{M}\sum_{m=1}^{M}\nabla_w f_m(\theta_m^k)\right\rangle - \frac{\eta_k}{M}\sum_{m=1}^{M}\langle\nabla_{\beta_m}f_m(\hat{\theta}^k), \nabla_{\beta_m}f_m(\theta_m^k)\rangle$$

$$\leq -\frac{\eta_k}{2}\left\|\frac{1}{M}\sum_{m=1}^{M}\nabla f_m(\hat{\theta}_m^k)\right\|^2 - \frac{\eta_k}{2}\left\|\frac{1}{M}\sum_{m=1}^{M}\nabla f_m(\theta_m^k)\right\|^2 + \frac{\eta_k L^2}{2}V^k.$$

*Proof.* By (35), we have

$$-\eta_k\left\langle\frac{1}{M}\sum_{m=1}^{M}\nabla_w f_m(\hat{\theta}_m^k), \frac{1}{M}\sum_{m=1}^{M}\nabla_w f_m(\theta_m^k)\right\rangle$$

$$= -\frac{\eta_k}{2}\left\|\frac{1}{M}\sum_{m=1}^{M}\nabla_w f_m(\hat{\theta}_m^k)\right\|^2 - \frac{\eta_k}{2}\left\|\frac{1}{M}\sum_{m=1}^{M}\nabla_w f_m(\theta_m^k)\right\|^2$$

$$+ \frac{\eta_k}{2}\left\|\frac{1}{M}\sum_{m=1}^{M}\left(\nabla_w f_m(\hat{\theta}_m^k) - \nabla_w f_m(\theta_m^k)\right)\right\|^2$$

$$\leq -\frac{\eta_k}{2}\left\|\frac{1}{M}\sum_{m=1}^{M}\nabla_w f_m(\hat{\theta}_m^k)\right\|^2 - \frac{\eta_k}{2}\left\|\frac{1}{M}\sum_{m=1}^{M}\nabla_w f_m(\theta_m^k)\right\|^2$$

$$+ \frac{\eta_k}{2M}\sum_{m=1}^{M}\left\|\nabla_w f_m(\hat{\theta}_m^k) - \nabla_w f_m(\theta_m^k)\right\|^2,$$

where the last inequality follows from (36). We also have

$$-\eta_k\langle\nabla_{\beta_m}f_m(\hat{\theta}^k),\nabla_{\beta_m}f_m(\theta_m^k)\rangle$$
$$= -\frac{\eta_k}{2}\|\nabla_{\beta_m}f_m(\hat{\theta}_m^k)\|^2 - \frac{\eta_k}{2}\|\nabla_{\beta_m}f_m(\theta_m^k)\|^2 + \frac{\eta_k}{2}\|\nabla_{\beta_m}f_m(\hat{\theta}_m^k) - \nabla_{\beta_m}f_m(\theta_m^k)\|^2.$$

Thus,

$$-\frac{\eta_k}{M}\langle\nabla_{\beta_m}f_m(\hat{\theta}^k),\nabla_{\beta_m}f_m(\theta_m^k)\rangle = -\frac{\eta_k}{2M}\sum_{m=1}^{M}\|\nabla_{\beta_m}f_m(\hat{\theta}_m^k)\|^2 - \frac{\eta_k}{2M}\sum_{m=1}^{M}\|\nabla_{\beta_m}f_m(\theta_m^k)\|^2$$

$$+ \frac{\eta_k}{2M}\sum_{m=1}^{M}\|\nabla_{\beta_m}f_m(\hat{\theta}_m^k) - \nabla_{\beta_m}f_m(\theta_m^k)\|^2$$

$$\leq -\frac{\eta_k}{2}\left\|\frac{1}{M}\sum_{m=1}^{M}\nabla_{\beta_m}f_m(\hat{\theta}_m^k)\right\|^2 - \frac{\eta_k}{2}\left\|\frac{1}{M}\sum_{m=1}^{M}\nabla_{\beta_m}f_m(\theta_m^k)\right\|^2$$

$$+ \frac{\eta_k}{2M}\sum_{m=1}^{M}\|\nabla_{\beta_m}f_m(\hat{\theta}_m^k) - \nabla_{\beta_m}f_m(\theta_m^k)\|^2.$$

Combining the above equations, we have

$$-\eta_k\left\langle\frac{1}{M}\sum_{m=1}^{M}\nabla_w f_m(\hat{\theta}_m^k),\frac{1}{M}\sum_{m=1}^{M}\nabla_w f_m(\theta_m^k)\right\rangle - \frac{\eta_k}{M}\sum_{m=1}^{M}\langle\nabla_{\beta_m}f_m(\hat{\theta}_m^k),\nabla_{\beta_m}f_m(\theta_m^k)\rangle$$

$$\leq -\frac{\eta_k}{2}\left\|\frac{1}{M}\sum_{m=1}^{M}\nabla f_m(\hat{\theta}_m^k)\right\|^2 - \frac{\eta_k}{2}\left\|\frac{1}{M}\sum_{m=1}^{M}\nabla f_m(\theta_m^k)\right\|^2$$

$$+ \frac{\eta_k}{2M}\sum_{m=1}^{M}\left\|\nabla f_m(\hat{\theta}_m^k) - \nabla f_m(\theta_m^k)\right\|^2$$

$$\overset{(i)}{\leq} -\frac{\eta_k}{2}\left\|\frac{1}{M}\sum_{m=1}^{M}\nabla f_m(\hat{\theta}_m^k)\right\|^2 - \frac{\eta_k}{2}\left\|\frac{1}{M}\sum_{m=1}^{M}\nabla f_m(\theta_m^k)\right\|^2 + \frac{\eta_k L^2}{2M}\sum_{m=1}^{M}\left\|w_m^k - w^k\right\|^2$$

$$= -\frac{\eta_k}{2}\left\|\frac{1}{M}\sum_{m=1}^{M}\nabla f_m(\hat{\theta}_m^k)\right\|^2 - \frac{\eta_k}{2}\left\|\frac{1}{M}\sum_{m=1}^{M}\nabla f_m(\theta_m^k)\right\|^2 + \frac{\eta_k L^2}{2}V^k,$$

where $(i)$ is by Assumption 4. $\qquad\square$

**Lemma 11.** *Under Assumptions 4 and 7, we have*

$$-\eta_k\left\langle\frac{1}{M}\sum_{m=1}^{M}\nabla_w f_m(\hat{\theta}_m^k),\frac{1}{M}\sum_{m=1}^{M}\nabla_w f_m(\theta_m^k)\right\rangle - \frac{\eta_k}{M}\sum_{m=1}^{M}\langle\nabla_{\beta_m}f_m(\hat{\theta}^k),\nabla_{\beta_m}f_m(\theta_m^k)\rangle$$

$$\leq -\eta_k \mu \left( \frac{1}{M} \sum_{m=1}^{M} \nabla f_m(\hat{\theta}_m^k) - f^* \right) - \frac{\eta_k}{2} \left\| \frac{1}{M} \sum_{m=1}^{M} \nabla f_m(\theta_m^k) \right\|^2 + \frac{\eta_k L^2}{2} V^k.$$

*Proof.* The proof follows directly from Lemma 10 and Assumption 7. □

**Lemma 12.** *Suppose Assumptions 5 and 6 hold. For $k_p + 1 \leq k \leq v_p$, we have*

$$\mathbb{E}\left[V^k\right] \leq \lambda(\tau-1)(C_1+1) \sum_{t=k_p}^{k-1} \eta_t^2 \mathbb{E}\left[\left\| \frac{1}{M} \sum_{m=1}^{M} \nabla f_m(\theta_m^t) \right\|^2\right]$$

$$+ \sigma_{dif}^2(\tau-1)(C_1+1) \sum_{t=k_p}^{k-1} \eta_t^2 + \frac{\sigma_1^2(\tau-1)}{B} \sum_{t=k_p}^{k-1} \eta_t^2.$$

*Note that $V^{k_p} = 0$.*

*Proof.* Note that $w^{k_p} = w_m^{k_p}$ for all $m \in [M]$. Thus, for $k_p + 1 \leq k \leq v_p$, we have

$$\|w_m^k - w^k\|^2 = \left\| w_m^{k_p} - \sum_{t=k_p}^{k-1} \eta_t g_{m,1}^t - w^{k_p} - \sum_{t=k_p}^{k-1} \eta_t h^t \right\|^2 = \left\| \sum_{t=k_p}^{k-1} \eta_t g_{m,1}^t - \sum_{t=k_p}^{k-1} \eta_t h^t \right\|^2.$$

Since

$$\frac{1}{M} \sum_{m=1}^{M} \sum_{t=k_p}^{k-1} \eta_t g_{m,1}^t = \sum_{t=k_p}^{k-1} \eta_t h^t,$$

we have

$$\frac{1}{M} \sum_{m=1}^{M} \|w_m^k - w^k\|^2 = \frac{1}{M} \sum_{m=1}^{M} \left\| \sum_{t=k_p}^{k-1} \eta_t g_{m,1}^t - \sum_{t=k_p}^{k-1} \eta_t h^t \right\|^2$$

$$= \frac{1}{M} \sum_{m=1}^{M} \left\| \sum_{t=k_p}^{k-1} \eta_t g_{m,1}^t \right\|^2 - \left\| \sum_{t=k_p}^{k-1} \eta_t h^t \right\|^2 \leq \frac{1}{M} \sum_{m=1}^{M} \left\| \sum_{t=k_p}^{k-1} \eta_t g_{m,1}^t \right\|^2 \qquad (37)$$

$$\leq \frac{k-k_p}{M} \sum_{m=1}^{M} \sum_{t=k_p}^{k-1} \eta_t^2 \|g_{m,1}^t\|^2 \leq \frac{\tau-1}{M} \sum_{m=1}^{M} \sum_{t=k_p}^{k-1} \eta_t^2 \|g_{m,1}^t\|^2.$$

Given $\{\theta_m^k\}_{m \in [M]}$, we have

$$\mathbb{E}_{\xi^k}\left[ \frac{1}{M} \sum_{m=1}^{M} \|g_{m,1}^k\|^2 \right] = \frac{1}{M} \sum_{m=1}^{M} \mathbb{E}_{\xi_m^k}\left[ \|g_{m,1}^k\|^2 \right]$$

$$= \frac{1}{M} \sum_{m=1}^{M} \mathbb{E}_{\xi_m^k}\left[ \|g_{m,1}^k - \nabla_w f_m(\theta_m^k)\|^2 \right] + \frac{1}{M} \sum_{m=1}^{M} \|\nabla_w f_m(\theta_m^k)\|^2$$

$$\leq \frac{1}{M} \sum_{m=1}^{M} \left[ (C_1+1)\nabla \|f_m(\theta_m^k)\|^2 + \frac{\sigma_1^2}{B} \right] + \frac{1}{M} \sum_{m=1}^{M} \|\nabla f_m(\theta_m^k)\|^2$$

$$= \frac{C_1+1}{M} \sum_{m=1}^{M} \|\nabla f_m(\theta_m^k)\|^2 + \frac{\sigma_1^2}{B},$$

where the expectation is taken with respect to the randomness in $\xi^k$. Thus, by the independence of $\xi^{(1)}, \xi^{(2)}, \ldots, \xi^k$ and taking an unconditional expectation on both sides of (37), we have

$$
\begin{aligned}
\mathbb{E}\left[V^k\right] &= (\tau - 1) \sum_{t=k_p}^{k-1} \eta_t^2 \mathbb{E}\left[\mathbb{E}_{\xi^t}\left[\frac{1}{M} \sum_{m=1}^M \|g_{m,1}^t\|^2\right]\right] \\
&\leq (\tau - 1)(C_1 + 1) \sum_{t=k_p}^{k-1} \eta_t^2 \mathbb{E}\left[\frac{1}{M} \sum_{m=1}^M \|\nabla f_m(\theta_m^t)\|^2\right] + \frac{(\tau - 1)\sigma_1^2}{B} \sum_{t=k_p}^{k-1} \eta_t^2 \\
&\leq \lambda(\tau - 1)(C_1 + 1) \sum_{t=k_p}^{k-1} \eta_t^2 \mathbb{E}\left[\left\|\frac{1}{M} \sum_{m=1}^M \nabla f_m(\theta_m^t)\right\|^2\right] \\
&\quad + \sigma_{\text{dif}}^2(\tau - 1)(C_1 + 1) \sum_{t=k_p}^{k-1} \eta_t^2 + \frac{(\tau - 1)\sigma_1^2}{B} \sum_{t=k_p}^{k-1} \eta_t^2,
\end{aligned}
$$

where the last inequality follows Assumption 6. $\qquad\square$

With these preliminaries, we are ready to prove Theorem 3 and Theorem 4.

### B.7.1 Proof of Theorem 3

Under Assumptions 4-6, given $\{\theta_m^k\}_{m \in [M]}$, it follows from Lemmas 8-10 that

$$
\begin{aligned}
\mathbb{E}_{\xi^k}&\left[\frac{1}{M} \sum_{m=1}^M f_m(\hat{\theta}_m^{k+1})\right] - \frac{1}{M} \sum_{m=1}^M f_m(\hat{\theta}_m^k) \\
&\leq -\frac{\eta}{2}\left\|\frac{1}{M} \sum_{m=1}^M f_m(\hat{\theta}_m^k)\right\|^2 - \frac{\eta}{2}\left\|\frac{1}{M} \sum_{m=1}^M f_m(\theta_m^k)\right\|^2 + \frac{\eta L^2}{2} V^k \\
&\quad + \frac{1}{2}\eta^2 L\lambda\left(\frac{C_1}{M} + C_2 + 1\right)\left\|\frac{1}{M} \sum_{m=1}^M f_m(\theta_m^k)\right\|^2 \\
&\quad + \frac{1}{2}\eta^2 L\lambda\left\{\left(\frac{C_1}{M} + C_2 + 1\right)\sigma_{\text{dif}}^2 + \frac{\sigma_1^2}{MB} + \frac{\sigma_2^2}{B}\right\},
\end{aligned}
$$

where the expectation is taken with respect to the randomness in $\xi^k$. Thus, taking the unconditional expectation on both sides of the equation above, we have

$$
\begin{aligned}
\mathbb{E}&\left[\frac{1}{M} \sum_{m=1}^M f_m(\hat{\theta}_m^{k+1}) - \frac{1}{M} \sum_{m=1}^M f_m(\hat{\theta}_m^k)\right] \\
&\leq -\frac{\eta}{2}\mathbb{E}\left[\left\|\frac{1}{M} \sum_{m=1}^M f_m(\hat{\theta}_m^k)\right\|^2\right] - \frac{\eta}{2}\mathbb{E}\left[\left\|\frac{1}{M} \sum_{m=1}^M f_m(\theta_m^k)\right\|^2\right] + \frac{\eta L^2}{2}\mathbb{E}\left[V^k\right] \\
&\quad + \frac{1}{2}\eta^2 L\lambda\left(\frac{C_1}{M} + C_2 + 1\right)\mathbb{E}\left[\left\|\frac{1}{M} \sum_{m=1}^M f_m(\theta_m^k)\right\|^2\right] \\
&\quad + \frac{1}{2}\eta^2 L\lambda\left\{\left(\frac{C_1}{M} + C_2 + 1\right)\sigma_{\text{dif}}^2 + \frac{\sigma_1^2}{MB} + \frac{\sigma_2^2}{B}\right\},
\end{aligned}
$$

which implies that

$$\mathbb{E}\left[\frac{1}{M}\sum_{m=1}^{M}f_m(\hat{\theta}_m^{k_{p+1}})-\frac{1}{M}\sum_{m=1}^{M}f_m(\hat{\theta}_m^{k_p})\right]$$

$$=\sum_{k=k_p}^{v_p}\mathbb{E}\left[\frac{1}{M}\sum_{m=1}^{M}f_m(\hat{\theta}_m^{k+1})-\frac{1}{M}\sum_{m=1}^{M}f_m(\hat{\theta}_m^{k})\right]$$

$$\leq -\frac{\eta}{2}\sum_{k=k_p}^{v_p}\mathbb{E}\left[\left\|\frac{1}{M}\sum_{m=1}^{M}f_m(\hat{\theta}_m^{k})\right\|^2\right] \tag{38}$$

$$+\frac{\eta}{2}\left\{-1+\eta L\lambda\left(\frac{C_1}{M}+C_2+1\right)\right\}\sum_{k=k_p}^{v_p}\mathbb{E}\left[\left\|\frac{1}{M}\sum_{m=1}^{M}f_m(\theta_m^{k})\right\|^2\right]$$

$$+\frac{\eta L^2}{2}\sum_{k=k_p}^{v_p}\mathbb{E}\left[V^k\right]+\frac{1}{2}\eta^2 L\lambda\left\{\left(\frac{C_1}{M}+C_2+1\right)\sigma_{\text{dif}}^2+\frac{\sigma_1^2}{MB}+\frac{\sigma_2^2}{B}\right\}\sum_{k=k_p}^{v_p}1.$$

By Lemma 12, for all $k_p \leq k \leq v_p$, we have that

$$\mathbb{E}\left[V^k\right]\leq \lambda\eta^2(\tau-1)(C_1+1)\sum_{k=k_p}^{k-1}\mathbb{E}\left[\left\|\frac{1}{M}\sum_{m=1}^{M}\nabla f_m(\theta_m^{k})\right\|^2\right]$$

$$+\eta^2\sigma_{\text{dif}}^2(\tau-1)(C_1+1)(k-k_p)+\frac{\eta^2\sigma_1^2(\tau-1)}{B}(k-k_p)$$

$$\leq \lambda\eta^2(\tau-1)(C_1+1)\sum_{k=k_p}^{v_p}\mathbb{E}\left[\left\|\frac{1}{M}\sum_{m=1}^{M}\nabla f_m(\theta_m^{k})\right\|^2\right]$$

$$+\eta^2\sigma_{\text{dif}}^2(\tau-1)^2(C_1+1)+\frac{\eta^2\sigma_1^2(\tau-1)^2}{B}.$$

Therefore, we have

$$\frac{\eta L^2}{2}\sum_{k=k_p}^{v_p}\mathbb{E}\left[V^k\right]\leq \frac{1}{2}\lambda\eta^3 L^2(\tau-1)\tau(C_1+1)\sum_{k=k_p}^{v_p}\mathbb{E}\left[\left\|\frac{1}{M}\sum_{m=1}^{M}\nabla f_m(\theta_m^{k})\right\|^2\right]$$

$$+\frac{1}{2}\eta^3 L^2\sigma_{\text{dif}}^2(\tau-1)^2(C_1+1)\sum_{k=k_p}^{v_p}1+\frac{\eta^3 L^2\sigma_1^2(\tau-1)^2}{2B}\sum_{k=k_p}^{v_p}1.$$

Combined with (38), we have

$$\mathbb{E}\left[\frac{1}{M}\sum_{m=1}^{M}f_m(\hat{\theta}_m^{k_{p+1}})-\frac{1}{M}\sum_{m=1}^{M}f_m(\hat{\theta}_m^{k_p})\right]$$

$$\leq -\frac{\eta}{2}\sum_{k=k_p}^{v_p}\mathbb{E}\left[\left\|\frac{1}{M}\sum_{m=1}^{M}f_m(\hat{\theta}_m^{k})\right\|^2\right]$$

$$+\frac{\eta}{2}\left\{-1+\eta L\lambda\left(\frac{C_1}{M}+C_2+1\right)+\lambda\eta^2 L^2(\tau-1)\tau(C_1+1)\right\}\sum_{k=k_p}^{v_p}\mathbb{E}\left[\left\|\frac{1}{M}\sum_{m=1}^{M}f_m(\theta_m^{k})\right\|^2\right]$$

$$+\frac{1}{2}\eta^2 L\lambda\left\{\left(\frac{C_1}{M}+C_2+1\right)\sigma_{\text{dif}}^2+\frac{\sigma_1^2}{MB}+\frac{\sigma_2^2}{B}\right\}\sum_{k=k_p}^{v_p}1$$

$$+\frac{1}{2}\eta^3 L^2\sigma_{\text{dif}}^2(\tau-1)^2(C_1+1)\sum_{k=k_p}^{v_p}1+\frac{\eta^3 L^2\sigma_1^2(\tau-1)^2}{2B}\sum_{k=k_p}^{v_p}1.$$

Since we require that

$$-1 + \eta L\lambda \left(\frac{C_1}{M} + C_2 + 1\right) + \eta^2 L^2(\tau-1)\tau(C_1+1) \le 0,$$

the equation above implies that

$$\mathbb{E}\left[\frac{1}{M}\sum_{m=1}^{M} f_m(\hat{\theta}_m^{k_{p+1}}) - \frac{1}{M}\sum_{m=1}^{M} f_m(\hat{\theta}_m^{k_p})\right]$$

$$\le -\frac{\eta}{2}\sum_{k=k_p}^{v_p}\mathbb{E}\left[\left\|\frac{1}{M}\sum_{m=1}^{M} f_m(\hat{\theta}_m^k)\right\|^2\right] + \frac{1}{2}\eta^2 L\lambda\left\{\left(\frac{C_1}{M} + C_2 + 1\right)\sigma_{\text{dif}}^2 + \frac{\sigma_1^2}{MB} + \frac{\sigma_2^2}{B}\right\}\sum_{k=k_p}^{v_p} 1$$

$$+ \frac{1}{2}\eta^3 L^2\sigma_{\text{dif}}^2(\tau-1)^2(C_1+1)\sum_{k=k_p}^{v_p} 1 + \frac{\eta^3 L^2\sigma_1^2(\tau-1)^2}{2B}\sum_{k=k_p}^{v_p} 1.$$

Since we have assumed that $K = k_{\bar{p}}$ for some $\bar{p} \in \mathbb{N}^+$, we further have

$$\frac{1}{K}\mathbb{E}\left[\left(\frac{1}{M}\sum_{m=1}^{M} f_m(\hat{\theta}_m^K) - f^*\right) - \left(\frac{1}{M}\sum_{m=1}^{M} f_m(\hat{\theta}_m^0) - f^*\right)\right]$$

$$= \frac{1}{K}\mathbb{E}\left[\frac{1}{M}\sum_{m=1}^{M} f_m(\hat{\theta}_m^K) - \frac{1}{M}\sum_{m=1}^{M} f_m(\hat{\theta}_m^0)\right]$$

$$= \frac{1}{K}\sum_{p=0}^{\bar{p}-1}\mathbb{E}\left[\frac{1}{M}\sum_{m=1}^{M} f_m(\hat{\theta}_m^{k_{p+1}}) - \frac{1}{M}\sum_{m=1}^{M} f_m(\hat{\theta}_m^{k_p})\right]$$

$$\le -\frac{\eta}{2K}\sum_{p=0}^{\bar{p}-1}\sum_{k=k_p}^{v_p}\mathbb{E}\left[\left\|\frac{1}{M}\sum_{m=1}^{M} f_m(\hat{\theta}_m^k)\right\|^2\right]$$

$$+ \frac{1}{2}\eta^2 L\lambda\left\{\left(\frac{C_1}{M} + C_2 + 1\right)\sigma_{\text{dif}}^2 + \frac{\sigma_1^2}{MB} + \frac{\sigma_2^2}{B}\right\}\frac{1}{K}\sum_{p=0}^{\bar{p}-1}\sum_{k=k_p}^{v_p} 1$$

$$+ \frac{1}{2}\eta^3 L^2\sigma_{\text{dif}}^2(\tau-1)^2(C_1+1)\frac{1}{K}\sum_{p=0}^{\bar{p}-1}\sum_{k=k_p}^{v_p} 1 + \frac{\eta^3 L^2\sigma_1^2(\tau-1)^2}{2B}\frac{1}{K}\sum_{p=0}^{\bar{p}-1}\sum_{k=k_p}^{v_p} 1$$

$$= -\frac{\eta}{2K}\sum_{k=0}^{K-1}\mathbb{E}\left[\left\|\frac{1}{M}\sum_{m=1}^{M} f_m(\hat{\theta}_m^k)\right\|^2\right] + \frac{1}{2}\eta^2 L\lambda\left\{\left(\frac{C_1}{M} + C_2 + 1\right)\sigma_{\text{dif}}^2 + \frac{\sigma_1^2}{MB} + \frac{\sigma_2^2}{B}\right\}$$

$$+ \frac{1}{2}\eta^3 L^2\sigma_{\text{dif}}^2(\tau-1)^2(C_1+1) + \frac{\eta^3 L^2\sigma_1^2(\tau-1)^2}{2B}.$$

This implies that

$$\frac{1}{K}\sum_{k=0}^{K-1}\mathbb{E}\left[\left\|\frac{1}{M}\sum_{m=1}^{M} f_m(\hat{\theta}_m^k)\right\|^2\right]$$

$$\le \frac{2\mathbb{E}\left[\frac{1}{M}\sum_{m=1}^{M} f_m(\hat{\theta}_m^0) - f^*\right]}{\eta K} + \eta L\lambda\left\{\left(\frac{C_1}{M} + C_2 + 1\right)\sigma_{\text{dif}}^2 + \frac{\sigma_1^2}{MB} + \frac{\sigma_2^2}{B}\right\}$$

$$+ \eta^2 L^2\sigma_{\text{dif}}^2(\tau-1)^2(C_1+1) + \frac{\eta^2 L^2\sigma_1^2(\tau-1)^2}{B}$$

and the proof is complete.

### B.7.2 Proof of Theorem 4

By Lemmas 8, 9, 11 and 12, for $k_p + 1 \le k \le v_p$, we have

$$
\mathbb{E}\left[ \frac{1}{M} \sum_{m=1}^{M} f_m(\hat{\theta}_m^{k+1}) - f^* \right]
$$

$$
\le \Delta_k \mathbb{E}\left[ \frac{1}{M} \sum_{m=1}^{M} f_m(\hat{\theta}_m^k) - f^* \right]
$$

$$
+ \frac{\eta_k}{2} \left\{ -1 + \eta_k \lambda L \left( \frac{C_1}{M} + C_2 + 1 \right) \right\} \mathbb{E}\left[ \left\| \frac{1}{M} \sum_{m=1}^{M} \nabla f_m(\theta_m^k) \right\|^2 \right]
$$

$$
+ B_k \sum_{t=k_p}^{k-1} \eta_t^2 \mathbb{E}\left[ \left\| \frac{1}{M} \sum_{m=1}^{M} \nabla f_m(\theta_m^{(t)}) \right\|^2 \right] + c_k,
$$

where

$$
\Delta_k = 1 - \eta_k \mu,
$$

$$
B_k = \frac{1}{2} \eta_k L^2 \lambda (\tau - 1)(C_1 + 1), \text{ and}
$$

$$
c_k = \frac{\eta_k L^2}{2} \left\{ \sigma_{\text{dif}}^2 (\tau - 1)(C_1 + 1) \sum_{t=k_p}^{k-1} \eta_t^2 + \frac{\sigma_1^2(\tau - 1)}{B} \sum_{t=k_p}^{k-1} \eta_t^2 \right\} \tag{39}
$$

$$
+ \frac{\eta_k^2 L}{2} \left\{ \sigma_{\text{dif}}^2 \left( \frac{C_1}{M} + C_2 + 1 \right) + \frac{\sigma_1^2}{MB} + \frac{\sigma_2^2}{B} \right\}.
$$

Let

$$
a_k = \mathbb{E}\left[ \frac{1}{M} \sum_{m=1}^{M} f_m(\hat{\theta}_m^k) - f^* \right],
$$

$$
D = \lambda L \left( \frac{C_1}{M} + C_2 + 1 \right), \text{ and}
$$

$$
e_k = \mathbb{E}\left[ \left\| \frac{1}{M} \sum_{m=1}^{M} \nabla f_m(\theta_m^k) \right\|^2 \right],
$$

and denote

$$
\sum_{k=k_p}^{k_p-1} \eta_k^2 e_t = 0,
$$

$$
c_{k_p} = \frac{\alpha_{k_p}^2 L}{2} \left\{ \sigma_{\text{dif}}^2 \left( \frac{C_1}{M} + C_2 + 1 \right) + \frac{\sigma_1^2}{MB} + \frac{\sigma_2^2}{B} \right\}.
$$

Then

$$
a_{k+1} \le \Delta_k a_k + \frac{\eta_k}{2}(-1 + D\eta_k) e_k + B_k \sum_{t=k_p}^{k-1} \eta_k^2 e_t + c_k,
$$

for all $k_p \le k \le v_p$. Under the conditions on $\beta$ and $\tau$, by Lemmas 13 and 14, we have

$$
a_{v_p+1} \le \left( \prod_{k=k_p}^{v_p} \Delta_k \right) a_{k_p} + \sum_{k=k_p}^{v_p-1} \left( \prod_{i=k+1}^{v_p} \Delta_i \right) c_k + c_{v_p}. \tag{40}
$$

Let $z_k = (k+b)^2$, where $b = \beta\tau + 1$. Then

$$\Delta_k \frac{z_k}{\eta_k} = (1 - \mu\eta_k)\mu(k+b)^3 = (1 - \frac{1}{k+b})\mu(k+b)^3 = \mu(k+b-1)(k+b)^2 \leq \mu(k+b-1)^3 = \frac{z_{k-1}}{\eta_{k-1}}$$

and, thus,

$$\frac{z_{v_p}}{\eta_{v_p}}\left(\prod_{i=k+1}^{v_p}\Delta_i\right) = \frac{z_{v_p}}{\eta_{v_p}}\Delta_{v_p}\left(\prod_{i=k+1}^{v_p-1}\Delta_i\right) \leq \frac{z_{v_p-1}}{\eta_{v_p-1}}\left(\prod_{i=k+1}^{v_p-1}\Delta_i\right)\ldots \leq \frac{z_k}{\eta_k}.$$

Note that $v_p + 1 = k_{p+1}$. Plugging the above inequality into (40), we then get

$$\frac{z_{v_p}}{\eta_{v_p}}a_{k_{p+1}} \leq \frac{z_{k_p}}{\eta_{k_p}}a_{k_p} + \sum_{k=k_p}^{v_p}\frac{z_k}{\eta_k}c_k.$$

Since we have assumed that $K = k_{\bar{p}}$, we thus have

$$\frac{z_{K-1}}{\eta_{K-1}}a_K = \frac{z_{v_{\bar{p}-1}}}{\eta_{v_{\bar{p}-1}}}a_{k_{\bar{p}}} \leq \frac{z_{k_{\bar{p}-1}}}{\eta_{k_{\bar{p}-1}}}a_{k_{\bar{p}-1}} + \sum_{t=k_{\bar{p}-1}}^{v_{\bar{p}-1}}\frac{z_t}{\eta_t}c_t \ldots \leq \frac{z_0}{\eta_0}a_0 + \sum_{k=0}^{K-1}\frac{z_k}{\eta_k}c_k. \tag{41}$$

Since, for $k_p \leq k \leq v_p$, we have

$$c_k = \frac{\eta_k L^2}{2}\left\{\sigma_{\text{dif}}^2(\tau-1)(C_1+1)\sum_{k=k_p}^{t-1}\eta_k^2 + \frac{\sigma_1^2(\tau-1)}{B}\sum_{k=k_p}^{t-1}\eta_k^2\right\}$$

$$+ \frac{\eta_k^2 L}{2}\left\{\sigma_{\text{dif}}^2\left(\frac{C_1}{M} + C_2 + 1\right) + \frac{\sigma_1^2}{MB} + \frac{\sigma_2^2}{B}\right\}$$

$$\leq \frac{\eta_k \eta_{\lfloor\frac{k}{\tau}\rfloor\tau}^2 L^2(\tau-1)^2}{2}\left\{\sigma_{\text{dif}}^2(C_1+1) + \frac{\sigma_1^2}{B}\right\}$$

$$+ \frac{\eta_k^2 L}{2}\left\{\sigma_{\text{dif}}^2\left(\frac{C_1}{M} + C_2 + 1\right) + \frac{\sigma_1^2}{MB} + \frac{\sigma_2^2}{B}\right\},$$

we also have

$$\sum_{k=0}^{K-1}\frac{z_k}{\eta_k}c_k \leq \frac{L^2(\tau-1)^2}{2}\left\{\sigma_{\text{dif}}^2(C_1+1) + \frac{\sigma_1^2}{B}\right\}\sum_{k=0}^{K-1}z_k\eta_{\lfloor\frac{k}{\tau}\rfloor\tau}^2$$

$$+ \frac{L}{2}\left\{\sigma_{\text{dif}}^2\left(\frac{C_1}{M} + C_2 + 1\right) + \frac{\sigma_1^2}{MB} + \frac{\sigma_2^2}{B}\right\}\sum_{k=0}^{K-1}z_k\eta_k. \tag{42}$$

Assume that $k = p\tau + r$, where $0 \leq r \leq \tau - 1$. Then

$$\left\lfloor\frac{t}{\tau}\right\rfloor\tau + b = p\tau + \beta\tau + 1 = (p+\beta)\tau + 1 \geq \beta\tau \geq r,$$

as we have assumed that $\beta > 1$. Thus

$$2\left(\left\lfloor\frac{k}{\tau}\right\rfloor\tau + b\right) \geq (p+\beta)\tau + 1 + r = k + b$$

and

$$\sum_{k=0}^{K-1}z_k\eta_{\lfloor\frac{k}{\tau}\rfloor\tau}^2 = \frac{1}{\mu^2}\sum_{k=0}^{K-1}\left(\frac{k+b}{\lfloor\frac{k}{\tau}\rfloor\tau + b}\right)^2 \leq \frac{4K}{\mu^2}.$$

Next, note that

$$\sum_{k=0}^{K-1} z_k \eta_k = \frac{1}{\mu} \sum_{k=0}^{K-1} (k+b) \leq \frac{K(K+2b)}{2\mu}. \tag{43}$$

Combining equations (41)-(43), we have

$$\mathbb{E}\left[\frac{1}{M} \sum_{m=1}^{M} f_m\left(\hat{\theta}_m^K\right) - f^*\right]$$

$$\leq \frac{b^3}{(K+\beta\tau)^3} \mathbb{E}\left[\frac{1}{M} \sum_{m=1}^{M} f_m\left(\hat{\theta}_m^0\right) - f^*\right] + \frac{2L^2(\tau-1)^2 K}{\mu^3(K+\beta\tau)^3}\left\{\sigma_{\text{dif}}^2(C_1+1) + \frac{\sigma_1^2}{B}\right\}$$

$$+ \frac{LK(K+2\beta\tau+2)}{4\mu^2(K+\beta\tau)^3}\left\{\sigma_{\text{dif}}^2\left(\frac{C_1}{M} + C_2 + 1\right) + \frac{\sigma_1^2}{MB} + \frac{\sigma_2^2}{B}\right\},$$

which completes the proof.

### B.7.3 Auxiliary Results

We give two technical lemmas that are used to prove Theorem 4.

**Lemma 13.** *Consider the sequence $\{a_k\}_{k_p \leq k \leq v_p}$ in the proof of Theorem 4 that satisfies*

$$a_{k+1} \leq \Delta_k a_k + \frac{\eta_k}{2}(-1 + D\eta_k)e_k + B_k \sum_{t=k_p}^{k-1} \eta_t^2 e_t + c_k,$$

*where $\Delta_k$, $B_k$, and $c_k$ are defined in (39). Suppose the sequence of learning rates $\{\eta_k\}$ satisfies*

$$\eta_{v_p} \leq D^{-1}, \tag{44}$$

$$\eta_{v_p-1} \leq \left(D + \frac{2B_{v_p}}{\Delta_{v_p}}\right)^{-1}, \tag{45}$$

$$\vdots$$

$$\eta_{k_p} \leq \left(D + \frac{2\left(B_{v_p} + \sum_{j=k_p+1}^{v_p-1} B_j \prod_{i=j+1}^{v_p} \Delta_i\right)}{\prod_{i=k_p+1}^{v_p} \Delta_i}\right)^{-1}. \tag{46}$$

*Then*

$$a_{v_p+1} \leq \left(\prod_{k=k_p}^{v_p} \Delta_k\right) a_{k_p} + \sum_{k=k_p}^{v_p-1}\left(\prod_{i=k+1}^{v_p} \Delta_i\right) c_k + c_{v_p}.$$

*Proof.* We start by noting that

$$a_{v_p+1} \leq \Delta_{v_p} a_{v_p} + \frac{\eta_{v_p}}{2}(-1 + D\eta_{v_p})e_{v_p} + B_{v_p} \sum_{k=k_p}^{v_p-1} \eta_k^2 e_k + c_{v_p}$$

$$\leq \Delta_{v_p} a_{v_p} + B_{v_p} \sum_{k=k_p}^{v_p-1} \eta_k^2 e_k + c_{v_p},$$

where the last inequality is due to (44). Thus, we have

$$a_{v_p+1} \leq \Delta_{v_p} a_{v_p} + B_{v_p} \sum_{k=k_p}^{v_p-1} \eta_k^2 e_k + c_{v_p}$$

$$= \Delta_{v_p} a_{v_p} + B_{v_p} \left( \sum_{k=k_p}^{v_p-2} \eta_k^2 e_k + \eta_{v_p-1}^2 e_{v_p-1} \right) + c_{v_p}$$

$$\leq \Delta_{v_p} \left( \Delta_{v_p-1} a_{v_p-1} + \frac{\eta_{v_p-1}}{2}(-1 + D\eta_{v_p-1})e_{v_p-1} + B_{v_p-1} \sum_{k=k_p}^{v_p-2} \eta_k^2 e_k + c_{v_p-1} \right)$$

$$+ B_{v_p} \left( \sum_{k=k_p}^{v_p-2} \eta_k^2 e_k + \eta_{v_p-1}^2 e_{v_p-1} \right) + c_{v_p}$$

$$= \Delta_{v_p} \Delta_{v_p-1} a_{v_p-1} + \frac{\eta_{v_p-1} \Delta_{v_p}}{2} \left[ -1 + D\eta_{v_p-1} + \frac{2B_{v_p}\eta_{v_p-1}}{\Delta_{v_p}} \right] e_{v_p-1}$$

$$+ \left( \Delta_{v_p} B_{v_p-1} + B_{v_p} \right) \sum_{k=k_p}^{v_p-2} \eta_k^2 e_k + \left( \Delta_{v_p} c_{v_p-1} + c_{v_p} \right).$$

By (45), we have

$$-1 + D\eta_{v_p-1} + \frac{2B_{v_p}\eta_{v_p-1}}{\Delta_{v_p}} \leq 0.$$

Therefore,

$$a_{v_p+1} \leq \Delta_{v_p} \Delta_{v_p-1} a_{v_p-1} + \left( \Delta_{v_p} B_{v_p-1} + B_{v_p} \right) \sum_{k=k_p}^{v_p-2} \eta_k^2 e_k + \left( \Delta_{v_p} c_{v_p-1} + c_{v_p} \right).$$

Under the assumptions on $\eta_k$, repeating the process above, we have

$$a_{v_p+1} \leq \left( \prod_{i=k_p+1}^{v_p} \Delta_i \right) a_{k_p+1}$$

$$+ \left[ \left( \prod_{i=k_p+2}^{v_p} \Delta_i \right) B_{k_p+1} + \left( \prod_{i=k_p+3}^{v_p} \Delta_i \right) B_{k_p+2} + \cdots + \Delta_{v_p} B_{v_p-1} + B_{v_p} \right] \eta_{k_p}^2 e_{k_p}$$

$$+ \sum_{k=k_p}^{v_p-1} \left( \prod_{i=k+1}^{v_p} \Delta_i \right) c_k.$$

Since

$$a_{k_p+1} \leq \Delta_{k_p} a_{k_p} + \frac{\eta_{k_p}}{2}(-1 + D\eta_{k_p})e_{k_p} + c_{k_p},$$

combining with (46), the final result follows. $\qquad\square$

**Lemma 14.** *Let* $\eta_k = (\mu(k + \beta\tau + 1))^{-1}$ *where*

$$\beta > \max \left\{ \frac{2\lambda L}{\mu} \left( \frac{C_1}{M} + C_2 + 1 \right) - 2, \frac{2L^2\lambda(C_1 + 1)}{\mu^2} \right\}.$$

*and*

$$\tau \geq \sqrt{\frac{\max \left\{ (2L^2\lambda(C_1 + 1)/\mu^2)e^{1/\beta} - 4, 0 \right\}}{\beta^2 - (2L^2\lambda(C_1 + 1)/\mu^2)e^{\frac{1}{\beta}}}}.$$

*Then the conditions in Lemma 13 are satisfied for* $\eta_k$ *for all* $k \geq 0$.

*Proof.* Let $\Delta_k$ and $B_k$ be defined as in (39). Since $\Delta_k < 1$ for all $k$, after $p$-th communication, for the right hand side of (46), we have

$$\left( D + \frac{2 \left( B_{v_p} + \sum_{j=k_p+1}^{v_p-1} B_j \prod_{i=j+1}^{v_p} \Delta_i \right)}{\prod_{i=k_p+1}^{v_p} \Delta_i} \right)^{-1}$$

$$\leq \left(D + \frac{2\left(B_{v_p} + \sum_{j=k_p+2}^{v_p-1} B_j \prod_{i=j+1}^{v_p} \Delta_i\right)}{\prod_{i=k_p+1}^{v_p} \Delta_i}\right)^{-1}$$

$$\leq \left(D + \frac{2\left(B_{v_p} + \sum_{j=k_p+2}^{v_p-1} B_j \prod_{i=j+1}^{v_p} \Delta_i\right)}{\prod_{i=k_p+2}^{v_p} \Delta_i}\right)^{-1}.$$

Thus, by induction, we have

$$\left(D + \frac{2\left(B_{v_p} + \sum_{j=k_p+1}^{v_p-1} B_j \prod_{i=j+1}^{v_p} \Delta_i\right)}{\prod_{i=k_p+1}^{v_p} \Delta_i}\right)^{-1}$$

$$\leq \left(D + \frac{2\left(B_j + \sum_{j=k_p+2}^{v_p-1} B_j \prod_{i=j+1}^{v_p} \Delta_i\right)}{\prod_{i=k_p+2}^{v_p} \Delta_i}\right)^{-1} \tag{47}$$

$$\leq \dots$$

$$\leq \left(D + \frac{2B_{v_p}}{\Delta_{v_p}}\right)^{-1}$$

$$\leq D^{-1}.$$

As $k$ increases, we have that $\eta_k$ decreases, $\Delta_k$ increases, and $B_k$ decreases. Thus, for $1 \leq k \leq K$, we have $\eta_K \leq \eta_{K-1} \leq \dots \leq \eta_1$. On the other hand, we can lower bound the right hand side of (46) as

$$\left(D + \frac{2\left(B_{v_p} + \sum_{j=k_p+1}^{v_p-1} B_j \prod_{i=j+1}^{v_p} \Delta_i\right)}{\prod_{i=k_p+1}^{v_p} \Delta_i}\right)^{-1}$$

$$\geq \left(D + \frac{2\left(B_1 + \sum_{j=k_p+1}^{v_p-1} B_1 \prod_{i=j+1}^{v_p} \Delta_K\right)}{\prod_{i=k_p+1}^{v_p} \Delta_1}\right)^{-1}$$

$$\geq \left(D + \frac{2B_1\left(1 + \sum_{j=k_p+1}^{v_p-1} \Delta_K^{v_p-j}\right)}{\Delta_1^{\tau-1}}\right)^{-1} \tag{48}$$

$$\geq \left(D + \frac{2B_1\left(1 + \sum_{j=k_p+1}^{v_p-1} 1\right)}{\Delta_1^{\tau-1}}\right)^{-1}$$

$$= \left(D + \frac{2B_1(\tau-1)}{\Delta_1^{\tau-1}}\right)^{-1}.$$

If

$$\eta_1 \leq \frac{1}{D + \frac{2B_1(\tau-1)}{\Delta_1^{\tau-1}}}, \tag{49}$$

then the conditions on stepsizes in Lemma 13 are satisfied for all $\eta_k$ by combining (47)-(49). Thus, we only need to show that (49) is satisfied to complete the proof.

To that end, we need to have

$$\left(D + \frac{2B_1(\tau-1)}{\Delta_1^{\tau-1}}\right)\tau_1 \leq 1$$

$$\iff \left(\lambda L\left(\frac{C_1}{M} + C_2 + 1\right) + \frac{\eta_1 L^2 \lambda (\tau-1)^2 (C_1+1)}{(1-\eta_1\mu)^{\tau-1}}\right)\eta_1 \leq 1$$

$$\iff \lambda L \left( \frac{C_1}{M} + C_2 + 1 \right)(1 - \eta_1 \mu)^{\tau-1} + \eta_1 L^2 \lambda (\tau - 1)^2 (C_1 + 1) \le \frac{(1 - \eta_1 \mu)^{\tau-1}}{\eta_1}.$$

To satisfy the above equation, we need

$$\begin{cases} \lambda L \left( \frac{C_1}{M} + C_2 + 1 \right)(1 - \eta_1 \mu)^{\tau-1} & \le (2\eta_1)^{-1}(1 - \eta_1 \mu)^{\tau-1} \\ \eta_1 L^2 \lambda (\tau - 1)^2 (C_1 + 1) & \le (2\eta_1)^{-1}(1 - \eta_1 \mu)^{\tau-1}. \end{cases} \tag{50}$$

Note that $\eta_1 = 1/(\mu(\beta\tau + 2))$. Thus, to satisfy the first inequality in (50), we need

$$2\lambda L \left( \frac{C_1}{M} + C_2 + 1 \right) \le \frac{1}{\eta_1} = \mu(\beta\tau + 2).$$

Since $\mu(\beta\tau + 2) \ge \mu(\beta + 2)$, the condition above follows if

$$\beta \ge \frac{2\lambda L}{\mu} \left( \frac{C_1}{M} + C_2 + 1 \right) - 2. \tag{51}$$

Next, to satisfy the second inequality in (50), we need

$$2\eta_1^2 L^2 \lambda (\tau - 1)^2 (C_1 + 1) \le (1 - \eta_1 \mu)^{\tau-1}$$

$$\iff \frac{2L^2 \lambda (C_1 + 1)}{\mu^2} \left( \frac{\tau - 1}{\beta\tau + 2} \right)^2 \left( \frac{\beta\tau + 2}{\beta\tau + 1} \right)^{\tau-1} \le 1.$$

Since

$$\left( \frac{\beta\tau + 2}{\beta\tau + 1} \right)^{\tau-1} = \left( 1 + \frac{1}{\beta\tau + 1} \right)^{\tau-1} = \left( 1 + \frac{(\tau - 1)/(\beta\tau + 1)}{\tau - 1} \right)^{\tau-1} \le \exp \left\{ \frac{\tau - 1}{\beta\tau + 1} \right\} \le e^{\frac{1}{\beta}},$$

we need

$$\frac{2L^2 \lambda (C_1 + 1)}{\mu^2} \left( \frac{\tau - 1}{\beta\tau + 2} \right)^2 e^{\frac{1}{\beta}} \le 1.$$

Let $\nu = 2L^2 \lambda (C_1 + 1)/\mu^2$. Then the above equation is equivalent to

$$(\beta^2 - \nu e^{\frac{1}{\beta}})\tau^2 + 2(\beta + \nu e^{\frac{1}{\beta}})\tau + (4 - \nu e^{\frac{1}{\beta}}) \ge 0.$$

First, we let $\beta^2 - \nu e^{\frac{1}{\beta}} > 0$ or equivalently

$$\frac{\beta^2}{e^{\frac{1}{\beta}}} > \frac{2L^2 \lambda (C_1 + 1)}{\mu^2}. \tag{52}$$

Then we need $\tau$ to be large enough such that

$$\tau \ge \frac{-2(\beta + \nu e^{\frac{1}{\beta}}) + \sqrt{4(\beta + \nu e^{\frac{1}{\beta}})^2 - \max \left\{ 4(\beta^2 - \nu e^{\frac{1}{\beta}})(4 - \nu e^{\frac{1}{\beta}}), 0 \right\}}}{2(\beta^2 - \nu e^{\frac{1}{\beta}})}.$$

Since $\sqrt{a^2 + b} \le |a| + \sqrt{|b|}$ for any $a, b \in \mathbb{R}$, the left hand side is smaller or equal to

$$\sqrt{\frac{\max \left\{ \nu e^{1/\beta} - 4, 0 \right\}}{\beta^2 - \nu e^{\frac{1}{\beta}}}} = \sqrt{\frac{\max \left\{ (2L^2 \lambda (C_1 + 1)/\mu^2)e^{1/\beta} - 4, 0 \right\}}{\beta^2 - (2L^2 \lambda (C_1 + 1)/\mu^2)e^{\frac{1}{\beta}}}}.$$

Therefore, we need

$$\tau \ge \sqrt{\frac{\max \left\{ (2L^2 \lambda (C_1 + 1)/\mu^2)e^{1/\beta} - 4, 0 \right\}}{\beta^2 - (2L^2 \lambda (C_1 + 1)/\mu^2)e^{\frac{1}{\beta}}}}. \tag{53}$$

The final result follows from the combination of (51)-(53). $\qquad\square$

## B.8 Proof of Theorem 6

**Nesterov's worst case objective. (Nesterov, 2018)** Let $h' : \mathbb{R}^\infty \to \mathbb{R}$ be the Nesterov's worst case objective (see), i.e., $h'(y) = \frac{1}{2} y^\top A y - e_1^\top y$ with tridiagonal $A$ having diagonal elements equal to $2 + c$ (for some $c > 0$) and offdiagonal elements equal to $1$.[5] The proof rationale is to show that a $k$-th iterate of any first order method must satisfy $\|y^k\|_0 \le k$ and consequently

$$\|y^k - y^*\|^2 \ge \left(\frac{\sqrt{\kappa} - 1}{\sqrt{\kappa} + 1}\right)^{2k} \|y^*\|^2 \tag{54}$$

where $y^* := \arg\min_{y \in \mathbb{R}^\infty} h'(y)$, $\kappa := \frac{\lambda_{\max}(A)}{\lambda_{\min}(A)}$.

**Finite sum worst case objective. (Lan & Zhou, 2018)** The construction of the worst case finite-sum objective[6] $h : \mathbb{R}^\infty \to \mathbb{R}, h(z) = \frac{1}{n} \sum_{j=1}^n h_j(z)$ is such that $h_j$ corresponds only on a $j$-th block of the coordinates; in particular if $z = [z_1, z_2, \ldots, z_n]$; $z_1, z_2, \ldots, z_n \in \mathbb{R}^\infty$ we set $h_j(z) = h'(z_j)$. It was shown that to reach $\|z^k - z^*\|^2 \le \epsilon$ one requires at least $\Omega\left(\left(n + \sqrt{\frac{n\mathcal{L}}{\mu}}\right) \log \frac{1}{\epsilon}\right)$ iterations for $\mathcal{L}$-smooth functions $h_j$ and $\mu$−strongly convex $h$.

**Distributed worst case objective. (Scaman et al., 2018)** Define

$$g_1'(z) := \frac{1}{2} \left(c_1 \|z\|^2 + c_2 \left(e_1^\top z + z^\top M_1 z\right)\right)$$

$$g_2'(z) = g_3'(z) = \cdots = g_M'(z) := \frac{1}{2(M-1)} \left(c_1 \|z\|^2 + c_2 z^\top M_2 z\right)$$

where $M_1$ is an infinite block diagonal matrix with blocks $\begin{pmatrix} 1 & 1 & 0 & 0 \\ 1 & 2 & 1 & 0 \\ 0 & 1 & 1 & 0 \\ 0 & 0 & 0 & 0 \end{pmatrix}$ and $M_2 := \begin{pmatrix} 1 & 0 & \\ 0 & 0 & \\ & & M_1 \end{pmatrix}$ and

$c_1, c_2 > 0$ are some constants determining the smoothness and strong convexity of the objective. The worst case objective of Scaman et al. (2018) is now $g(z) = \frac{1}{M} \sum_{m=1}^m g_m'(z)$.

**Distributed worst case objective with local finite sum. (Hendrikx et al., 2021)** The given construction is obtained from the one of Scaman et al. (2018) in the same way as the worst case finite sum objective (Lan & Zhou, 2018) was obtained from the construction of Nesterov (2018). In particular, one would set $g_{m,j}(z) = g_m'(z_j)$ where $z = [z_1, z_2, \ldots, z_n]$. Next, it was shown that such a construction with properly chosen $c_1, c_2$ yields a lower bound on the communication complexity of order $\Omega\left(\sqrt{\frac{L}{\mu}} \log \frac{1}{\epsilon}\right)$ and the lower bound on the local computation of order $\Omega\left(\left(n + \sqrt{\frac{n\mathcal{L}}{\mu}}\right) \log \frac{1}{\epsilon}\right)$ where $\mathcal{L}$ is a smoothness constant of $g_{m,j}$, $L$ is a smoothness constant of $g_m(z) = \frac{1}{n} \sum_{j=1}^n g_j(z)$ and $\mu$ is the strong convexity constant of $g(z) = \frac{1}{M} \sum_{m=1}^M g_m(z)$.

**Our construction and sketch of the proof.** Now, our construction is straightforward – we set $f_m(w, \beta_m) = g(w) + h(\beta_m)$ with $g$, $h$ scaled appropriately such that the strong convexity ratio is as per Assumption 1. Clearly, to minimize the global part $g(w)$, we require at least $\Omega\left(\sqrt{\frac{L^w}{\mu}} \log \frac{1}{\epsilon}\right)$ iterations and at least $\Omega\left(\left(n + \sqrt{\frac{n\mathcal{L}^w}{\mu}}\right) \log \frac{1}{\epsilon}\right)$ stochastic gradients of $g$. Similarly, to minimize $h$, we require at least $\Omega\left(\left(n + \sqrt{\frac{n\mathcal{L}^\beta}{\mu}}\right) \log \frac{1}{\epsilon}\right)$ stochastic gradients of $h$. Therefore, Theorem 6 is established.

---

[5]This is for the strongly convex case; one can do convex similarly.

[6]We have lifted their construction to the infinite-dimensional space for the sake of simplicity. One can get a similar finite-dimensional results.

### B.9 Proof of Theorem 8

Taking the stochastic gradient step followed by the proximal step with respect to $\psi$, both with stepsize $\eta$, is equivalent to (Hanzely et al., 2020b):

$$
\begin{aligned}
\text{w.p. } p_w: \quad &
\begin{cases}
w^+ = w - \eta \left( \dfrac{1}{p_w} \left( \dfrac{1}{M} \sum_{m=1}^{M} \nabla_w f_{m,j}(w, \beta_m) - \dfrac{1}{M} \sum_{m=1}^{M} \nabla_w f_{m,j}(w', \beta'_m) \right) \right. \\
\qquad \left. + \nabla_w F(w', \beta') \right), \\
\beta_m^+ = \beta_m - \frac{\eta}{M} \nabla f_m(w', \beta'_m)
\end{cases} \\[2mm]
\text{w.p. } p_\beta: \quad &
\begin{cases}
w^+ = w - \eta \nabla_w F(w', \beta'), \\
\beta_m^+ = \beta_m - \frac{\eta}{M} \left( \frac{1}{p_\beta} \left( \nabla_\beta f_{m,j}(w, \beta_m) - \nabla_\beta f_{m,j}(w', \beta'_m) \right) + \nabla_\beta f_m(w', \beta'_m) \right).
\end{cases}
\end{aligned}
\tag{55}
$$

Let $x = [w, \beta_1, \ldots, \beta_M]$, $x' = [w', \beta'_1, \ldots, \beta'_M]$. The update rule (55) can be rewritten as

$$
x^+ = x - \eta \left( g(x) - g(x') + \nabla F(x') \right),
$$

where $g(x)$ corresponds to the described unbiased stochastic gradient obtained by subsampling both the space and the finite sum simultaneously. To give the rate of the aforementioned method, we shall determine the expected smoothness constant. To achieve that, we introduce the following two lemmas.

**Lemma 15.** *Suppose that Assumptions 1 and 2 hold. Then*

$$
\mathbb{E}\|(g(x) - g(x') + \nabla F(x')) - \nabla F(x)\|^2 \leq 2\mathcal{L} D_F(x, y),
$$

*where $\mathcal{L} := 2 \max \left( \frac{\mathcal{L}^w}{p_w}, \frac{\mathcal{L}^\beta}{p_\beta} \right)$.*

*Proof.* Let $d_\beta := \sum_{m=1}^{m} d_m$. We have:

$$
\begin{aligned}
& \mathbb{E}\|(g(x) - g(x') + \nabla F(x')) - \nabla F(x)\|^2 \\
& \quad \leq \mathbb{E}\|g(x) - g(x')\|^2 \\
& \quad = p_w \mathbb{E} \left\| p_w^{-1} \frac{1}{M} \sum_{m=1}^{M} (\nabla_w f_{m,j}(w, \beta_m) - \nabla_w f_{m,j}(w', \beta'_m)) \right\|^2 \mid \zeta = 1 \\
& \qquad + p_\beta \frac{1}{M^2} \sum_{m=1}^{M} \mathbb{E}\| p_\beta^{-1} \nabla f_{m,j}(w, \beta_m) - p_\beta^{-1} \nabla f_{m,j}(w', \beta'_m) \|^2 \mid \zeta = 2 \\
& \quad = p_w^{-1} \mathbb{E} \left\| \frac{1}{M} \sum_{m=1}^{M} (\nabla_w f_{m,j}(w, \beta_m) - \nabla_w f_{m,j}(w', \beta'_m)) \right\|^2 \mid \zeta = 1 \\
& \qquad + p_\beta^{-1} \frac{1}{M^2} \sum_{m=1}^{M} \mathbb{E}\| \nabla f_{m,j}(w, \beta_m) - \nabla f_{m,j}(w', \beta'_m) \|^2 \mid \zeta = 2 \\
& \quad = \mathbb{E}(F_j(x) - \nabla F_j(x'))^\top \begin{pmatrix} p_w^{-1} I^{d_0 \times d_0} & 0 \\ 0 & p_\beta^{-1} I^{d_\beta \times d_\beta} \end{pmatrix} (F_j(x) - \nabla F_j(x')) \\
& \quad \overset{(*)}{\leq} \mathbb{E} 4 \max \left( \frac{\mathcal{L}^w}{p_w}, \frac{L^\beta}{p_\beta} \right) D_{F_j}(x, x') \\
& \quad = 4 \max \left( \frac{\mathcal{L}^w}{p_w}, \frac{L^\beta}{p_\beta} \right) D_F(x, x'),
\end{aligned}
$$

where $(*)$ holds due to the $(\mathcal{L}^w, \mathcal{L}^\beta)$-smoothness of $F_j$ (from Assumption 2) and Lemma 16. $\qquad \square$

**Lemma 16.** *Let $H(x, y) : \mathbb{R}^{d_x + d_y} \to \mathbb{R}$ be a (jointly) convex function such that*

$$\nabla_x^2 H(x, y) \preceq L_x \boldsymbol{I} \quad and \quad \nabla_y^2 H(x, y) \preceq L_y \boldsymbol{I}.$$

*Then*

$$\nabla^2 H(x, y) \preceq 2 \begin{pmatrix} L_x \boldsymbol{I} & 0 \\ 0 & L_y \boldsymbol{I} \end{pmatrix} \tag{56}$$

*and*

$$D_H((x, y), (x'y'))$$

$$\geq \frac{1}{2} \left( \nabla H(x, y) - \nabla H(x', y') \right)^\top \begin{pmatrix} \frac{1}{2} L_x^{-1} \boldsymbol{I} & 0 \\ 0 & \frac{1}{2} L_y^{-1} \boldsymbol{I} \end{pmatrix} \left( \nabla H(x, y) - \nabla H(x', y') \right). \tag{57}$$

*Proof.* To show (56), observe that

$$
\begin{aligned}
2 \begin{pmatrix} L_x \boldsymbol{I} & 0 \\ 0 & L_y \boldsymbol{I} \end{pmatrix} - \nabla^2 H(x, y) &= \begin{pmatrix} 2 L_x \boldsymbol{I} - \nabla_{x,x}^2 H(x, y) & -\nabla_{x,y}^2 H(x, y) \\ -\nabla_{y,x}^2 H(x, y) & 2 L_y \boldsymbol{I} - \nabla_{y,y}^2 H(x, y) \end{pmatrix} \\
&\succeq \begin{pmatrix} \nabla_{x,x}^2 H(x, y) & -\nabla_{x,y}^2 H(x, y) \\ -\nabla_{y,x}^2 H(x, y) & \nabla_{y,y}^2 H(x, y) \end{pmatrix} \\
&\succeq \nabla_{x,x}^2 H(x, -y) \\
&\succeq 0.
\end{aligned}
$$

Finally, we note that (57) is a direct consequence of (56) and joint convexity of $H$. $\qquad \square$

We are now ready to state the convergence rate of ASVRCD-PFL.

**Theorem 9.** *Iteration complexity of Algorithm 3 with*

$$\eta = \frac{1}{4\mathcal{L}}, \quad \theta_2 = \frac{1}{2}, \quad \gamma = \frac{1}{\max\{2\mu, 4\theta_1/\eta\}},$$

$$\nu = 1 - \gamma\mu, \quad and \quad \theta_1 = \min\left\{ \frac{1}{2}, \sqrt{\eta\mu \max\left\{ \frac{1}{2}, \frac{\theta_2}{\rho} \right\}} \right\}$$

*is*

$$\mathcal{O}\left( \left( \frac{1}{\rho} + \sqrt{\frac{\max\left( \frac{\mathcal{L}^w}{p_w}, \frac{L^\beta}{p_\beta} \right)}{\rho\mu}} \right) \log \frac{1}{\epsilon} \right).$$

*Setting $p_w = \frac{\mathcal{L}^w}{\mathcal{L}^\beta + \mathcal{L}^w}$ yields the complexity*

$$\mathcal{O}\left( \left( \frac{1}{\rho} + \sqrt{\frac{\mathcal{L}^w + \mathcal{L}^\beta}{\rho\mu}} \right) \log \frac{1}{\epsilon} \right).$$

*Proof.* The proof follows from Lemma 15 and Theorem 4.1 of Hanzely et al. (2020b), thus is omitted. $\quad \square$

Overall, the algorithm requires

$$\mathcal{O}\left( \left( \frac{1}{\rho} + \sqrt{\frac{\mathcal{L}^w + \mathcal{L}^\beta}{\rho\mu}} \right) \left( \log \frac{1}{\epsilon} \right) (\rho n + p_w) \right)$$

communication rounds and the same number of gradient calls w.r.t. parameter $w$. Setting $\rho = \frac{p_w}{n}$, we have

$$
\left( \frac{1}{\rho} + \sqrt{\frac{\mathcal{L}^w + \mathcal{L}^\beta}{\rho\mu}} \right) \left( \log \frac{1}{\epsilon} \right) (\rho n + p_w) = 2 \left( \frac{1}{\rho} + \sqrt{\frac{\mathcal{L}^w + \mathcal{L}^\beta}{\rho\mu}} \right) \left( \log \frac{1}{\epsilon} \right) \rho n
$$

$$
= 2 \left( n + \sqrt{\frac{\rho n^2 (\mathcal{L}^w + \mathcal{L}^\beta)}{\mu}} \right) \left( \log \frac{1}{\epsilon} \right)
$$

$$
= 2 \left( n + \sqrt{\frac{n\mathcal{L}^w}{\mu}} \right) \left( \log \frac{1}{\epsilon} \right),
$$

which shows that Algorithm 3 enjoys both communication complexity and the global gradient complexity of order $\mathcal{O}\left( \left( n + \sqrt{\frac{n\mathcal{L}^w}{\mu}} \right) \log \frac{1}{\epsilon} \right)$. Analogously, setting $\rho = \frac{p_\beta}{n}$ yields personalized/local gradient complexity of order $\mathcal{O}\left( \left( n + \sqrt{\frac{n\mathcal{L}^\beta}{\mu}} \right) \log \frac{1}{\epsilon} \right)$.

