# OpenReview forum: "Personalized Federated Learning: A Unified Framework and Universal Optimization Techniques"
_TMLR — Accepted by TMLR_

### Review · Reviewer_qSw9 · 2022-12-31

**Summary Of Contributions:**

This work proposes a unified framework for studying optimization aspects of many personalized federated learning objectives and corresponding algorithms. They work under the formulation where each users' objective function depends on two sets of variables --- shared variables ($w$) and personal variables ($\beta$), and the global objective is to minimize the average of each users' objectives. Many of the proposed personalized FL algorithms fit in this framework. The main contributions of this paper are to recover known guarantees of these algorithms which fit in this framework and obtain (as a corollary to the guarantees of their framework) a few new guarantees for previous work where optimization guarantees were not provided. They also provide lower bounds on the number of gradients required for both the personal and shared variables and on the communication complexity, by a simple combination of previous results to construct the hard instance. Lastly, they provide an accelerated and variance reduced version of the basic local SGD type algorithm to achieve minimax optimal sample complexity both in terms of number of gradients of the shared and the personal variables, though not both simultaneously.

**Audience:**

Yes

**Broader Impact Concerns:**

No concerns for broader impact.

**Claims And Evidence:**

Yes

**Requested Changes:**

Minor points:
1. \hat{f} is not defined in page 3 section 1.2, under local objectives.
2. In table 2, instead of writing equation 4,5, etc. it would be more fruitful to write algorithm names and paper references. Same with table 3.

Major points:
1. The weakness points 2 and 3 mentioned above, I would like to understand the theorem statements clearly to evaluate this work. Not just the parameters settings but examples showing when they are reasonable would be helpful.
2. I think the claim that this is a unified framework for all of personalized FL is a bit too strong, equation 1 is one way to combine all the objectives of all the users, certainly not the only or the best way. I'd recommend changing the title to make it more realistic, currently it sounds like it is the paper which encompasses all of personalized FL.
3. Weakness 4 mentioned above, I don't expect technical changes, but upfrontness in writing in the earlier parts of the paper.

A point I didn't fully understand, maybe I missed it:
There is a lot of mention about reparameterization of w space being important to make LSGD work, but I don't see where the reparameterization is happening in the algorithm. Can you shed some light on where this is happening?


**Strengths And Weaknesses:**

Strengths:
1. The paper is largely clearly written and a good amount of research is done into finding algorithms and formulations that fit well within the proposed framework.
2. The calculation of the global and local smoothness constants for different formulations is clear and places the previous work very well in context of this framework.
3. Convergence guarantees are provided for the proposed algorithms and it is made explicit when they are or are not minimax optimal.

Weaknesses:
1. Most of the theorems are simple modifications to previous work, and the key contribution seems to be observing the similarity in a lot of previous work and proposing a natural generalization which encompasses a lot of the work. Although it leads to a clearer understanding, I find this contribution to be more pedagogical rather than it being research.
2. Some conditions are just stated as complex equations, and no example values are provided after the theorem illustrating reasonable values for the variable where these assumptions may be satisfied (e.g. eqn 20 of Theorem 3, and in Theorem 4). I think this is a major weakness for two reasons - one cannot easily verify whether there is any setting of parameters which satisfies the conditions and still gives reasonable convergence and other, while implementing the algorithm, it is important to know what you can set the algorithm parameters as which is hard to check with complicated equations.
3. For SVRCD type algorithms, the setting of all hyperparameters is not defined in the main body and it is only set in the proof. It is important to know what hyperparameters get the rates without combing through the proof both for a reader to implement these algorithms and run sanity checks.
4. ASVRCD achieves minimax optimal rates in terms of gradient complexity for shared variables and personalized variables for different parameter settings, thus claiming it is minimax optimal is misleading because its two *different* algorithms that are minimax optimal in different aspects.

---

> ### Author Response · Authors · 2023-03-11
> **Response to Reviewer qSw9**
>
> Thank you for your efforts in reviewing our paper. In the following, we would like to address your concerns.
>
> **Minor points.** We have clarified the definition of $\hat{f}_m$ below the equation (2). In addition, based on your suggestion, we changed the first column of Table 2 and Table 3 to objective names and paper references. Please see the updated submission.
>
> **The weakness points 2 and 3.** First, we have incorporated your suggestion and added the choices of hyperparameters into Theorem 8. Secondly, we would like to point out that as long as the stepsize is chosen small enough, the conditions such as equation (20) can always be satisfied, making these assumptions easy to meet. Additionally, this type of assumption that the stepsize needs to be smaller than some functions of problem parameters is standard and common in optimization literature. For example, please refer to Theorem V in Appendix D.2 of [1]. While these functions may seem complicated, one can always find optimization parameters that satisfy the conditions, such as by choosing a small enough step size. Finally, the more challenging issue is to choose the largest learning rate that satisfies the conditions, which usually depends on unknown problem parameters. For most cases, these problem parameters are difficult to estimate. In such cases, people usually try a set of hyperparameters and choose the one that yields the best practical performance.
>
>
>
> **The claim is too strong.** We have carefully examined the paper and removed the claims that we can recover all objectives. Instead, we state that we can recover many existing objectives. Additionally, we added the following paragraph at the end of Section 1.1 to emphasize this point:
>
> *Despite the aforementioned benefits of our proposed unified framework, we acknowledge that this is neither the only nor the universally best approach for personalized federated learning. However, providing a general framework that can include many existing methods as special cases can help us get a clear understanding and motivate us to propose new personalized methods.*
>
> **Weakness 4.** Thanks for the suggestion. We have emphasized this point in contribution section when stating Minimax optimal rates.
>
> **Reparameterization.** The reparameterization happens when we formulate the objective functions. For example, in equations (8), (11), (14), (16) and (18), we divide the global parameter $w$ by $\sqrt{M}$.
>
> Please see the updated submission for more details.
>
> [1] Karimireddy, Sai Praneeth, et al. "Scaffold: Stochastic controlled averaging for federated learning." International Conference on Machine Learning. PMLR, 2020.

---

### Review · Reviewer_D89j · 2023-01-26

**Summary Of Contributions:**

1. This paper proposes a general objective function for personalized federated learning, and the main idea is to optimize the global parameters and local parameters simultaneously, while most of the previous work only focus on some parts of the problem. The proposed general objective function can recover various existing FL problems as special cases, and it also leads to several new/more challenging problems.

2. The authors develop several optimization methods for the proposed general objective function. The first algorithm, LSGD-PFL, is a variant of LSGD, and the main difference is that the local servers also optimize the local parameters via SGD. The authors prove that this algorithm enjoys minimax optimal bound. The then propose accelerated algorithms based on variance reduction.


3. Finally, the authors provide experimental results to demonstrate the effectiveness of the proposed objective function and algorithms.

**Audience:**

Yes

**Claims And Evidence:**

Yes

**Requested Changes:**

Equation 8, 12, 15, 16, 17, 19: I wonder if it should be sum_{m=1} instead of sum_{i=1}, or I missed something here?

For the new personalized FL objectives, it would be great if the authors can add more discussion and justify why they are meaningful with some real-world applications.

**Strengths And Weaknesses:**

Strengths:
This paper is generally well-written and easy to follow. The proposed framework provides a unified view for FL optimization and analysis, which covers many existing problems as special cases. Moreover, it also leads to several new problems, which may of independent interests. The algorithms proposed in this paper, such as LSGD-PFL, although are variants of existing algorithms, are novel and significant enough to me, as it is not obvious (for me) that the right convergence rate can be obtained easily by simply minimizing the parameters simultaneously. The authors also did a very solid work in the experiment section and many results are provided.

Weakness:

Previous work focus on optimizing local parameters or global parameters separately, and it is not very surprising to me that we can come up with a general framework which tries to optimize all parameters and thus can recover previous problems as special cases.

---

> ### Author Response · Authors · 2023-03-11
> **Response to Reviewer D89j**
>
> Thank you for your efforts in reviewing our paper. We have made the necessary corrections for Equation 8, 12, 15, 16, 17, and 19 --- $\sum^M_{i=1}$ should be $\sum^M_{m=1}$. We apologize for the error and thank you for bringing it to our attention.
> Regarding the significance of the extended personalized FL objectives in our paper, we would like to emphasize that our main focus is not proposing new personalized FL objectives. Instead, we provide a unified framework that offers universal optimizers. Therefore, our emphasis is on the optimization perspective rather than the statistical perspective, which is more relevant to the design of the objective function.
> However, in Section 6.4 of the updated submission, we have added some empirical observations to show why a more general framework can potentially be beneficial. Please refer to the updated submission for detailed discussions. We acknowledge that a rigorous study of the statistical benefits of the proposed objectives is essential, and we plan to explore this in future research. In fact, to tackle heterogeneity, many results in multitask literature might be helpful, which can also be of interest for future research.

---

### Review · Reviewer_AYaN · 2023-02-27

**Summary Of Contributions:**

In this work, the authors suggest a unified framework for personalized federated learning, i.e., an empirical risk minimization problem with local variables. They show that many existing works in personalized federated learning satisfy this framework. They further adapt three existing optimization algorithms to solve this personalized federated learning problem. Moreover, under strong assumptions on the objective (strong convexity and smoothness), they show that their adapted algorithms can recover best-known optimization guarantees, i.e. bound on the communication rounds. Some numerical experiments are provided to support their claims.


**Audience:**

Yes

**Claims And Evidence:**

Yes

**Requested Changes:**

Based on the weakness points listed above, I have the following suggestions:
1. use larger data sets such as CIFAR10
2. compare with some existing personalized federated learning algorithms

**Strengths And Weaknesses:**

Strength:
1. The unified personalized federated learning (PFL) framework is clear. Given there are many PFL works, it is nice to have such a unified framework.
2. Although the proposed algorithms are not new, I appreciate that the authors spent time proving that they can recover the best-known convergence rates.
3. The derivation of the lower complexity bound is also very important, and it is interesting to see that the Accelerated block Coordinate Descent algorithm is actually already optimal.

Weakness:
I think compared with the theoretical results, the numerical experiments are relatively weak. For example, the authors only use tiny real-data sets (MNIST, KMNIST, FMNIST). Also, although the authors have theoretically proven that they can recover best-known optimization guarantees, it would be great if they could provide numerical comparisons.

---

> ### Author Response · Authors · 2023-03-11
> **Response to Reviewer AYaN**
>
> Thank you for your efforts in reviewing our paper. We have incorporated your suggestions and added experiments on CIFAR10 in Section 6. Furthermore, in Section 5.1, we compared our proposed algorithms with two existing personalized federated learning algorithms, namely  L2SGD+ [1] and pFedMe [2]. Please refer to the updated submission for further details.
>
> [1] Filip Hanzely and Peter Richtárik. Federated learning of a mixture of global and local models. arXiv preprint
> arXiv:2002.05516, 2020.
>
> [2] Canh T Dinh, Nguyen Tran, and Tuan Dung Nguyen. Personalized federated learning with moreau envelopes.
> Advances in Neural Information Processing Systems, 33, 2020.

---

### Author Response · Authors · 2023-03-11
**Updated Submission is Uploaded**

We have uploaded an updated submission. Specifically, we added more experiments in Section 6, corrected typos and revised some writings and our contribution claim to make it more appropriate. Please see the updated submission for details. Thanks.

---

### Decision · Action_Editors · 2023-04-20

**Recommendation:** Accept as is

**Comment:**

This main contribution of this work is to proposes a unified framework for studying optimization aspects of many personalized federated learning objectives and corresponding algorithms. Many of the proposed personalized FL algorithms fit in this framework. The paper recovers known guarantees of these algorithms which fit in this framework and obtain a few new guarantees for previous work where optimization guarantees were not provided. They also provide some lower bounds. They further provide accelerated and variance reduced versions.

The reviewers found the paper well written and correct. Reviewers also appreciated the proposal of the unifying framework which helps the community. The reviewers found limited novelty though and found the experiments a little weak but were all overall supportive of acceptance.

**Audience:**

This paper would be of interest to folks studying Federated Learning.

**Claims And Evidence:**

The reviewers found the claims made in the submission to be supported by clear and convincing evidence.